# Mapping the Neuro-Symbolic AI Landscape by Architectures: A Handbook on Augmenting Deep Learning Through Symbolic Reasoning

## Abstract

Integrating symbolic techniques with statistical ones is a long-standing problem in artificial intelligence. The motivation is that the strengths of either area match the weaknesses of the other, and – by combining the two – the weaknesses of either method can be limited. Neuro-symbolic AI focuses on this integration where the statistical methods are in particular neural networks. In recent years, there has been significant progress in this research field, where neuro-symbolic systems outperformed logical or neural models alone. Yet, neuro-symbolic AI is, comparatively speaking, still in its infancy and has not been widely adopted by machine learning practitioners. In this survey, we present the first mapping of neuro-symbolic techniques into families of frameworks based on their architectures, with several benefits: Firstly, it allows us to link different strengths of frameworks to their respective architectures. Secondly, it allows us to illustrate how engineers can augment their neural networks while treating the symbolic methods as black-boxes. Thirdly, it allows us to map most of the field so that future researchers can identify closely related frameworks.

## 1 Introduction

Over the last decades, machine learning has achieved outstanding performance in pattern recognition across a range of applications. In particular, in the areas of computer vision (LeCun et al., 1998; Goodfellow et al., 2020; Dosovitskiy et al., 2021; Liu et al., 2022), natural language processing (NLP) (Hochreiter & Schmidhuber, 1997; Mikolov et al., 2013; Vaswani et al., 2017; Devlin et al., 2019), and recommendation systems (He et al., 2017; Zheng et al., 2018; Rashed et al., 2019), neural networks have outperformed more traditional machine learning models. These breakthroughs have led to significant progress in fields such as medicine (Kearnes et al., 2016; Xie et al., 2019; Huang et al., 2019), finance (Deng et al., 2016; Bao et al., 2017), and autonomous driving (Bojarski et al., 2016; Luo et al., 2018).

Despite these strides, purely neural models still have important limitations (Marcus, 2018). Five important limitations are particularly worth mentioning:

1. **Structured reasoning.** Neural networks are particularly suited for pattern recognition but do not lend themselves well to hierarchical or composite reasoning and do not differentiate between causality and correlation (Lake & Baroni, 2017).

2. **Data need.** To achieve robustness in the predictions of neural models, large amounts of data are needed (Halevy et al., 2009; Ba & Caruana, 2014). However, large datasets are unavailable in many applications, making neural networks an unviable choice.

3. **Knowledge integration.** Given that humans have extensive knowledge in many areas that we try to tackle with machine learning, it would be helpful to integrate this knowledge into models. However, neural networks do not easily support to integrate expert or even common sense knowledge (Davis & Marcus, 2015). Enabling knowledge integration would reduce the amount of required training data and the training cost.

4. **Explainability.** Neural networks are black-box systems. In other words, it is often impossible to understand how the model reached its predictions for a given input. This can be seen as the inverse of the preceding point. Neural networks are not only not amenable to integrating knowledge but also to extracting knowledge. Lack of explainability has serious consequences for ethics, security, and extending human knowledge (Ribeiro et al., 2016; Samek et al., 2017).

5. **Guarantees.** Neural networks compute a probability distribution over possible outcomes. Inherently, outcomes may be predicted that violate constraints, which can be consequential in safety-critical applications (Gopinath et al., 2018; Cardelli et al., 2019; Ruan et al., 2019).

While neural networks are arguably the more known models to the general public, logical models have been the more prominent research direction in artificial intelligence (AI) (Russell & Norvig, 2016) prior to the resurgence of neural networks (Schmidhuber, 2015). In contrast to neural models, logical models are particularly suited for symbolic reasoning tasks that depend on the ability to capture and identify relations and causality (Vennekens et al., 2009; De Raedt & Kimmig, 2015). However, logical models have their own set of limitations.[1] The first limitation of logical models is their inability to deal with uncertainty both in the data and in the theory as, traditionally, each proposition must either be true or false (Pearl, 1988). The second major bottleneck, which arguably has led to falling behind in popularity behind neural models, is scalability, as computational complexity generally grows exponentially with the size of the alphabet and the lengths of formulae in the logical theory (Bradley & Manna, 2007).

To tackle the first problem, the field of statistical relational learning (SRL) aims at unifying logical and probabilistic frameworks (Getoor & Taskar, 2007; Raedt et al., 2016). In fact, this unification has been a long-standing goal in machine learning (Russell, 2015). Logical notions capture objects, properties and relations, focusing on learning processes, dependencies and causality. On the other hand, the underlying probabilistic theory addresses uncertainty and noisy knowledge acquisition with a focus on learning correlations. In contrast to neural networks, SRL frameworks allow us to reason at a symbolic level, generalise from little data, and integrate domain expertise easily. In addition, they are easy to interpret as the logical theories are close to natural language. However, by combining models in logic and probability theory – two independently computationally-hard problems – SRL frameworks fail to scale well in general (Natarajan et al., 2015). As illustrated by Table 1, the weaknesses of one area are the strengths of the other, and thus, it should come as no surprise that further unification of these two areas is necessary.

To tackle the second limitation, building on the strengths of SRL, the area of neuro-symbolic AI takes this unification further by combining logical models with neural networks (d'Avila Garcez et al., 2022). This approach is taken as one reason neural networks have gained so much attention is their increased scalability compared to logical AI. The scalability has been supported through improved hardware, in particular, GPUs (Mittal & Vaishay, 2019) and hardware specifically designed for neural computation (Schuman et al., 2017). Often, as we will see throughout this survey, the symbolic component of a neuro-symbolic system is an SRL framework. Thus, neuro-symbolic AI builds heavily on SRL. On the one hand, neuro-symbolic AI leverages the power of deep neural networks, which offer a complex and high-dimensional hypothesis space. On the other hand, owing to hard constraints (e.g. in safety-critical applications), data efficiency, transferability (e.g. one-shot and zero-shot learning) and interpretability (e.g. program induction), symbolic constructs are increasingly seen as an explicit representation language for the output of neural networks.

In recent years, numerous neuro-symbolic frameworks have been proposed, delivering on some of the expectations the research community had, including improved accuracy (Gu et al., 2019; Mao et al., 2019; Zareian et al., 2020), reduced model complexity and data need by integrating background knowledge (Huang et al., 2021; Buffelli & Tsamoura, 2023; Feldstein et al., 2023a), or offering a more explainable architecture (Zhang et al., 2022).

However, despite these successes, machine learning practitioners have not yet widely adopted neuro-symbolic models. We believe that one reason for the low uptake is that neuro-symbolic AI requires knowledge of two very distinct areas – logic and neural networks. For that reason, we start this survey with a reasonably

---

[1]We use the term "logical models" to collectively refer to approaches for modelling systems using logic, including knowledge representation, verification and automated planning (Bradley & Manna, 2007).

| Statistical Relational Learning | Neural Networks |
|---|---|
| Symbolic reasoning | Pattern recognition |
| **Can generalise on limited data** | Data hungry |
| **Easy knowledge integration** | Difficult knowledge integration |
| Scales poorly | **Scales well** |
| **White-box system** | Black-box system |
| Poor robustness to noise | **High accuracy and robustness** |

Table 1: Opposing strengths and weaknesses of SRL and neural networks.

detailed background on logic, probability, and SRL. Then, this survey focuses mainly on neuro-symbolic frameworks that allow users to treat the symbolic models as black-boxes. We discuss the construction of the architectures and the benefits one can derive from those architectures.

## 2 Contributions and Structure of the Survey

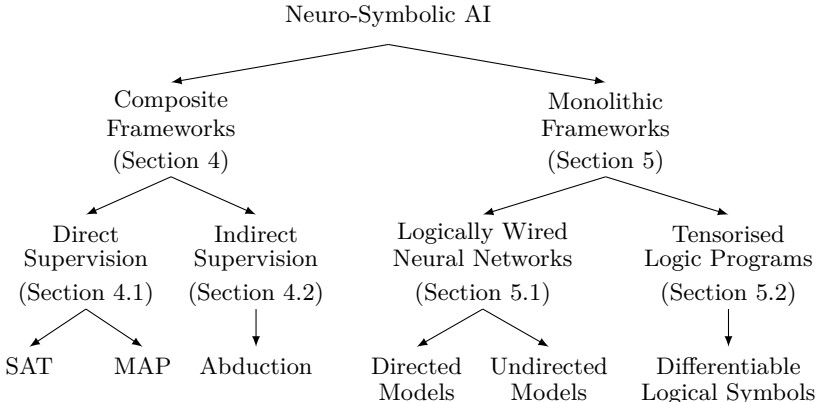

Figure 1: A map of neuro-symbolic frameworks based on their high-level architectures.

This section describes the contributions of this survey, from which we derive the structure of the remainder of this work.

**A gentle introduction to SRL.** Our first contribution is an in-depth, yet succinct, introduction to SRL in Section 3. To this end, we discuss various concepts of probability, logic, and SRL. We connect the different concepts through examples, which we continuously extend throughout this survey to illustrate the commonalities and differences between the methods. We aim to give enough details to enable a robust understanding of the frameworks discussed in this survey without getting lost in details that are not pertinent.

**A map of the neuro-symbolic AI landscape by architectures.** Our second contribution is a map of the neuro-symbolic AI research area, illustrated in Figure 1, through the lens of the architectures of the different frameworks. At the top level, we distinguish between *composite* and *monolithic* frameworks. While composite frameworks keep the symbolic (i.e. logical) and neural components separate, monolithic frameworks integrate logical reasoning directly into the architecture of neural networks. The two groups are, thus, complement of each other. At lower levels of the hierarchy, we differentiate how symbolic and neural components are connected, as well as the types of neural models and inference properties of the symbolic methods used in the frameworks.

**A handbook for extending existing AI models with neuro-symbolic concepts.** Our third contribution is a guide for researchers on how to augment their existing neural networks with concepts from

neuro-symbolic AI. We therefore focus primarily on composite frameworks (Section 4). As these frameworks tend to be model agnostic, they allow for a simple extension of an existing logical or neural model.

**A comprehensive account of neuro-symbolic concepts.**   While it is not in the scope of this survey to discuss every paper, we aim to cover the area more broadly to position regularisation techniques in relation to other approaches. To this end, Section 5 discusses the complement of composite frameworks, i.e. monolithic frameworks. We keep the discussion of such frameworks brief and refer the reader to (d'Avila Garcez et al., 2019) for more details.

**A discussion on the desiderata and actual achievements of neuro-symbolic AI.**   Our final contribution is a qualitative perspective on how the different architectures achieve the expectations for neuro-symbolic AI, namely support for complex reasoning, knowledge integration, explainability, reduction of data need, and guarantee satisfaction. To this end, we provide a short discussion at the end of each main section comparing different frameworks that fit within each category and end our survey, in Section 7, with a discussion on the achievements in neuro-symbolic AI to date and an outlook on the main open problems.

## 3   Preliminaries

This section briefly covers probabilistic models (Section 3.1), propositional and first-order logic (Section 3.2), SRL (Section 3.3), and neural networks (Section 3.4). Since the field is developing, there are proposals where some of the structures are adapted to fit for purpose, and covering all variations is beyond the scope of this survey. We aim to cover the essentials which should be sufficient for someone to get started in the field of neuro-symbolic AI.

**Remark 1** (Notation).  *We introduce a general notation to illustrate commonalities across different concepts. In some cases, consistency throughout the survey is favoured over consistency with the literature. For example, a logic program is, typically, denoted $\mathcal{P}(R, F)$, with $R$ the set of logic rules and $F$ the set of facts, whereas we denote it by $\mathcal{P}(\boldsymbol{\rho}, \overline{\boldsymbol{\alpha}})$. Table 2 summarises the notations used in this survey.*

| Entity | Notation | Example |
|---|---|---|
| Random Variables | uppercase | $X, Y, Z$ |
| Parameterised RVs | calligraphic uppercase | $\mathcal{X}, \mathcal{Y}, \mathcal{Z}$ |
| Logical Variables | uppercase typewriter | $\mathtt{A}, \mathtt{B}, \mathtt{C}$ |
| Logical Constants | lowercase typewriter | $\mathtt{a}, \mathtt{b}, \mathtt{c}, \mathtt{alice}$ |
| Sets of elements | bold | $\boldsymbol{X}, \boldsymbol{\mathcal{X}}, \mathbf{A}$ |
| Instantiation | $\mathcal{I}(\cdot)$ | $\boldsymbol{x} = \mathcal{I}(\boldsymbol{X})$ |
| Set of all possible instantiations | $\boldsymbol{\mathcal{I}}(\cdot)$ | $\boldsymbol{x} \in \boldsymbol{\mathcal{I}}(\boldsymbol{X})$ |
| Probability Distribution | $P$ | $P_{\mathcal{F}}, P_{\mathcal{N}}, P_{\mathcal{L}}$ |
| Partition Function | $Z$ | $Z$ |
| Factor | $f$ | $f_i$ |
| Factor Graph | $\mathcal{F}$ | $\mathcal{F}(\boldsymbol{X}, \boldsymbol{f})$ |
| Feature Function | $F$ | $F_i$ |
| Parameterised Factor | $\phi$ | $\phi_i$ |
| Parameterised Factor Graph | $\Phi$ | $\Phi(\boldsymbol{\mathcal{X}}, \boldsymbol{\phi})$ |
| Predicate | small caps | $\textsc{friends}(\mathtt{a}, \mathtt{b})$ |
| Generic logic components | greek | $\alpha$ (atom), $\overline{\boldsymbol{\alpha}}$ (ground atoms) |
| Logical Formula | greek | $\varphi$ (generic formula), $\rho$ (rule) |
| Abducibles, Outcomes | calligraphic | $\mathcal{A}, \mathcal{O}$ |
| Models | calligraphic | $\mathcal{N}$ (neural), $\mathcal{L}$ (logical), $\mathcal{P}$ (program) |

Table 2:  Notation used in this survey.

### 3.1 Probabilistic Graphical Models

***Probabilistic graphical models*** are probabilistic models, where a graph expresses the conditional dependencies between random variables. We start, in Section 3.1.1, by introducing ***factor graphs*** (Pearl, 1988) – a general undirected graphical model, which makes the factorisation of probability distributions explicit. Then, in Section 3.1.2, we introduce ***parameterised factor graphs*** – a model that allows us to consider factor graphs at a higher level of abstraction, offering a more succinct representation.

#### 3.1.1 Factor Graphs

**TL;DR** (Factor Graph). *Factor graphs (Pearl, 1988) are graphical models that make the factorisation of a function explicit, e.g. a factorisation of a joint probability distribution. Both directed (e.g. Bayesian networks (Pearl, 1988)) and undirected probabilistic models (e.g. Markov random fields (Kindermann & Snell, 1980)) can be mapped to equivalent factor graphs, allowing us to generalise discussions in the remainder of the survey.*

Consider a factorisable function $g(\boldsymbol{x}) = \prod_{i=1}^{M} f_i(\boldsymbol{x}_i)$, with $\boldsymbol{x}$ a set of variables, and $\boldsymbol{x}_i \subseteq \boldsymbol{x}$. A ***factor graph*** $\mathcal{F}(\boldsymbol{x}, \boldsymbol{f})$ is an undirected bipartite graph representing $g(\boldsymbol{x})$, where the two sets of nodes are the ***factors*** $\boldsymbol{f} = \{f_i\}_{i=1}^{M}$ and the variables $\boldsymbol{x} = \{x_i\}_{i=1}^{N}$, and there is an edge between each $x_i \in \boldsymbol{x}$ and $f_j \in \boldsymbol{f}$, if and only if $x_i \in \boldsymbol{x}_j$ of $f_j(\boldsymbol{x}_j)$.

Let us fix a set of random variables (RVs) $\boldsymbol{X} = \{X_i\}_{i=1}^{N}$, each with its own domain. An ***instantiation*** $\mathcal{I}(\boldsymbol{X})$, maps the RVs to values from the domain of the corresponding RV. We use $\boldsymbol{\mathcal{I}}(\boldsymbol{X})$ to denote the set of all sets that can be obtained by instantiating the random variables in $\boldsymbol{X}$ in all possible ways. A factor graph $\mathcal{F}(\boldsymbol{X}, \boldsymbol{f})$, where all factors map to non-negative real numbers, also known as ***potentials***, then defines a joint probability as

$$P_{\mathcal{F}}(\boldsymbol{X} = \boldsymbol{x}) := \frac{1}{Z} \prod_{i=1}^{M} f_i(\boldsymbol{x}_i) \,, \tag{1}$$

where, assuming $\boldsymbol{x} = \mathcal{I}(\boldsymbol{X})$, for some instantiation $\mathcal{I}$, $\boldsymbol{x}_i = \mathcal{I}(\boldsymbol{X}_i)$, with $i$ ranging over factors. $Z$ is the ***partition function***, i.e. a normalising constant, given by

$$Z = \sum_{\boldsymbol{x} \in \boldsymbol{\mathcal{I}}(\boldsymbol{X})} \prod_{i=1}^{M} f_i(\boldsymbol{x}_i) \,. \tag{2}$$

**Log-linear models.** Factor graphs can always be represented in a ***log-linear model***, by replacing each factor by an exponentiated weighted ***feature function*** $F_i$ of the state as

$$P_{\mathcal{F}}(\boldsymbol{X} = \boldsymbol{x}) = \frac{1}{Z} \exp\left(\sum_{i=1}^{M} w_i F_i(\boldsymbol{x}_i)\right) \,, \tag{3}$$

where $w_i$ is its corresponding coefficient. A feature function can be any real-valued function evaluating the ***state*** of (part of) the system. A state is a specific instantiation $\mathcal{I}$ of the variables. In this survey, we consider binary features, i.e. $F_i(\boldsymbol{x}_i) \in \{0, 1\}$, and features mapping to the unit interval, i.e. $F_i(\boldsymbol{x}_i) \in [0, 1]$.

**Example 1** (Factor Graph). Let us consider a set of seven Boolean RVs $\{X_1, \ldots, X_7\}$, i.e. $X_i \in \{0, 1\}$, and three factors $\{f_1, f_2, f_3\}$:

$$f_1(X_1, X_2, X_3, X_4) = \begin{cases} 1 & \text{if } X_1 = X_2 = X_3 = 1 \text{ and } X_4 = 0 \\ 2 & \text{otherwise} \end{cases}$$

$$f_2(X_3, X_5, X_6) \quad = \begin{cases} 1 & \text{if } X_3 = X_5 = 1 \text{ and } X_6 = 0 \\ 4 & \text{otherwise} \end{cases}$$

$$f_3(X_4, X_6, X_7) \quad = \begin{cases} 1 & \text{if } X_4 = X_6 = 1 \text{ and } X_7 = 0 \\ 3 & \text{otherwise.} \end{cases}$$

These factors can be visualised by a factor graph (Figure 2), or written in log-linear form by assigning weights $w_1 = \ln(2)$, $w_2 = \ln(4)$, $w_3 = \ln(3)$ and feature functions:

$$F_1(X_1, X_2, X_3, X_4) = \begin{cases} 0 & \text{if } X_1 = X_2 = X_3 = 1 \text{ and } X_4 = 0 \\ 1 & \text{otherwise} \end{cases}$$

$$F_2(X_3, X_5, X_6) = \begin{cases} 0 & \text{if } X_3 = X_5 = 1 \text{ and } X_6 = 0 \\ 1 & \text{otherwise} \end{cases}$$

$$F_3(X_4, X_6, X_7) = \begin{cases} 0 & \text{if } X_4 = X_6 = 1 \text{ and } X_7 = 0 \\ 1 & \text{otherwise.} \end{cases}$$

Note that using these features in the log-linear model of Equation (3) is equivalent to using the original factors in Equation (1), e.g. $\exp(1 \cdot w_1) = \exp(\ln(2)) = 2$ and $\exp(0 \cdot w_1) = 1$.

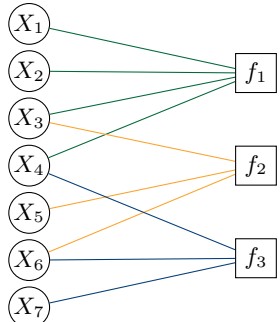

Figure 2: The factor graph representing the factors in Example 1.

**Marginal distributions.** Computing probabilities for assignments to a subset $\boldsymbol{X}' \subseteq \boldsymbol{X}$, from the full joint distribution, is known as computing ***marginals***. Let $\boldsymbol{X}'^c$ denote the complement of $\boldsymbol{X}'$ in $\boldsymbol{X}$, then

$$P(\boldsymbol{X}' = \boldsymbol{x}') = \sum_{\boldsymbol{x}'^c \in \mathcal{I}(\boldsymbol{X}'^c)} P(\boldsymbol{X} = \boldsymbol{x}) \ . \tag{4}$$

Computing marginals is generally intractable. Consider for example a distribution over just 100 Boolean variables, then, to compute marginals one would need to compute $2^{100}$ states. However, note that the factorisation in Equation (1) implies that an RV only depends on the factors it is connected to. One option to compute marginals efficiently in factor graphs is ***belief propagation*** (Koller & Friedman, 2009). In this algorithm, messages, which encode the node's belief about its possible values, are passed between *connected* nodes. The marginalisation then reduces to a sum of products of simpler terms compared to the full joint distribution (which is why the algorithm is also referred to as ***sum-product message passing***), thereby reducing the computational complexity.

**Conditional probabilities.** Computing marginals allows us to compute ***conditional probabilities***. Let $\boldsymbol{X}_o \subseteq \boldsymbol{X}$ denote the subset of ***observed*** RVs, i.e. RVs with known values, and let $\boldsymbol{X}_u \subseteq \boldsymbol{X}$ denote the subset of ***unobserved*** RVs, i.e. RVs with unknown values. For a subset of unobserved variables $\boldsymbol{X}'_u \subseteq \boldsymbol{X}_u$, the conditional probability of $\boldsymbol{X}'_u = \boldsymbol{x}'_u$ given $\boldsymbol{X}_o = \boldsymbol{x}_o$ as ***evidence*** is given by

$$P(\boldsymbol{X}'_u = \boldsymbol{x}'_u \mid \boldsymbol{X}_o = \boldsymbol{x}_o) = \frac{P(\boldsymbol{X}'_u = \boldsymbol{x}'_u, \boldsymbol{X}_o = \boldsymbol{x}_o)}{P(\boldsymbol{X}_o = \boldsymbol{x}_o)} \ . \tag{5}$$

Computing conditional probabilities, typically, also relies on sum-product message passing.

**Maximum a posteriori state.** Computing conditional probabilities, in turn, allows us to compute the *maximum a posteriori state* (MAP). The goal of MAP is to compute the most likely joint assignment to $\boldsymbol{X}'_u$, given an assignment $\boldsymbol{x}_o$ to the variables $\boldsymbol{X}_o$:

$$\text{MAP}(\boldsymbol{X}'_u = \boldsymbol{x}'_u \mid \boldsymbol{X}_o = \boldsymbol{x}_o) = \arg\max_{\boldsymbol{x}'_u} P(\boldsymbol{X}'_u = \boldsymbol{x}'_u \mid \boldsymbol{X}_o = \boldsymbol{x}_o)$$

The MAP can be computed using *max-product message passing*, where instead of summing messages the maximum is chosen. The operation $\text{MAP}(\boldsymbol{X}_u = \boldsymbol{x}_u \mid \boldsymbol{X}_o = \boldsymbol{x}_o)$, i.e. predicting all unobserved RVs, is called the *most probable explanation* (MPE).

**Example 2** (Probability computation in a factor graph)**.** Building on Example 1, assume $X_4 = X_6 = 1$ is given as evidence, and the goal is to compute $P(X_7 = 1)$. The naïve approach would be to calculate all marginals, which would require computing the probability of $2^7 = 128$ states, as we have seven binary variables. However, from Figure 2, we find that, given $X_4$ and $X_6$ as evidence, $X_7$ is independent of $X_1$, $X_2$, $X_3$, and $X_5$.

$$
\begin{aligned}
P(X_7 = 1 | X_4 = 1, X_6 = 1) &= \frac{P(X_7 = 1, X_4 = 1, X_6 = 1)}{P(X_4 = 1, X_6 = 1)} \\
&= \frac{P(X_7 = 1, X_4 = 1, X_6 = 1)}{P(X_7 = 1, X_4 = 1, X_6 = 1) + P(X_7 = 0, X_4 = 1, X_6 = 1)} \\
&= \frac{f_3(X_7 = 1, X_4 = 1, X_6 = 1)}{f_3(X_7 = 1, X_4 = 1, X_6 = 1) + f_3(X_7 = 0, X_4 = 1, X_6 = 1)} \\
&= \frac{3}{4} \,,
\end{aligned}
$$

where, first, we used Equation (5), second, we used Equation (4) and third, we cancelled contributions from $f_1$, $f_2$, and the partition function $Z$.

**Markov random fields (MRFs) (Kindermann & Snell, 1980).** Markov random fields or *Markov networks* are undirected probabilistic graphical models, where the nodes of the graph represent RVs and the edges describe *Markov properties*. The *local Markov property* states that any RV is conditionally independent of all other RVs given its neighbours. There are three Markov properties (pairwise, local, and global). However, for positive distributions (i.e. distributions with non-zero probabilities for all variables) the three are equivalent (Koller & Friedman, 2009). Each maximal clique in the graph is associated with a potential (in contrast to factor graphs, the potentials are not explicit in the graph), and the MRF then defines a probability distribution equivalently to Equation (1). An MRF can be converted to a factor graph by creating a factor node for each maximal clique and connecting it to each RV from that clique. Figure 3 shows the MRF on the left for the equivalent factor graph on the right from Example 1.

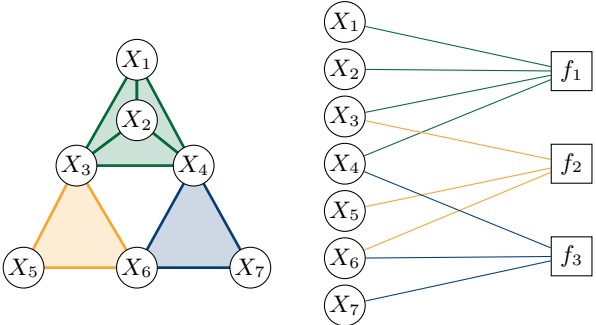

Figure 3: The MRF (left) and the equivalent factor graph (right) for Example 1.

**Bayesian networks (BNs) (Pearl, 1988).** Bayesian Networks or *belief networks* are directed probabilistic graphical models where each node corresponds to a random variable, and each edge represents the

conditional probability for the corresponding random variables. The probability distribution expressed by the BN is defined by providing the conditional probabilities for each node given its parent nodes' states. A BN can be converted to a factor graph by introducing a factor $f_i$ for each RV $X_i$ and connecting the factor with the parent nodes and itself. $f_i$ represents the conditional probability distribution of $X_i$ given its parents. Figure 4 illustrates an example of a BN and an equivalent factor graph.

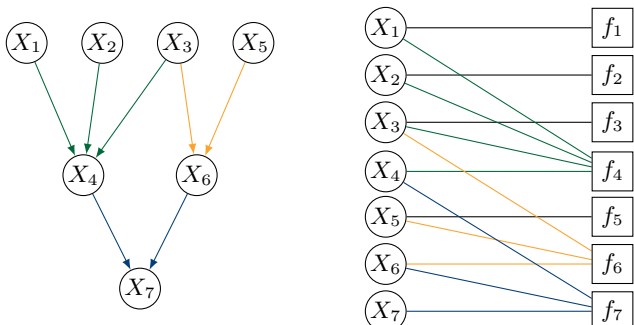

Figure 4: Example of a Bayesian network (left) and an equivalent factor graph (right).

### 3.1.2 Parameterised Factor Graphs

**TL;DR** (Parameterised factor graphs). *Parameterised factor graphs (Poole, 2003) act as templates to instantiate symmetric factor graphs. Parameterised factors provide a relational language to succinctly represent sets of factors sharing the same potential function and the same structure but differing in their set of RVs.*

A **parameterised factor graph** (par-factor) $\Phi(\boldsymbol{\mathcal{X}}, \boldsymbol{\phi})$, consist of a set of **parameterised RVs** (par-RVs) $\boldsymbol{\mathcal{X}} = \{\mathcal{X}_i\}_{i=1}^N$ and a set of **parameterised-factors** (par-factors) $\boldsymbol{\phi} = \{\phi_i\}_{i=1}^M$. A par-factor $\phi_i$ is a function (for a subset $\boldsymbol{\mathcal{X}}_i \subseteq \boldsymbol{\mathcal{X}}$) from $\mathcal{I}(\mathcal{I}_{\mathcal{C}}(\boldsymbol{\mathcal{X}}_i))$ to non-negative real numbers. Here, the first instantiation maps from par-RVs to RVs, i.e. $\boldsymbol{X}' = \mathcal{I}_{\mathcal{C}}(\boldsymbol{\mathcal{X}}_i)$, and the second instantiation maps from RVs to their values, i.e. $\boldsymbol{x}' = \mathcal{I}(\mathcal{I}_{\mathcal{C}}(\boldsymbol{\mathcal{X}}_i))$, where $\boldsymbol{X}'$ is one possible instantiation of $\boldsymbol{\mathcal{X}}_i$. Thus, par-RVs help us to abstract RVs. How RVs can be instantiated from par-RVs is defined by a **constraints set** $\mathcal{C}$.

Just as factor graphs, par-factor graphs define probability distributions. Let $\boldsymbol{X}$ be the set composed of all RVs that can instantiate the par-RVs in $\boldsymbol{\mathcal{X}}$. The probability distribution defined by $\Phi(\boldsymbol{\mathcal{X}}, \boldsymbol{\phi})$ is given by

$$P_{\Phi}(\boldsymbol{X} = \boldsymbol{x}) := \frac{1}{Z} \prod_{i=1}^M \prod_{\boldsymbol{X}_j \in \mathcal{I}_{\mathcal{C}}(\boldsymbol{\mathcal{X}}_i)} \phi_i(\boldsymbol{x}_j) \,, \tag{6}$$

where $\boldsymbol{X}_j \subseteq \boldsymbol{X}$ are the different sets that can be instantiated from the par-RVs $\boldsymbol{\mathcal{X}}_i$ participating in the par-factor $\phi_i$, and $\boldsymbol{x}_j \subseteq \boldsymbol{x} = \mathcal{I}(\boldsymbol{X})$, such that $\boldsymbol{x}_j = \mathcal{I}(\boldsymbol{X}_j)$.

An important goal of parameterised factor graphs is to serve as a syntactic convenience, providing a much more compact representation of relationships between objects in probabilistic domains. The same models can be represented just as well as propositional or standard graphical models. However, as we will see in Section 3.3.1, the more succinct representation can also lead to more efficient inference.

**Example 3** (Par-factor graph). Let us consider a par-factor graph $\Phi$ with just one par-factor $\phi_1(\mathcal{X}_1, \mathcal{X}_2, \mathcal{Y})$, where the par-RVs can be instantiated as follows $\mathcal{I}_{\mathcal{C}}(\mathcal{X}_1) \in \{X_1, X_2\}$, $\mathcal{I}_{\mathcal{C}}(\mathcal{X}_2) \in \{X_1, X_2\}$, and $\mathcal{I}_{\mathcal{C}}(\mathcal{Y}) \in \{Y_{11}, Y_{12}, Y_{21}, Y_{22}\}$. The constraint set $\mathcal{C}$ is given as: if $\mathcal{I}_{\mathcal{C}}(\mathcal{X}_1) = X_i$ and $\mathcal{I}_{\mathcal{C}}(\mathcal{X}_2) = X_j$, then $\mathcal{I}_{\mathcal{C}}(\mathcal{Y}) = Y_{ij}$. Then, we can instantiate the following factors $f_{11}(X_1, X_1, Y_{11})$, $f_{12}(X_1, X_2, Y_{12})$, $f_{21}(X_2, X_1, Y_{21})$ and $f_{22}(X_2, X_2, Y_{22})$. Figure 5 shows the par-factor graph $\Phi$ on the left and the instantiated factor graph $\mathcal{F}$ on the right.

### 3.2 Logic

**TL;DR** (Logic). *Logic allows us to reason about connections between objects and express rules to model worlds. The goal of logic programming is threefold:*

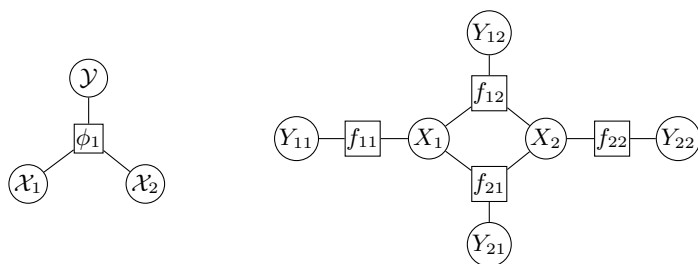

Figure 5: The par-factor graph (left) and instantiated factor graph (right) of Example 3.

- *State what is true: Alice likes Star Wars.*

- *Check whether something is true: Does Alice like Star Wars?*

- *Check what is true: Who likes Star Wars?*

This section begins by introducing propositional logic (Section 3.2.1) and first-order logic (Section 3.2.2), and finishes with a brief introduction to logic programming (Section 3.2.3) – a programming paradigm which allows us to reason about a database using logical theories. We present the syntaxes of the different languages and their operations, for the semantics, the reader is referred to Bradley & Manna (2007).

### 3.2.1 Propositional Logic

The language of ***propositional logic*** (Bradley & Manna, 2007) consists of Boolean variables and logical connectives. An ***atom*** in propositional logic is a logical variable. A ***literal*** is either an atom X or its negation ¬X. ***Formulae*** in propositional logic are expressions formed over literals and the logical connectives ¬ (negation), ∧ (conjunction), ∨ (disjunction) and → (implication). Let $\varphi$ be a propositional formula. A formula $\varphi_1 \wedge \varphi_2$ is called a ***conjunction***, with $\varphi_1$ and $\varphi_2$ its ***conjuncts***. A conjunction is `True` if both conjuncts are `True`. A formula $\varphi_1 \vee \varphi_2$ is called a ***disjunction***, with $\varphi_1$ and $\varphi_2$ its ***disjuncts***. A disjunction is `True` if either of the disjuncts is `True`. A formula $\varphi_1 \rightarrow \varphi_2$ is called a (material) ***implication***, $\varphi_1 \rightarrow \varphi_2 \equiv \neg\varphi_1 \vee \varphi_2$. A ***clause*** is a disjunction of one or more literals. A ***theory $\varphi$*** is a set of ***sentences***, with a sentence being a formula in which each variable is quantified. Theories are interpreted as a conjunction of their sentences. A theory is in ***conjunctive normal form*** (CNF) if it is a conjunction of clauses. An ***interpretation*** of a propositional formula $\varphi$ is an instantiation of each of its variables to either `True` or `False`, which we, therefore, also denote by $\mathcal{I}$. A ***model $\mathcal{M}$*** of $\varphi$ is an interpretation that makes $\varphi$ evaluate to `True`, which we denote by $\mathcal{M} \models \varphi$. While checking whether an assignment to logical variables satisfies a logical theory can be done in polynomial time, finding a solution for a theory is the original NP-complete problem (Cook, 1971). This problem, known as the ***satisfiability problem***, is often abbreviated as SAT.

**Example 4** (Propositional theory)**.** Recommendation systems are a typical AI application where users are suggested items they might like based on their previous purchases, their social network, and the features of the user as well as the item. Let us assume the following:

1. If Alice is a computer science student ($\mathtt{S}_A$) AND Bob is a computer science student ($\mathtt{S}_B$) AND both take the same class ($\mathtt{C}_{AB}$), THEN they are friends ($\mathtt{F}_{AB}$).

2. If Alice is a computer science student ($\mathtt{S}_A$) AND she likes Star Wars ($\mathtt{L}_{AW}$), THEN she also likes Star Trek ($\mathtt{L}_{AT}$).

3. If Alice and Bob are friends ($\mathtt{F}_{AB}$) AND Alice likes Star Trek ($\mathtt{L}_{AT}$), THEN Bob also likes Star Trek ($\mathtt{L}_{BT}$).

By considering the different sub-statements as propositional logic variables $\{S_A, S_B, C_{AB}, F_{AB}, L_{AW}, L_{AT}, L_{BT}\}$, the above statements can be written as a theory $\boldsymbol{\varphi}$:

$$\varphi_1 := S_A \land S_B \land C_{AB} \to F_{AB}$$
$$\varphi_2 := S_A \land L_{AW} \to L_{AT} \tag{7}$$
$$\varphi_3 := F_{AB} \land L_{AT} \to L_{BT}$$

All three formulae are written as logical implications and form together the theory $\boldsymbol{\varphi}$. $\{S_A = \top, S_B = \top, C_{AB} = \top, F_{AB} = \top, L_{AW} = \top, L_{AT} = \bot, L_{BT} = \top\}$ is an interpretation of the theory, but not a model, since $\{S_A = \top, L_{AW} = \top, L_{AT} = \bot\} \not\models \varphi_2$.

A problem of propositional logic becomes apparent from this example: once we want to generalise these rules to a large set of people, we would need to create a new logical variable for each person. First-order logic allows us to reason at a higher level and abstract the problem away by reasoning about groups rather than individual instances.

### 3.2.2 First-Order Logic

In first-order logic (FOL) (Bradley & Manna, 2007), a **term** is either a **variable** or a **constant**. A **substitution** $\sigma$ is a mapping from variables to constants. An **atom** in FOL is an expression of the form $P(\mathbf{t})$, where $P$ is a relational **predicate** and $\mathbf{t}$ is a vector of terms. An atom $P(\mathbf{t})$ is **ground**, if $\mathbf{t}$ includes only constants. For example, $\alpha := \text{FRIENDS}(U_1, U_2)$ is an atom consisting of the predicate FRIENDS and variables $U_1$ and $U_2$, $\overline{\alpha} := \text{FRIENDS}(\texttt{alice}, \texttt{bob})$ is a ground atom, and, for a substitution $\sigma := \{U_1 \mapsto \texttt{alice}, U_2 \mapsto \texttt{bob}\}$, $\alpha\sigma \equiv \overline{\alpha}$. In classical Boolean logic, each ground atom is mapped to either `True` or `False`. However, other logical formalisms may map ground atoms to the unit interval [0,1]. A **function** in FOL maps constants to constants. For example, the function friendOf could map `alice` to `bob`, i.e. friendOf(`alice`) would evaluate to `bob`.

A **literal** in FOL is an atom $\alpha$ or its negation $\neg\alpha$. **Formulae** in FOL are expressions that, similarly to propositional logic, are formed over literals and the logical connectives $\neg$, $\land$, $\lor$, and $\to$, with the addition of universal $\forall$ and existential $\exists$ quantifiers. A formula is **instantiated** or ground if each atom in the formula is ground. A **Rule** $\rho$ is a universally quantified formula of the form $\alpha_1 \land \cdots \land \alpha_n \to \alpha_{n+1}$, where each term occurring in the atom $\alpha_{n+1}$ also occurs in some atom $\alpha_j \in \{\alpha_j\}_{j=1}^n$. The left-hand side of the implication is referred to as the **premise** and the right-hand side as the **conclusion** of the rule. We denote a theory consisting only of rules by $\boldsymbol{\rho}$.

FOL allows us, thus, to reason about groups rather than individual instances. On one hand, this allows for a more succinct representation but as we will see later it also allows for more efficient computations. The process of going from propositional to first-order logic is often referred to as **lifting**, and models that support first-order logic are said to be lifted.

**Example 5** (First-order logic). Let us consider just the last rule from Example 4 as our new theory $\boldsymbol{\rho}$. Lifting this rule to first-order logic give us

$$\boldsymbol{\rho} := \forall U_1, U_2 \in \texttt{Users}, \forall I \in \texttt{Items} : \text{FRIENDS}(U_1, U_2) \land \text{LIKES}(U_1, I) \to \text{LIKES}(U_2, I) \,,$$

where $\texttt{Users} = \{\texttt{alice}, \texttt{bob}\}$ and $\texttt{Items} = \{\texttt{startrek}\}$ are sets of constants. $\varphi_3$ in (7) is, thus, equivalent to a ground or instantiated case of this rule.

### 3.2.3 Logic Programming

A **logic program** $\mathcal{P}$ is a tuple $(\boldsymbol{\rho}, \overline{\boldsymbol{\alpha}})$, where $\boldsymbol{\rho}$ is a set of rules, and $\overline{\boldsymbol{\alpha}}$ is a set of ground atoms typically called **facts** (Sterling & Shapiro, 1994).

The **Herbrand universe** $\mathcal{H}_U(\mathcal{P})$ is the (possibly infinite) set of all terms that one can construct using all constants and function symbols in the logic program $\mathcal{P}$. In most neuro-symbolic frameworks, however, the program is limited to function-free theories over finite sets of constants. Therefore, the Herbrand universe is simply the set of all constants. For a given Herbrand universe, its **Herbrand base** $\mathcal{H}_B(\mathcal{P})$ is the set of all

possible ground atoms that can be created by instantiating the atoms in the set of rules $\boldsymbol{\rho}$ with the terms from $\mathcal{H}_U(\mathcal{P})$.

**Example 6** (Herbrand Base)**.** The Herbrand universe of a logic program $\mathcal{P}(\boldsymbol{\rho}, \overline{\boldsymbol{\alpha}})$ with $\boldsymbol{\rho}$ from Example 5 is $\mathcal{H}_U = \{\texttt{alice}, \texttt{bob}, \texttt{startrek}\}$, and the Herbrand base is

$$\mathcal{H}_B(\mathcal{P}) = \{\text{FRIENDS}(\texttt{alice}, \texttt{alice}), \text{FRIENDS}(\texttt{alice}, \texttt{bob}),$$
$$\text{FRIENDS}(\texttt{bob}, \texttt{bob}), \text{FRIENDS}(\texttt{bob}, \texttt{alice}),$$
$$\text{LIKES}(\texttt{alice}, \texttt{startrek}), \text{LIKES}(\texttt{bob}, \texttt{startrek})\} \, .$$

The Herbrand base in logic programming consists of two sets: the ***abducibles*** $\mathcal{A}$, with $\overline{\boldsymbol{\alpha}} \subseteq \mathcal{A}$, and the ***outcomes*** $\mathcal{O}$. The two sets are disjoint, i.e. $\mathcal{A} \cap \mathcal{O} = \emptyset$. In logic programming, we operate under the ***closed world asummption***, i.e. any ground atom in the abducibles that is not in the input facts is assumed to be false: $\forall \overline{\alpha} \in \mathcal{A} \setminus \overline{\boldsymbol{\alpha}} : \overline{\alpha}$ is `False`. Further, all groundings of atoms in the conclusion of the rules are in $\mathcal{O}$.

The ***Herbrand instantiation*** $\mathcal{H}_\mathcal{I}(\mathcal{P})$ is the set of all ground rules obtained after replacing the variables in each rule in $\boldsymbol{\rho}$ with terms from its Herbrand universe in every possible way. A ***Herbrand interpretation*** or ***possible world*** $\mathcal{I}$, is a mapping of the ground atoms in the Herbrand base to truth values. For brevity, we will use the notation $\overline{\alpha} \in \mathcal{I}$ (resp. $\overline{\alpha} \notin \mathcal{I}$), when a ground atom $\overline{\alpha}$ is mapped to `True` (resp. `False`) in $\mathcal{I}$. A ***partial interpretation*** is a truth assignment to a subset of atoms of the Herbrand base. A Herbrand interpretation $\mathcal{I}$ is a ***model*** $\mathcal{M}$ of $\mathcal{P}$ if all rules in $\boldsymbol{\rho}$ are satisfied. A model $\mathcal{M}$ of $\mathcal{P}$ is ***minimal*** if no other model $\mathcal{M}'$ of $\mathcal{P}$ has fewer atoms mapped to `True` than $\mathcal{M}$. Such a model is called the ***least Herbrand model***. Each logic program has a unique least Herbrand model, which is computed via the consequence operator.

**Definition 1** (Consequence operator)**.** *For a logic program $\mathcal{P}$ and a partial interpretation $\mathcal{I}$ of $\mathcal{P}$, the consequence operator is defined by*

$$T_\mathcal{P}(\mathcal{I}) := \{\overline{\alpha} \mid \overline{\alpha} \in \mathcal{I} \; or \; \exists (\overline{\alpha} \leftarrow \bigwedge_{\overline{\alpha}_\mathsf{p} \in \overline{\boldsymbol{\alpha}}_\mathsf{p}} \overline{\alpha}_\mathsf{p}) \in \mathcal{H}_\mathcal{I}(\mathcal{P}), \; s.t. \; \forall \overline{\alpha}_\mathsf{p} \in \overline{\boldsymbol{\alpha}}_\mathsf{p}, \overline{\alpha}_\mathsf{p} \in \mathcal{I} \; is \; \texttt{True}\}$$

*where $\overline{\boldsymbol{\alpha}}_\mathsf{p}$ denotes the ground atoms in the premise of a rule in the Herbrand instantiation.*

Consider the sequence $\mathcal{I}_1 = T_\mathcal{P}(\emptyset)$, $\mathcal{I}_2 = T_\mathcal{P}(\mathcal{I}_1), \ldots$ Let $n$ be the smallest positive integer such that $\mathcal{I}_n = T_\mathcal{P}(\mathcal{I}_n)$, then $\mathcal{I}_n$ is the least Herbrand model of $\mathcal{P}$.

**Entailment.** A program $\mathcal{P}$ ***entails*** a ground atom $\overline{\alpha}$ – denoted as $\mathcal{P} \models \overline{\alpha}$ or $\boldsymbol{\rho} \cup \overline{\boldsymbol{\alpha}} \models \overline{\alpha}$ – if $\overline{\alpha}$ is `True` in every model of $\mathcal{P}$. Note that if $\overline{\alpha}$ is `True` in every model of $\mathcal{P}$, then it is `True` in the least Herbrand model of $\mathcal{P}$.

**Queries.** A ***query*** $Q$ is an expression of the form $\text{P}(\mathbf{t})$, where $\text{P}$ is a predicate, and $\mathbf{t}$ is a tuple of terms. If $\mathbf{t}$ is a tuple of constants, then $Q$ is Boolean. The answer to a Boolean query $Q$ is `True` when $\mathcal{P} \models Q$ and false otherwise. If $Q$ is non-Boolean, then a substitution $\sigma$ is an ***answer*** to $Q$ on $\mathcal{P}$, if $\mathcal{P} \models Q\sigma$. The task of finding all answers to $Q$ on $\mathcal{P}$ is called ***query answering under rules***, or simply ***query answering***. Note that if $Q$ is Boolean, then $\sigma$ is the empty substitution.

**Example 7** (Logic Program)**.** Let us extend Example 5 such that $\boldsymbol{\rho}$ consists of three rules

$$\forall \text{U} \in \textbf{Users}, \forall \text{I}_1, \text{I}_2 \in \textbf{Items} : \text{SIMILAR}(\text{I}_1, \text{I}_2) \wedge \text{LIKES}(\text{U}, \text{I}_1) \rightarrow \text{LIKES}(\text{U}, \text{I}_2) \tag{8}$$

$$\forall \text{U}_1, \text{U}_2 \in \textbf{Users}, \forall \text{I} \in \textbf{Items} : \text{FRIENDS}(\text{U}_1, \text{U}_2) \wedge \text{LIKES}(\text{U}_1, \text{I}) \rightarrow \text{LIKES}(\text{U}_2, \text{I}) \tag{9}$$

$$\forall \text{U} \in \textbf{Users}, \forall \text{I} \in \textbf{Items} : \text{KNOWNLIKES}(\text{U}, \text{I}) \rightarrow \text{LIKES}(\text{U}, \text{I}) \tag{10}$$

with $\textbf{Users} = \{\texttt{alice}, \texttt{bob}\}$, $\textbf{Items} = \{\texttt{starwars}, \texttt{startrek}\}$, and a set of input facts

$$\overline{\boldsymbol{\alpha}} = \{\text{FRIENDS}(\texttt{alice}, \texttt{alice}), \text{FRIENDS}(\texttt{alice}, \texttt{bob}),$$
$$\text{FRIENDS}(\texttt{bob}, \texttt{bob}), \text{FRIENDS}(\texttt{bob}, \texttt{alice}),$$
$$\text{KNOWNLIKES}(\texttt{alice}, \texttt{starwars}), \text{SIMILAR}(\texttt{starwars}, \texttt{startrek})\} \, .$$

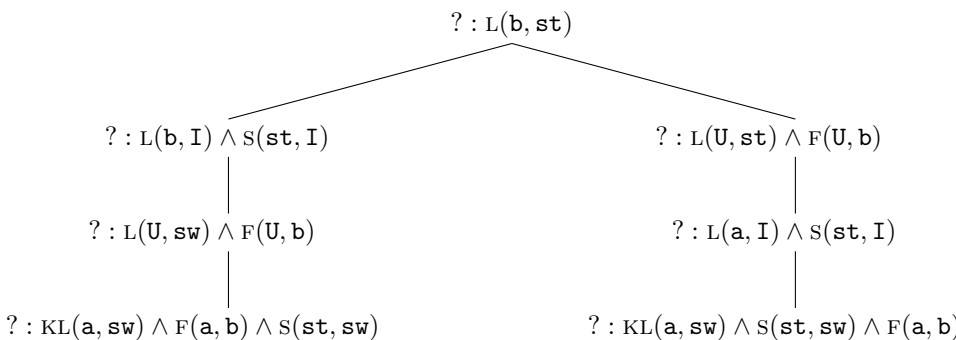

Figure 6: A proof tree for Example 7. We abbreviate LIKES by L, FRIENDS by F, SIMILAR by S, KNOWNLIKES by KL, bob by b, alice by a, starwars by sw, and startrek by st.

Rule (10) is a solution to enable partial knowledge of item preferences. It states that for the cases where we know that a user likes an item, we can assign LIKES(U, I) to True. This is necessary since the predicates in the heads of the rules need to be distinct from the predicates in the input facts.

We can check whether program $\mathcal{P}$ entails LIKES(bob, startrek) by building a proof using **backward chaining** as shown in Figure 6. On the top level of the proof tree, we check whether LIKES(bob, startrek) $\in \overline{\alpha}$. Since that is not the case, we examine at the second level two ways of proving LIKES(bob, startrek): either bob likes a movie similar to startrek or he is friends with someone who likes startrek. Neither case is to be found directly in $\overline{\alpha}$. However, the SIMILAR and FRIENDS atoms are part of the abducibles, and specifically, we find SIMILAR(starwars, startrek) and FRIENDS(bob, alice) in the input facts. Since SIMILAR(starwars, startrek) is in the input, at the third level on the left branch, we check whether there is a friend of bob who likes starwars; and at the third level on the right branch, we check whether alice likes a movie similar to startrek, since we know that FRIENDS(bob, alice). The program then finds on the last level of the proof that since we have KNOWNLIKES(alice, starwars) that LIKES(alice, starwars) is True, and since we known that SIMILAR(starwars, startrek), alice also likes startrek, and since alice is friends with bob he also likes startrek. Note that both proofs are the same. Similarly, we can perform a non-boolean query for all possible assignments to LIKES(U, startrek), which from the above proof would return LIKES(alice, startrek) and LIKES(bob, startrek).

Alternatively, one can reason about this query via **forward chaining** by applying the consequence operator from Definition 1. After the first iteration,

$$T_{\mathcal{P}}(\overline{\alpha}) = \overline{\alpha} \cup \{\text{LIKES}(\text{alice}, \text{starwars})\},$$

where the new fact is derived from (10). Then, in the second step, we can derive

$$T_{\mathcal{P}}^2(\overline{\alpha}) = T_{\mathcal{P}}(\overline{\alpha}) \cup \{\text{LIKES}(\text{alice}, \text{startrek}), \text{LIKES}(\text{bob}, \text{starwars})\},$$

where the first new fact is derived from (8), and the second from (9). Finally,

$$T_{\mathcal{P}}^3(\overline{\alpha}) = T_{\mathcal{P}}^2(\overline{\alpha}) \cup \{\text{LIKES}(\text{bob}, \text{startrek})\},$$

where the new fact is derived from (9). Since LIKES(bob, startrek) $\in T_{\mathcal{P}}^3(\overline{\alpha})$, we can stop the computation here and return an affirmative answer to the query.

**Logical abduction (Kakas, 2017).** In the logic programming community, **abduction** for a query $Q \subset \mathcal{O}$ consists in finding all possible sets of input facts $\overline{\alpha} \subseteq \mathcal{A}$, such that $\mathcal{P}(\rho, \overline{\alpha}) \models Q$. Abduction then returns a formula $\varphi_{\mathcal{A}}$, which is a disjunction where each disjunct is a conjunction of the ground atoms in $\overline{\alpha} \subseteq \mathcal{A}$ such that $\rho \cup \overline{\alpha} \models Q$, i.e.

$$\varphi_{\mathcal{A}}^Q := \bigvee_{\substack{\overline{\alpha}_i \subseteq \mathcal{A} \\ \text{s.t. } \overline{\alpha}_i \cup \rho \models Q}} \left( \bigwedge_{\overline{\alpha}_j \in \overline{\alpha}_i} \overline{\alpha}_j \right). \tag{11}$$

The disjuncts, i.e. $\bigwedge_{\overline{\alpha}_j \in \overline{\boldsymbol{\alpha}}_i} \overline{\alpha}_j$, are called **abductive proofs**, and we denote the process of finding all abductive proofs by $\mathtt{abduce}(\boldsymbol{\rho}, \mathcal{A}, Q)$. The leaves of Figure 6 are two abductive proofs of $\mathtt{abduce}(\boldsymbol{\rho}, \mathcal{A}, \text{LIKES}(\mathtt{bob}, \mathtt{startrek}))$.

### 3.3 Statistical Relational Learning

A limitation of classical Boolean logic is that atoms are either mapped to `True` or `False`. However, the real world is often uncertain and imprecise. Therefore, there is a need to associate facts and logical formulae with some notion of uncertainty that quantifies the extent to which facts are true and the confidence in the formulae.

**TL;DR** (Statistical relational learning). *Statistical relational learning (SRL) (Getoor & Taskar, 2007) is the area of research that concerns itself with the unification of logic and probability to allow for logical reasoning under uncertainty both in the data, and the theory.*

This section describes different concepts from SRL, namely, **lifted graphical models** (Section 3.3.1), **weighted model counting** (Section 3.3.3), and **probabilistic logic programs** (Section 3.3.4). As we will see in Section 4, these frameworks often form the logical components in neuro-symbolic architectures.

#### 3.3.1 Lifted Graphical Models

**TL;DR** (Lifted graphical models). *Lifted graphical models (LGMs) are SRL frameworks combining probabilistic graphical models with first-order logic. Par-factor graphs (Section 3.1.2) lend themselves particularly well for this purpose, where the feature functions compute the extent to which the formulae are satisfied.*

This section outlines the general steps taken to construct and use LGMs. We defer details, such as what it means for a formula to be satisfied, to the next section as those depend on the specific LGM. We denote an LGM by $\mathcal{L}(\boldsymbol{\varphi}, \boldsymbol{w}_{\mathcal{L}})$, where $\boldsymbol{\varphi}$ is a set of formulae $\varphi_i$ with confidence value $w_i \in \boldsymbol{w}_{\mathcal{L}}$. For a logical theory $\boldsymbol{\varphi}$, an LGM can be constructed as follows:

1. Assign a confidence value $w_i \in \mathbb{R}$ to each formula $\varphi_i \in \boldsymbol{\varphi}$ to soften the constraints of logical formulae. This confidence value allows the model to have assignments to atoms that contradict the formula. When $w_i \to \infty$, we obtain hard rules.

2. Treat each unground atom $\alpha_i$ in $\boldsymbol{\varphi}$ as a par-RV $\mathcal{X}_i$ and each possible grounding $\overline{\alpha}_{ij}$ as an RV $X_{ij}$, where the domains of the variables and atoms are the same.

3. As atoms in a formula are dependent on each other, assign each formula $\varphi_i$ a par-factor $\phi_i = \exp(w_i F_i(\boldsymbol{x}_i))$, where each feature function $F_i$ computes the satisfaction of a grounding $\overline{\varphi}_i$ of $\varphi_i$ given the truth assignments $\boldsymbol{x}_i$ to the ground atoms $\overline{\boldsymbol{\alpha}}_i$ in the formula. $F_i$ returns a value in $[0, 1]$ representing how much the formula $\overline{\varphi}_i$ is satisfied.

Given the probability density function of par-factor graphs in Equation (6), and knowing how to represent a set of FOL formulae as par-factors in a log-linear model, we can now define the probability distribution of an LGM as

$$P_{\mathcal{L}}(\boldsymbol{X} = \boldsymbol{x}) := \frac{1}{Z} \exp\left( \sum_{i=1}^{M} \sum_{\boldsymbol{X}_j \in \mathcal{I}(\mathcal{X}_i)} w_i F_i(\boldsymbol{x}_j) \right), \tag{12}$$

where $\boldsymbol{X}_j \subseteq \boldsymbol{X}$ are the different sets that can be instantiated from the par-RVs $\mathcal{X}_i$ participating in the par-factor $\phi_i$, and $\boldsymbol{x}_j \subseteq \boldsymbol{x} = \mathcal{I}(\boldsymbol{X})$, such that $\boldsymbol{x}_j = \mathcal{I}(\boldsymbol{X}_j)$. Here, $\mathcal{X}$ map to atoms, $\boldsymbol{X}$ to ground atoms, and $\boldsymbol{x}$ to truth assignments of the ground atoms. Note that $\boldsymbol{X} = \mathcal{I}_{\mathcal{C}}(\mathcal{X})$, i.e. the atoms of the theory are grounded in every possible way. Thus, the outer sum iterates over the different formulae and the inner sum iterates over the different groundings of each formula. Then, one can compute marginals and conditional probabilities as in Equations (4) and (5).

**Example 8** (Lifted graphical model). Let us construct an LGM for the theory of Example 5 with `Users` = $\{\mathtt{alice}, \mathtt{bob}\}$ and `Items` = $\{\mathtt{startrek}\}$.

1. We assign a confidence value $w_1$ to the formula:

$$w_1 : \forall \mathtt{U}_1, \mathtt{U}_2 \in \mathtt{Users}, \forall \mathtt{I} \in \mathtt{Items} : \textsc{friends}(\mathtt{U}_1, \mathtt{U}_2) \wedge \textsc{likes}(\mathtt{U}_1, \mathtt{I}) \rightarrow \textsc{likes}(\mathtt{U}_2, \mathtt{I}) \qquad (13)$$

2. We map $\textsc{likes}(\mathtt{U}_1, \mathtt{I})$ to $\mathcal{X}_1$, $\textsc{likes}(\mathtt{U}_2, \mathtt{I})$ to $\mathcal{X}_2$, and $\textsc{friends}(\mathtt{U}_1, \mathtt{U}_2)$ to $\mathcal{Y}$. Note that the constraint set $\mathcal{C}$ of the par-factor graph is contained in the constraint set of the formula ($\forall \mathtt{U}_1, \mathtt{U}_2 \in \mathtt{Users}, \forall \mathtt{I} \in \mathtt{Items}$) and implicitly by the structure of the formula, as these constrain how the par-RVs (the unground atoms) can be instantiated.

3. The left-hand side of Figure 7 shows how the par-RVs are connected through par-factors, and the right-hand side shows the instantiated factor graph for the given sets. Note that the same graphs are obtained as in Example 3.

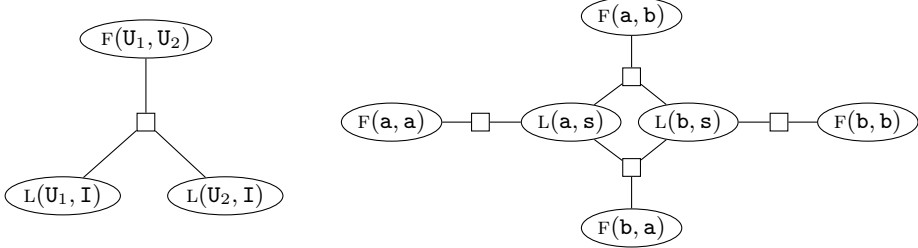

Figure 7: The par-factor graph representing Formula (13) (left) and an instantiated factor graph based on the sets $\mathtt{Users} = \{\mathtt{alice}, \mathtt{bob}\}$ and $\mathtt{Items} = \{\mathtt{starwars}\}$ (right). For readability, we used F for FRIENDS, L for LIKES, a for alice, b for bob, and s for startrek.

**Inference.** Inference consists in computing the MPE for a set of unobserved variables $\boldsymbol{X}_u$, given assignments to observed variables $\boldsymbol{X}_o = \boldsymbol{x}_o$ by means of a conditional distribution:

$$\widehat{\boldsymbol{x}}_u = \arg\max_{\boldsymbol{x}_u} P_{\mathcal{L}}(\boldsymbol{X}_u = \boldsymbol{x}_u | \boldsymbol{X}_o = \boldsymbol{x}_o; \boldsymbol{w}_{\mathcal{L}}) \qquad (14)$$

**Training.** Training is formalised as finding the $\boldsymbol{w}_{\mathcal{L}}$ maximising the log-likelihood of the assignments $\boldsymbol{x} \in \mathcal{D}$, where $\mathcal{D}$ is the training data provided to the LGM:

$$\widehat{\boldsymbol{w}}_{\mathcal{L}}^{t+1} = \arg\max_{\boldsymbol{w}_{\mathcal{L}}} \log P_{\mathcal{L}}(\boldsymbol{X} = \boldsymbol{x}; \boldsymbol{w}_{\mathcal{L}}^t) \qquad (15)$$

Equation (15) works when $\boldsymbol{x}$ provides truth assignments to all variables in $\boldsymbol{X}$. If this assumption is violated, i.e. unobserved variables exist in the training data, training resorts to an expectation-maximisation problem. Parameter learning in LGMs works via gradient ascent (Richardson & Domingos, 2006; Bach et al., 2017).

In most neuro-symbolic frameworks, it is assumed that a logical theory is given and that the only trainable parameters are $\boldsymbol{w}_{\mathcal{L}}$. However, algorithms to learn the theory, known as ***structure learning***, exist. The general approach consists of three steps: i) finding commonly recurrent patterns in the data, ii) extracting formulae from the patterns as potential candidates, iii) reducing the set of candidates to the formulae explaining the data best. The first step helps to reduce the search space, as generally, the number of possible formulae grows exponentially. Khot et al. (2015) use user-defined templates as a starting point to find formulae. However, this approach therefore still requires user input. Kok & Domingos (2010) and Feldstein et al. (2023b) present algorithms based on random walks to find patterns in a hypergraph representation of the relational data. However, the algorithms fail to scale past $\mathcal{O}(10^3)$ relations. Feldstein et al. (2024) present a scalable ($\mathcal{O}(10^6)$) algorithm that avoids expensive inference by estimating the "usefulness" of candidates up-front, but only find rules of a specific form.

### 3.3.2 Examples of Lifted Graphical Models

**Markov logic networks (MLNs) (Richardson & Domingos, 2006).** MLNs are LGMs that consist of a tuple $\mathcal{L}(\boldsymbol{\varphi}; \boldsymbol{w}_{\mathcal{L}})$, where each $\varphi_i \in \boldsymbol{\varphi}$ is a FOL formula, and $w_i \in \mathbb{R}$ is its weight. For a given set of constants, an MLN can be instantiated as a Markov network, similar to how a par-factor graph can be instantiated as a factor graph. Each $(\varphi_i, w_i)$ uniquely determines a par-factor $\phi_i(\boldsymbol{X}_i = \boldsymbol{x}_i) = \exp(w_i F_i(\boldsymbol{x}_i))$, where, as above, $\boldsymbol{X}_i$ are the groundings of $\varphi_i$, and $\boldsymbol{x}_i$ are the assignments of each ground element to either $\mathtt{True}$ or $\mathtt{False}$. Each feature function $F_i$, corresponding to a formula $\varphi_i$, of the MLN evaluates to 1 if $\boldsymbol{x}_i \models \varphi_i$, and 0 otherwise.

**Example 9** (Ground MLN). Consider mapping the atoms of the formulae in Example 4 to RVs as $\{\mathtt{S_B} \mapsto X_1, \mathtt{C_{AB}} \mapsto X_2, \mathtt{S_A} \mapsto X_3, \mathtt{F_{AB}} \mapsto X_4, \mathtt{L_{AW}} \mapsto X_5, \mathtt{L_{AT}} \mapsto X_6, \mathtt{L_{BT}} \mapsto X_7\}$ and assign weights $w_1 = \ln(2)$, $w_2 = \ln(4)$, $w_3 = \ln(3)$ to the respective formulae:

$$
\begin{aligned}
\varphi_1 &:= \mathtt{S_A} \wedge \mathtt{S_B} \wedge \mathtt{C_{AB}} \rightarrow \mathtt{F_{AB}} \\
\varphi_2 &:= \mathtt{S_A} \wedge \mathtt{L_{AW}} \rightarrow \mathtt{L_{AT}} \\
\varphi_3 &:= \mathtt{F_{AB}} \wedge \mathtt{L_{AT}} \rightarrow \mathtt{L_{BT}}
\end{aligned}
\tag{16}
$$

Then, the factors in Example 1 implement the formulae in (16) as a ground MLN, as illustrated in Figure 8. Firstly, the weights of the formulae match the weights of the log-linear model, and secondly, the features in Example 1 evaluate to 1 when the respective formulae in (16) are satisfied. For example, consider the feature $F_3$ that would evaluate $\varphi_3$

$$
F_3(X_4, X_6, X_7) = \begin{cases} 0 & \text{if } X_4 = X_6 = 1 \text{ and } X_7 = 0 \\ 1 & \text{otherwise.} \end{cases}
$$

Under the above mapping $\{\mathtt{F_{AB}} \mapsto X_4, \mathtt{L_{AT}} \mapsto X_6, \mathtt{L_{BT}} \mapsto X_7\}$, and $\{\mathtt{F_{AB}} = \mathtt{True}, \mathtt{L_{AT}} = \mathtt{True}, \mathtt{L_{BT}} = \mathtt{False}\} \not\models \varphi_3$, while all other truth assignments to the logical variables satisfy $\varphi_3$. Note that Figure 8 is equivalent to the MRF of Example 1 illustrated in Figure 3, where the RVs have been replaced by the atoms. In Example 2, we computed that $P(X_7 = 1 \mid X_4 = 1, X_6 = 1) = 0.75$. Thus, we can conclude that, given $\mathtt{F_{AB}} = \mathtt{True}$ and $\mathtt{L_{AT}} = \mathtt{True}$ as evidence, the MLN in this example predicts $P(\mathtt{L_{BT}} = \mathtt{False}) = 0.75$.

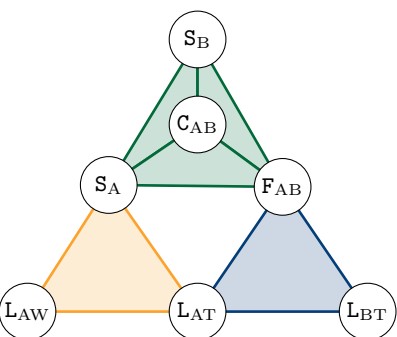

Figure 8: A grounded Markov logic network of the formulae in (16).

**Probabilistic soft logic (PSL) (Bach et al., 2017).** Similarly to MLNs, PSL defines, for a tuple $\mathcal{L}(\boldsymbol{\rho}; \boldsymbol{w}_{\mathcal{L}})$, how to instantiate an MRF from the grounded formulae. However, PSL has four major differences:

1. Formulae in PSL are universally quantified rules $\rho$ (Section 3.2.2).

2. While in MLNs, all ground atoms take on Boolean values ($\mathtt{True}$ or $\mathtt{False}$), in PSL, ground atoms take *soft truth values* from the unit interval $[0, 1]$. Depending on the application, this allows for two interpretations of $\text{LIKES}(\mathtt{alice}, \mathtt{starwars}) = 0.7$ and $\text{LIKES}(\mathtt{alice}, \mathtt{startrek}) = 0.5$: it could either be understood as a stronger confidence in the fact that $\mathtt{alice}$ likes $\mathtt{starwars}$ rather than $\mathtt{startrek}$ or it can be interpreted as $\mathtt{alice}$ liking $\mathtt{starwars}$ more than $\mathtt{startrek}$.

3. PSL uses a different feature function $F$ to compute the rule satisfaction: For a grounding $\boldsymbol{X}_i$ of a rule $\rho_i$ and an assignment $\boldsymbol{x}_i$, the satisfaction of $\rho_i$ computed by $F_i(\boldsymbol{X}_i = \boldsymbol{x}_i)$ in PSL is evaluated using the Łukasiewicz t-(co)norms, defined as follows [2]:

$$
\begin{aligned}
x_i \wedge x_j &= \max\{x_i + x_j - 1, 0\} \\
x_i \vee x_j &= \min\{x_i + x_j, 1\} \\
\neg x &= 1 - x
\end{aligned}
\tag{17}
$$

4. Finally, $\phi_i(\boldsymbol{X}_i = \boldsymbol{x}_i) = \exp(-w_i \cdot (1 - F_i(\boldsymbol{x}_i))^p)$, where $p \in \{1, 2\}$ provides a choice of the type of penalty imposed on violated rules. One can see $1 - F_i(\boldsymbol{x}_i)$ as measuring a distance to satisfaction of rule $\rho_i$.

**Remark 2** (PSL optimisation). *Note that, because the truth values are soft and the feature functions in PSL are continuous, in contrast to MLNs, maximising the potential functions becomes a convex optimisation problem and not a combinatorial one, allowing the use of standard optimisation techniques (e.g. quadratic programming). Bach et al. (2017) introduced an even more efficient MAP inference based on consensus optimisation, where the optimisation problem is divided into independent subproblems and the algorithm iterates over the subproblems to reach a consensus on the optimum.*

**Example 10** (PSL). Let us compute the soft truth value of LIKES($\mathtt{bob}$, $\mathtt{startrek}$) for the theory of Example 8, given a partial interpretation with soft truth values:

$$
\begin{aligned}
\{\text{FRIENDS}(\mathtt{alice}, \mathtt{alice}) &= 1, \text{FRIENDS}(\mathtt{alice}, \mathtt{bob}) = 0.7, \\
\text{FRIENDS}(\mathtt{bob}, \mathtt{alice}) &= 0.5, \text{FRIENDS}(\mathtt{bob}, \mathtt{bob}) = 1, \\
\text{LIKES}(\mathtt{alice}, \mathtt{startrek}) &= 0.7\}
\end{aligned}
$$

In the remainder of this example, we will abbreviate FRIENDS by F, LIKES by L, $\mathtt{alice}$ by $\mathtt{a}$, $\mathtt{bob}$ by $\mathtt{b}$, and $\mathtt{startrek}$ by $\mathtt{s}$.

**Step 1** Map each rule to its disjunctive normal form:

$$
\begin{aligned}
\text{F}(\mathtt{U}_1, \mathtt{U}_2) \wedge \text{L}(\mathtt{U}_1, \mathtt{I}) \to \text{L}(\mathtt{U}_2, \mathtt{I}) &\equiv \neg\big(\text{F}(\mathtt{U}_1, \mathtt{U}_2) \wedge \text{L}(\mathtt{U}_1, \mathtt{I})\big) \vee \text{L}(\mathtt{U}_2, \mathtt{I}) \\
&\equiv \neg\text{F}(\mathtt{U}_1, \mathtt{U}_2) \vee \neg\text{L}(\mathtt{U}_1, \mathtt{I}) \vee \text{L}(\mathtt{U}_2, \mathtt{I})
\end{aligned}
\tag{18}
$$

**Step 2** Find the possible groundings of Equation (18):

1. $\neg\text{F}(\mathtt{a}, \mathtt{b}) \vee \neg\text{L}(\mathtt{a}, \mathtt{s}) \vee \text{L}(\mathtt{b}, \mathtt{s})$
2. $\neg\text{F}(\mathtt{b}, \mathtt{a}) \vee \neg\text{L}(\mathtt{b}, \mathtt{s}) \vee \text{L}(\mathtt{a}, \mathtt{s})$
3. $\neg\text{F}(\mathtt{a}, \mathtt{a}) \vee \neg\text{L}(\mathtt{a}, \mathtt{s}) \vee \text{L}(\mathtt{a}, \mathtt{s})$
4. $\neg\text{F}(\mathtt{b}, \mathtt{b}) \vee \neg\text{L}(\mathtt{b}, \mathtt{s}) \vee \text{L}(\mathtt{b}, \mathtt{s})$

**Step 3** Compute the potentials:

1. $F(\boldsymbol{x}_1) = \min\{x_\mathtt{s} + (1 - 0.7) + (1 - 0.7), 1\} = \min\{0.6 + x_\mathtt{s}, 1\}$
2. $F(\boldsymbol{x}_2) = \min\{(1 - 0.7) + (1 - x_\mathtt{s}) + 0.5, 1\} = \min\{1.8 - x_\mathtt{s}, 1\} = 1$
3. $F(\boldsymbol{x}_3) = \min\{(1 - 1) + (1 - 0.7) + 0.7, 1\} = \min\{1, 1\} = 1$
4. $F(\boldsymbol{x}_4) = \min\{(1 - 1) + (1 - x_\mathtt{s}) + x_\mathtt{s}, 1\} = \min\{1, 01\} = 1$

---

[2]Notice that we use the same symbols as in classical logic for the logical connectives for convenience. However, the connectives here are interpreted as defined above.

$x_i$ are the soft truth values of the ground atoms of the ground rule $i$ (of Step 2), $x_s$ is the to-be-determined soft truth value of LIKES(bob, startrek), and we applied (17).

**Step 4** Then, the probability density function is given as

$$P(\boldsymbol{x}) = \frac{\exp\left(-0.5(1 - \min\{0.6 + x_s, 1\})\right)}{Z(\boldsymbol{x})} = \frac{\exp\left(-0.5 \max\{0.4 - x_s, 0\}\right)}{Z(\boldsymbol{x})},$$

with

$$
\begin{aligned}
Z(\boldsymbol{x}) &= \int_0^1 \exp(-0.5 \max\{0.4 - x_s, 0\}) \, \mathrm{d}x_s \\
&= \int_0^{0.4} \exp(-0.5(0.4 - x_s) \, \mathrm{d}x_s + \int_{0.4}^1 1 \, \mathrm{d}x_s \\
&= \exp(-0.2) \int_0^{0.4} \exp(0.5 x_s) \, \mathrm{d}x_s + 0.6 \\
&= \exp(-0.2)\left(2 \exp(0.5 \cdot 0.4) - 2 \exp(0)\right) + 0.6 \\
&\approx 0.963 \, .
\end{aligned}
$$

**Step 5** Compute the mean estimate for $P(\text{LIKES}(\text{bob}, \text{startrek}))$:

$$
\begin{aligned}
\langle x_s \rangle &= \int_0^1 x_s \cdot P(x_s) \, \mathrm{d}x_s \\
&= \int_0^{0.4} x_s \frac{\exp(-0.2 + 0.5 x_s)}{0.963} \, \mathrm{d}x_s + \int_{0.4}^1 x_s \frac{1}{0.963} \, \mathrm{d}x_s \\
&\approx 0.515 \, .
\end{aligned}
$$

Note that in general Step 4 and Step 5 are more complex, and PSL resorts to inference algorithms as per Remark 2

### 3.3.3 Weighted Model Counting

Another notion that was proposed to unify logic with probability theory is **weighted model counting** (WMC) and its extensions. Weighted model counting captures a variety of formalisms, such as Bayesian networks (Chavira & Darwiche, 2008), their variant for relational data (Chavira et al., 2006), factor graphs (Choi et al., 2013), probabilistic programs (Fierens et al., 2015), and probabilistic databases (Suciu et al., 2011).

**TL;DR** (Weighted model counting). *Weighted model counting (Chavira & Darwiche, 2008) is an extension of model counting (#SAT) with weights on literals that can be used to represent probabilities.*

**Definition 2** (WMC). *Let $\boldsymbol{\varphi}$ be a propositional theory, $\mathtt{X}_{\boldsymbol{\varphi}}$ be the set of all Boolean variables in $\boldsymbol{\varphi}$, $w \colon \mathtt{X}_{\boldsymbol{\varphi}} \to \mathbb{R}^{\geq 0}$ and $\overline{w} \colon \mathtt{X}_{\boldsymbol{\varphi}} \to \mathbb{R}^{\geq 0}$ be two functions that assign weights to all atoms of $\boldsymbol{\varphi}$, and $\mathcal{M}$ be a model of $\boldsymbol{\varphi}$. The WMC of $\boldsymbol{\varphi}$ is defined as*

$$\text{WMC}(\boldsymbol{\varphi}; w, \overline{w}) := \sum_{\mathcal{M} \models \boldsymbol{\varphi}} \prod_{\mathtt{X} \in \mathcal{M}} w(\mathtt{X}) \prod_{\neg \mathtt{X} \in \mathcal{M}} \overline{w}(\mathtt{X}) \, . \tag{19}$$

When $w$ captures exactly the probability of a Boolean variable $\mathtt{X}$ being true, i.e. $w(\mathtt{X}) \in [0, 1]$ and $\overline{w}(\mathtt{X}) = 1 - w(\mathtt{X})$, WMC captures the probability of a formula $\boldsymbol{\varphi}$ being satisfied, by treating each Boolean variable $\mathtt{X}$ as a Bernoulli variable that becomes true with probability $w(\mathtt{X})$ and false with probability $1 - w(\mathtt{X})$. The outer summation in Equation (19) iterates over all models of $\boldsymbol{\varphi}$, while the product computes the probability to instantiate $\mathcal{M}$, given the weights of the atoms, which, due to the assumption of independence, is the product of the weights of the literals in $\mathcal{M}$.

**Example 11.** Consider the last formula of Example 4 as the theory $\varphi$ for this example:

$$\varphi := \mathtt{F}_{AB} \wedge \mathtt{L}_{AT} \rightarrow \mathtt{L}_{BT}$$

We list all possible interpretations of $\varphi$ in Table 3. Observe that $\mathcal{I}_2$ is the only interpretation that is not a model of $\varphi$. Let us denote the interpretations $\mathcal{I}_i$ by $\mathcal{M}_i$, where it is a model.

|  | $\mathtt{F}_{AB}$ | $\mathtt{L}_{AT}$ | $\mathtt{L}_{BT}$ | $\varphi$ |
|---|---|---|---|---|
| $\mathcal{I}_1$ | True | True | True | True |
| $\mathcal{I}_2$ | True | True | False | False |
| $\mathcal{I}_3$ | True | False | True | True |
| $\mathcal{I}_4$ | False | True | True | True |
| $\mathcal{I}_5$ | True | False | False | True |
| $\mathcal{I}_6$ | False | True | False | True |
| $\mathcal{I}_7$ | False | False | True | True |
| $\mathcal{I}_8$ | False | False | False | True |

Table 3: Truth table for $\varphi := \mathtt{F}_{AB} \wedge \mathtt{L}_{AT} \rightarrow \mathtt{L}_{BT}$.

Given a weight function that assigns the following weights to the atoms of the theory, $w(\mathtt{F}_{AB}) = 0.1$, $w(\mathtt{L}_{AT}) = 0.9$, $w(\mathtt{L}_{BT}) = 0.5$, and $\overline{w}(\mathtt{X}) = 1 - w(\mathtt{X})$ for all variables, we can compute the WMC of $\varphi$ as

$$
\begin{aligned}
\mathrm{WMC}(\varphi; w, \overline{w}) &= \mathrm{W}(\mathcal{M}_1; w, \overline{w}) + \mathrm{W}(\mathcal{M}_3; w, \overline{w}) + \mathrm{W}(\mathcal{M}_4; w, \overline{w}) + \mathrm{W}(\mathcal{M}_5; w, \overline{w}) \\
&\quad + \mathrm{W}(\mathcal{M}_6; w, \overline{w}) + \mathrm{W}(\mathcal{M}_7; w, \overline{w}) + \mathrm{W}(\mathcal{M}_8; w, \overline{w}) \\
&= 0.1 \cdot 0.9 \cdot 0.5 + 0.1 \cdot 0.1 \cdot 0.5 + 0.9 \cdot 0.1 \cdot 0.5 + 0.1 \cdot 0.1 \cdot 0.5 \\
&\quad + 0.9 \cdot 0.9 \cdot 0.5 + 0.9 \cdot 0.1 \cdot 0.5 + 0.9 \cdot 0.1 \cdot 0.5 \\
&= 0.595 \,,
\end{aligned}
$$

where we used W to denote the weight of a single model, i.e. $\prod_{\mathtt{X} \in \mathcal{M}} w(\mathtt{X}) \prod_{\neg \mathtt{X} \in \mathcal{M}} \overline{w}(\mathtt{X})$.

Exact WMC solvers are based on knowledge compilation (Darwiche & Marquis, 2002) or exhaustive DPLL search (Sang et al., 2005). Knowledge compilation is a paradigm which aims to transform theories to a format, such as circuits, that allows one to compute queries such as WMC in time polynomial in the size of the new representation (Darwiche & Marquis, 2002; Van den Broeck et al., 2010; Muise et al., 2012). The compilation of such circuits is generally #P-complete, and thus exponential in the worst case. The benefit of compiling such circuits is that it allows for efficient repeated querying, i.e. once the circuit is compiled one can train the weights of the model efficiently. Approximate WMC algorithms use local search (Wei & Selman, 2005) or sampling (Chakraborty et al., 2014).

The definition we discussed above supports propositional formulae and assumes discrete domains. Several extensions to generalise WMC to first-order logic and continuous domains have been proposed. WFOMC (Van den Broeck et al., 2011) lifts the problem to first-order logic and allows in the two-variable fragment (i.e. logical theories with at most two variables) to reduce the computational cost from #P-complete to polynomial-time in the domain size. WMI (Belle et al., 2015) extends WMC to hybrid (i.e. mixed real and Boolean variables) domains, allowing us to reason over continuous values. WFOMI (Feldstein & Belle, 2021) combines the two extensions to allow for lifted reasoning in hybrid domains.

### 3.3.4 Probabilistic Logic Programs

**TL;DR** (Probabilistic Logic Programs). *Building upon logic programming, probabilistic logic programs (PLPs) $\mathcal{P}(\boldsymbol{\rho}, \boldsymbol{w})$ extend the abducibles $\mathcal{A}$ with probabilities encoded by $\boldsymbol{w}$. Each fact $\overline{\alpha}_i \in \mathcal{A}$ is $\mathtt{True}$ with probability $w_i \in \boldsymbol{w}$ and $\mathtt{False}$ with probability $1 - w_i$.*

Since the original work by Poole (1993) and Sato (1995) on possible world semantics, several variations of PLPs have been proposed, e.g. stochastic logic programs (Muggleton, 1996), CP-Logic (Vennekens et al., 2009), and ProbLog (De Raedt et al., 2007).

PLPs generally assume independence between atoms, i.e. the probability of a subset $\overline{\boldsymbol{\alpha}} \subseteq \mathcal{A}$ to be `True` (and all other facts in $\overline{\boldsymbol{\alpha}}$ to be `False`) is given by

$$P_{\mathcal{P}}(\overline{\boldsymbol{\alpha}}; \boldsymbol{w}) := \prod_{\overline{\alpha}_i \in \overline{\boldsymbol{\alpha}}} w_i \prod_{\overline{\alpha}_i \in \mathcal{A} \setminus \overline{\boldsymbol{\alpha}}} 1 - w_i. \tag{20}$$

**Inference.** As for logic programs, the set of outcomes $\mathcal{O}$ is disjoint from the abducibles $\mathcal{A}$, i.e. $\mathcal{A} \cap \mathcal{O} = \emptyset$. The **success probability** of a query $Q \in \mathcal{O}$ is defined as

$$P_{\mathcal{P}}(Q \mid \boldsymbol{w}) := \sum_{\overline{\boldsymbol{\alpha}}_i \in \boldsymbol{\varphi}_{\mathcal{A}}^Q} P_{\mathcal{P}}(\overline{\boldsymbol{\alpha}}_i; \boldsymbol{w}) , \tag{21}$$

where $\overline{\boldsymbol{\alpha}}_i \in \boldsymbol{\varphi}_{\mathcal{A}}^Q$ is the set of facts in a disjunct of the abuctive formula

$$\varphi_{\mathcal{A}}^Q := \bigvee_{\substack{\overline{\boldsymbol{\alpha}}_i \subseteq \mathcal{A} \\ \text{s.t. } \overline{\boldsymbol{\alpha}}_i \cup \boldsymbol{\rho} \models Q}} \left( \bigwedge_{\overline{\alpha}_j \in \overline{\boldsymbol{\alpha}}_i} \overline{\alpha}_j \right) . \tag{22}$$

Intuitively, it is the sum of all possible interpretations of $\mathcal{A}$ that together with the rules $\boldsymbol{\rho}$ entail $Q$. Finding all interpretations of $\overline{\boldsymbol{\alpha}}$ that entail $Q$ is done by abduction. A full abduction is generally intractable since there are $2^{\mathcal{A}}$ possible worlds. However, in general, we do not need a full abductive formula, since most facts in $\mathcal{A}$ have a zero probability and can thus be ignored. One option to reason efficiently over PLPs is, thus, to only extend branches in proof trees that have a high likelihood of being successful and ignore branches with facts with low probabilities (Huang et al., 2021). Another option is to avoid redundant computations in the construction of $\boldsymbol{\varphi}_{\mathcal{A}}^Q$ (Tsamoura et al., 2023). For example, notice how the two proofs in Example 7 are equivalent. Further, notice that the semantics of PLPs in Equations (20) and (21) match with Definition 2 of WMC, where $w$ now encodes the probabilities of facts and their complements. Thus, to compute the success probability we can compile the abductive formula $\boldsymbol{\varphi}_{\mathcal{A}}^Q$ into a circuit and compute $\text{WMC}(\boldsymbol{\varphi}_{\mathcal{A}}^Q; \boldsymbol{w})$. Other (fuzzy) semantics to compute the satisfaction of $\boldsymbol{\varphi}_{\mathcal{A}}^Q$ can be used (Donadello et al., 2017).

Often, the outcomes in $\mathcal{O}$ are mutually exclusive, i.e. $\underline{\bigvee}_{\overline{o}_i \in \mathcal{O}} \overline{o}_i$. For example, when predicting what action an autonomous agent should take, it should be one out of a finite list. Computing the most probable outcome is known as **deduction**, and is computed as

$$\overline{o} = \arg\max_{\overline{o} \in \mathcal{O}} P_{\mathcal{P}}(\overline{o} \mid \boldsymbol{w}), \tag{23}$$

which we denote by $\text{deduce}(\boldsymbol{\rho}; \boldsymbol{w})$.

**Example 12** (PLP)**.** Let us extend Example 7 with the following probabilistic facts:

$$\begin{array}{l} 1.0 :: \text{FRIENDS}(\text{alice}, \text{alice}) \\ 1.0 :: \text{FRIENDS}(\text{bob}, \text{bob}) \\ 0.8 :: \text{FRIENDS}(\text{bob}, \text{alice}) \\ 0.8 :: \text{FRIENDS}(\text{alice}, \text{bob}) \\ 0.7 :: \text{KNOWNLIKES}(\text{alice}, \text{starwars}) \\ 0.9 :: \text{SIMILAR}(\text{starwars}, \text{startrek}) \end{array} \tag{24}$$

Note that all other facts in $\mathcal{A}$ are assumed to have zero probability of being `True`. To compute the probability of the PLP entailing $\text{LIKES}(\text{bob}, \text{startrek})$, we can:

1. Find all possible worlds that entail $Q$, i.e. $\boldsymbol{\varphi}_{\mathcal{A}}^Q \equiv \text{abduce}(\boldsymbol{\rho}; \mathcal{A}; \text{LIKES}(\text{bob}, \text{startrek}))$.
2. Sum the probabilities of each proof, i.e. $\sum_{\overline{\boldsymbol{\alpha}}_j \in \varphi_{\mathcal{A}}^Q} \prod_{\overline{\alpha}_i \in \overline{\boldsymbol{\alpha}}_j} w_i \prod_{\overline{\alpha}_i \in \mathcal{A} \setminus \overline{\boldsymbol{\alpha}}_j} 1 - w_i$.

From Example 7, we know that for $\text{LIKES}(\text{bob}, \text{startrek})$ to be `True` the last three facts of (24) need to be `True`, whereas the other three facts are inconsequential, i.e.

$$P_{\mathcal{P}}(\text{LIKES}(\text{bob}, \text{startrek}) | \boldsymbol{w}) = 0.8 \cdot 0.7 \cdot 0.9 = 0.504 .$$

**Training.** Learning the probabilities of facts in a PLP typically relies on circuit compilation of the abductive formula and performing WMC (Fierens et al., 2015). Similarly to LGMs, there are algorithms for learning the rules of PLPs (De Raedt & Kersting, 2004). More broadly, the field of learning the rules of logic programs is known as ***inductive logic programming*** (Muggleton & De Raedt, 1994), which is outside the scope of this survey as, generally, the frameworks presented here expect a (background) logical theory to be provided.

### 3.4  Neural Networks

This section briefly introduces concepts related to neural networks that will be useful in understanding the neuro-symbolic systems described in Sections 4 and 5. A variety of neural networks have been proposed, which have achieved outstanding performance in an array of applications such as cyber security (Berman et al., 2019), medical informatics (Litjens et al., 2017), speech recognition (Nassif et al., 2019), and human action recognition (Herath et al., 2017). However, most neuro-symbolic frameworks are model-agnostic, i.e. they do not depend on specific neural networks, and therefore, an in-depth understanding of the different models used in the experiments of the frameworks is not necessary. Where that is not the case, we will delve into more detail on the particularities of the neural networks.

**TL;DR** (Neural networks). *Neural networks are machine learning models that consist of a set of connected units – the* **neurons** *– typically organised in* **layers** *that are connected by (directed)* **edges***. Each neuron transmits a signal to the neurons it is connected to in the next layer. The parameters $\boldsymbol{w}_\mathcal{N}$ of a neural network are the weights associated with the neurons and edges. A neural network with more than three layers is called a* **deep neural network** *(DNN). Figure 9 shows an abstract representation of a feed-forward neural network.*

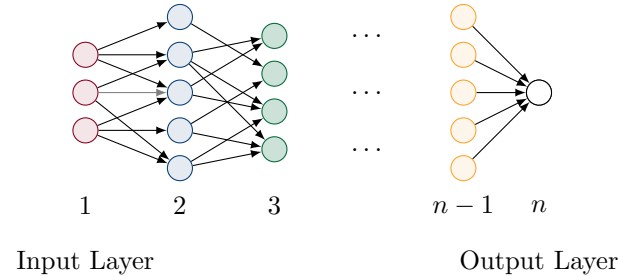

$$\text{Figure 9: Abstract representation of a feed-forward neural network.}$$

Figure 9: Abstract representation of a feed-forward neural network.

**Inference.** Inference in neural networks is performed by signals (real-valued numbers) traversing the different layers from the first layer – the ***input layer*** (Layer 1 in Figure 9) – to the final layer – the ***output layer*** (Layer $n$ in Figure 9). The input to the neural networks is data in vector form, e.g. pixels of an image organised as a vector. Then, for a layer of neurons, e.g. Layer 2 in Figure 9, the signal is propagated in the following way: the neurons take the output of the incoming connections (i.e. the neurons in Layer 1 in Figure 9), apply some (non-linear) function to the sum of the inputs, called the ***activation function***, and pass the result to the outgoing connections (i.e. the neurons in Layer 3 in Figure 9). A ***softmax*** function (a generalised logistic function) is commonly used as an output layer to obtain a well-defined probability distribution, i.e. a collection of real numbers in the interval $[0, 1]$ that sum to 1. We denote the resulting probability distribution of the neural network $\mathcal{N}$ for inputs $\boldsymbol{x}$ and parameters $\boldsymbol{w}_\mathcal{N}$ by $P_\mathcal{N}(\boldsymbol{Y} = \boldsymbol{y} | \boldsymbol{X} = \boldsymbol{x}; \boldsymbol{w}_\mathcal{N})$. From this probability distribution, one can compute its prediction as $\widehat{\boldsymbol{y}} = \arg\max_{\boldsymbol{y}} P_\mathcal{N}(\boldsymbol{Y} = \boldsymbol{y} | \boldsymbol{X} = \boldsymbol{x}; \boldsymbol{w}_\mathcal{N})$.

**Training.** Training in neural networks is performed by minimising a loss function, which typically has the form $\ell(\widehat{\boldsymbol{y}}, \boldsymbol{y})$, that represents some notion of distance between the predictions of the neural network $\widehat{\boldsymbol{y}}$ and the true labels $\boldsymbol{y}$. The goal is then to find the updated parameters $\widehat{\boldsymbol{w}}_\mathcal{N}^{t+1}$, by optimising $\widehat{\boldsymbol{w}}_\mathcal{N}^{t+1} = \arg\min_{\boldsymbol{w}_\mathcal{N}} \ell(\widehat{\boldsymbol{y}}, \boldsymbol{y})$. This is typically a non-convex optimisation. The loss is then backpropagated from the output layer to the input layer, in turn updating the weights of each layer. The optimisation is performed over several iterations,

called ***epochs***, where in each iteration the network computes predictions $\widehat{\boldsymbol{y}}$ using $\boldsymbol{w}_{\mathcal{N}}^{t}$ and then computes parameters $\boldsymbol{w}_{\mathcal{N}}^{t+1}$ that would minimise $\ell(\widehat{\boldsymbol{y}}, \boldsymbol{y})$ further.

### 3.4.1 Restricted Boltzmann Machines

Restricted Boltzmann machines (RBMs) (Smolensky, 1986) are a generative stochastic neural network inspired by Ising models from statistical physics, which describe interactions between spins in a magnetic system. In these models, lower energy states correspond to more stable and probable configurations, as lower energy reduces the entropy, or uncertainty, of the system. Similarly, RBMs use an energy function to measure the harmony between the visible and hidden layers, aiming to capture the dependencies in the data. By reducing the entropy, the information captured from the input data is maximised, as the model identifies and represents significant patterns in the data.

RBMs consist of two fully connected layers, where one layer consists of $n$ ***visible*** neurons $\boldsymbol{v}$ and the other of $m$ ***hidden*** neurons $\boldsymbol{h}$. For example, the visible neurons could be a vector of pixels from an image, while the hidden neurons describe features of the picture. Typically, all neurons are binary-valued. The connections are captured by a $n \times m$ weight matrix $\mathbf{W}$, where each entry $w_{ij}$ represents the connection from the visible neuron $v_i$ to the hidden neuron $h_j$. Note that in RBMs edges are *undirected*. In addition, a vector of bias weights can be added. We use $\boldsymbol{b}_v$ and $\boldsymbol{b}_h$ to denote the bias of the visible and the hidden neurons, respectively. Figure 10 shows an abstract representation of an RBM.

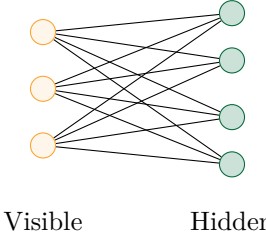

Visible       Hidden

Figure 10: Abstract representation of an RBM.

**Inference.** We can define an energy function for a state of the RBM as

$$E(\boldsymbol{v}, \boldsymbol{h}) := -\boldsymbol{b}_v^{\mathrm{T}} \boldsymbol{v} - \boldsymbol{b}_h^{\mathrm{T}} \boldsymbol{h} - \boldsymbol{v}^{\mathrm{T}} \mathbf{W} \boldsymbol{h} \ .$$

The probability of a state of the RBM can then be computed as

$$P_{\mathcal{N}}(\boldsymbol{v}, \boldsymbol{h}) := \frac{1}{Z} \exp(-E(\boldsymbol{v}, \boldsymbol{h})) \ , \tag{25}$$

where $Z$ is, as usual, the partition function computed over all possible assignments, implying that a lower energy state is a more probable one. Inference then consists in finding the assignments to $\boldsymbol{h}$ that maximise the probability in Equation (25), which is generally intractable, and thus resorts to sampling methods. For example, ***Gibbs sampling*** iteratively samples each variable conditioned on the current values of all other variables, cycling through all variables until the distribution converges.

**Training.** Training the weight matrix $\mathbf{W}$ can be implemented by a ***contrastive divergence*** algorithm (Hinton, 2002). Contrastive divergence is an iterative process consisting of a positive and negative phase. In the positive phase, the visible units are fixed, and the hidden units are sampled. In the negative phase, the visible units are sampled and the hidden units are fixed. The two sets of samples are used to update the weights.

## 4 Composite Frameworks

This section discusses neuro-symbolic architectures, where existing neural or logical models can be plugged in. These architectures have two separate building blocks: a neural component $\mathcal{N}$ and a logical component $\mathcal{L}$,

where, typically, the logical component is used to regularise the neural one. At the top level, we distinguish architectures based on how the neural network is supervised. Table 4 provides a high-level comparison of the loss functions, which will be explained in the relevant sections.

In **direct supervision** frameworks (Section 4.1), the output of the overall framework is the same as the output of the neural network. The logical component only provides an additional supervision signal to correct the neural network's predictions. The neural network is trained on the logical loss but also directly on the training labels (Table 4). These frameworks are particularly suited for applications where a neural network is already set up to solve a task, and the goal is to improve the model further (e.g. improving the accuracy with limited data or enforcing guarantees).

In **indirect supervision** frameworks (Section 4.2), the neural network identifies patterns, which are passed to a logical model for high-level reasoning. For example, the neural network identifies objects in a traffic scene (e.g. a red traffic light), and then the logical model deduces the action an autonomous car should take (e.g. stop). The output of the framework is the prediction of the logical model. The training labels are provided only for the reasoning step (e.g. stop), and the neural network is trained indirectly by having to identify patterns that allow the logical model to correctly predict the training label (e.g. a red traffic light or a stop sign but not a green traffic light).

| Architecture | Loss Function |
|---|---|
| Parallel Direct Supervision | $\widehat{\boldsymbol{w}}_{\mathcal{N}}^{t+1} = \arg\min_{\boldsymbol{w}_{\mathcal{N}}} (\ell(\widehat{\boldsymbol{y}}_{\mathcal{N}}, \boldsymbol{y}) + KL(P_{\mathcal{N}}, P_{\mathcal{L}}))$ |
| Stratified Direct Supervision | $\widehat{\boldsymbol{w}}_{\mathcal{N}}^{t+1} = \arg\min_{\boldsymbol{w}_{\mathcal{N}}} (\ell(\widehat{\boldsymbol{y}}_{\mathcal{N}}, \boldsymbol{y}) + (1 - \mathrm{SAT}(\boldsymbol{\varphi}(\widehat{\boldsymbol{y}}_{\mathcal{N}}))))$ |
| Indirect Supervision | $\widehat{\boldsymbol{w}}_{\mathcal{N}}^{t+1} = \arg\min_{\boldsymbol{w}_{\mathcal{N}}} -P_{\mathcal{L}}(\overline{o}|P_{\mathcal{N}}(\boldsymbol{y}|\boldsymbol{x}; \boldsymbol{w}_{\mathcal{N}}))$ |

Table 4: High-level comparison of loss functions in composite frameworks.

## 4.1 Direct Supervision

In direct supervision frameworks, the logical model serves as an additional supervision signal to train the neural model through regularisation. We split this family into two subgroups.

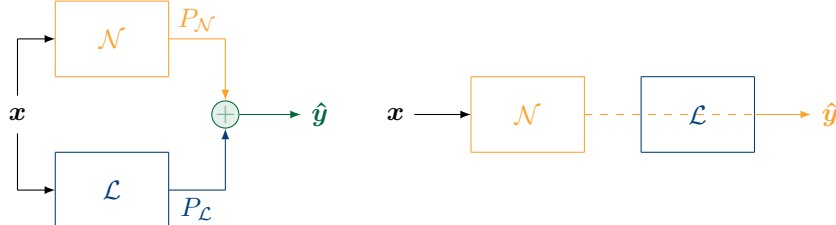

Figure 11: Abstract representation of parallel (left) and stratified (right) architectures. Symbol $\oplus$ denotes that the output $\hat{y}$ is computed by "composing" the neural predictions with the logical predictions.

In **parallel** approaches (left-hand side of Figure 11), the neural and logical model solve the same task. Thus, the input data to both models is the same or there exists a simple mapping to pre-process the data into the respective formats required by the two models. The output of both models maps to the same range. The difference between the predictions of the logical and neural models is then used as an additional loss term in the training function of the neural model. The output of the logical model is the probability $P_{\mathcal{L}}$ (e.g. Equation (12)) or the MAP (Equation (14)).

In **stratified** approaches (right-hand side of Figure 11), the neural model makes predictions first, and its outputs are then mapped to atoms of a logical theory. Violations of the logical theory are penalised in the loss function of the neural model. The output of the logical model is, generally, the SAT.

#### 4.1.1 Parallel Architectures

**TL;DR** (Parallel architectures). *Parallel architectures combine neural and logical models to distil knowledge from one into the other. The logical model can be set up using domain expertise, which can be distilled into the neural model by an additional regularisation term that measures the distance from the neural prediction to the logical prediction.*

We begin by describing the big picture and the operations in parallel supervision frameworks, abstracting away details. We will provide more details about specific frameworks that fit this architectural pattern in Section 4.1.2.

**Example 13** (Parallel architectures). Let us reconsider the running example of recommendation systems. In the case of parallel supervision, both the neural and the logical model would receive information about users and items as input, e.g. the users' job titles, age, gender, movie genres, ratings given by users for items, etc. Both models then try to predict the rating a user would give to items they have not rated before.

**Inference.** Typically, the neural model remains the main component in these frameworks, and the logical component is only used in the training phase. The inference is, therefore, generally performed in the same way as for standard neural networks (Section 3.4). Feldstein et al. (2023a) proposed a framework that takes advantage of the logical model in the inference stage by computing a weighted average of the neural and logical predictions.

**Training.** The neural model is trained by extending the loss function of a purely neural model that measures the distance between the true label $\boldsymbol{y}$ and the neural prediction $\widehat{\boldsymbol{y}}_{\mathcal{N}}$ with an additional regularisation term measuring the distance between the neural and the logical predictions $\widehat{\boldsymbol{y}}_{\mathcal{L}}$, i.e. $\widehat{\boldsymbol{w}}_{\mathcal{N}}^{t+1} = \arg\min_{\boldsymbol{w}_{\mathcal{N}}}(\ell(\widehat{\boldsymbol{y}}_{\mathcal{N}}, \boldsymbol{y}) + \ell(\widehat{\boldsymbol{y}}_{\mathcal{N}}, \widehat{\boldsymbol{y}}_{\mathcal{L}}))$. Typically, $\ell(\widehat{\boldsymbol{y}}_{\mathcal{N}}, \widehat{\boldsymbol{y}}_{\mathcal{L}})$ is implemented using the Kullback-Leibler (KL) divergence (Kullback & Leibler, 1951) between the probability distribution of the neural model and the logical model, i.e.

$$\widehat{\boldsymbol{w}}_{\mathcal{N}}^{t+1} = \arg\min_{\boldsymbol{w}_{\mathcal{N}}} \big(\pi\ell(\widehat{\boldsymbol{y}}_{\mathcal{N}}, \boldsymbol{y}) + (1-\pi)\operatorname{KL}(P_{\mathcal{N}}, P_{\mathcal{L}})\big), \tag{26}$$

where $\pi$ is an optional parameter to control the logical supervision. Example 14 illustrates why comparing entire probability distributions provides more information than simply comparing the predictions. The logical model can be trained by itself. However, recently proposed frameworks distil knowledge into the logical model. Wang & Poon (2018) compute a joint distribution from the neural and logical probability distributions and then minimise the KL divergence between the joint probability distribution and the logical probability distribution in an analogue fashion to the neural component. Feldstein et al. (2023a) consider the neural prediction as an additional predicate in the knowledge base of the logical model.

**Example 14** (Parallel architectures motivation). Consider the task of predicting the actions of people in images, and an image where a person is `crossing` a street. Mislabeling that action as `walking` is not as bad as mislabeling the action as `dancing` since `walking` is part of `crossing`. However, normally, the neural network is equally penalised for all incorrect predictions. A logical model can provide information about the likelihood of all labels in its probability distribution $P_{\mathcal{L}}$. Consider the following rules with weights $w_1 > w_2 \gg w3$:

$$w_1 \colon \forall \mathtt{P}_1, \mathtt{P}_2 \in \mathtt{People} \ \textsc{close}(\mathtt{P}_1, \mathtt{P}_2) \wedge \textsc{action}(\mathtt{P}_1, \mathtt{crossing}) \to \textsc{action}(\mathtt{P}_2, \mathtt{crossing})$$
$$w_2 \colon \forall \mathtt{P}_1, \mathtt{P}_2 \in \mathtt{People} \ \textsc{close}(\mathtt{P}_1, \mathtt{P}_2) \wedge \textsc{action}(\mathtt{P}_1, \mathtt{crossing}) \to \textsc{action}(\mathtt{P}_2, \mathtt{walking})$$
$$w_3 \colon \forall \mathtt{P}_1, \mathtt{P}_2 \in \mathtt{People} \ \textsc{close}(\mathtt{P}_1, \mathtt{P}_2) \wedge \textsc{action}(\mathtt{P}_1, \mathtt{crossing}) \to \textsc{action}(\mathtt{P}_2, \mathtt{dancing})$$

These rules state what the likelihood is of $\mathtt{P}_2$'s action if $\mathtt{P}_1$ is close to $\mathtt{P}_2$ in the image and $\mathtt{P}_1$ is `crossing`. Since $w_1 > w_2 \gg w3$, the most likely action is `crossing` but it is much more likely that $\mathtt{P}_2$ is `walking` rather than `dancing` in the streets. Note that using the MPE of $\mathcal{L}$ (Equation (14)) and optimising $\ell(\widehat{\boldsymbol{y}}_{\mathcal{N}}, \widehat{\boldsymbol{y}}_{\mathcal{L}})$ instead of $\operatorname{KL}(P_{\mathcal{N}}, P_{\mathcal{L}})$ in Equation (26) does not have the same effect. In this case, the neural network is only optimised with one additional label, which either has no effect (if $\widehat{\boldsymbol{y}}_{\mathcal{L}} = \boldsymbol{y}$, where $\boldsymbol{y}$ are the actual labels) or potentially a negative effect (if $\widehat{\boldsymbol{y}}_{\mathcal{L}} \neq \boldsymbol{y}$).

Using the entire distribution provides more information for the training than simply using the MAP (or SAT in stratified supervision). However, it has its disadvantages when it comes to constraint satisfaction. If the logical rules have large weights, the KL-term in Equation (26) encourages the neural model to satisfy these constraints but the constraints are not enforced. A solution to enforce constraints will be discussed in Section 4.1.3.

### 4.1.2 Examples of Direct Parallel Supervision Frameworks

This section compares three parallel frameworks (Figure 12). The main difference lies in how the neural and logical components are wired, whether the neural model informs the training of the logical model, and whether the logical model is used for inference.

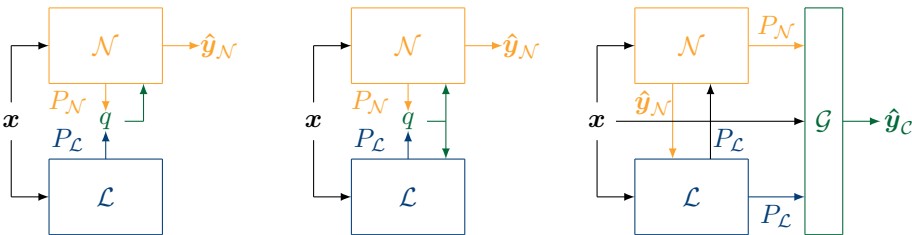

Figure 12: Abstract representation of parallel architectures used for direct supervision: (from left to right) Teacher-Student, Deep Probabilistic Logic, Concordia.

**Teacher-Student (Hu et al., 2016).** The teacher-student framework is one of the earliest attempts to combine the outputs of a neural and a logical model in a parallel fashion. Hu et al. (2016) propose to use posterior regularisation by constructing an intermediate probability distribution $q$ by solving a minimisation problem, where $q$ is optimised to be as close as possible to the neural distribution (using a KL divergence) while satisfying a set of soft constraints $\rho_i \in \boldsymbol{\rho}$ each weighted with a parameter $w_i$. $q$ is constructed by optimising

$$\min_{q, \boldsymbol{\xi} \geq 0} \mathrm{KL}(q, P_\mathcal{N}) + C \sum_{i,j} \xi_{i,j}, \text{ such that } \forall i, j : w_i(1 - \mathbb{E}_q[\rho_{i,j}(\boldsymbol{x}, \boldsymbol{y})]) \leq \xi_{i,j} , \tag{27}$$

where the $\xi_{i,j} \geq 0$ are the slack variables of the respective rules, $C \geq 0$ is the regularisation parameter, and $i$ loops over the different rules, while $j$ loops over the different groundings of each rule, with $\rho_{i,j}$ denoting the $j$-th grounding of the $i$-th rule. The logical model, therefore, implements soft logic but does not abide by the semantics of PSL due to the way the constraints are encoded in the optimisation problem. In particular, when solving the optimisation problem in Equation (27), the slack variables $\xi_{i,j}$ should approach 0 in the objective of the linear program so that the expectation of the rule satisfaction becomes 1 in the constraint of the linear program. However, when the rule satisfaction becomes 1, the weight $w_i$ of each constraint has no impact. The distribution $q$ is used as a teacher to distil the knowledge of the constraints into the neural network during training, which consists in optimising

$$\boldsymbol{w}_\mathcal{N}^{t+1} = \underset{\boldsymbol{w}_\mathcal{N}}{\arg\min}((1 - \pi)\ell(\widehat{\boldsymbol{y}}_\mathcal{N}, \boldsymbol{y}) + \pi\ell(\widehat{\boldsymbol{y}}_\mathcal{N}, \underset{y}{\arg\max}(q))) , \tag{28}$$

where $\pi$ is the user-defined ***imitation hyperparameter*** weighing the relative contribution. In contrast to other parallel frameworks, in the teacher-student framework, the neural model does not take advantage of the entire logical distribution but rather the MAP given $q$ (Equation (28)). The logical model is used to train the neural model, however, in contrast to the parallel frameworks discussed next, the neural model is not used to train the logical model. Inference is performed using only the neural model. The teacher-student framework has been used in supervised settings for named entity recognition and sentiment classification.

**DPL (Wang & Poon, 2018).** Deep Probabilistic Logic (DPL) is a framework for unsupervised learning. Similar to Hu et al. (2016), DPL computes an auxiliary distribution $q$ that is optimised with respect to the

joint distribution of the neural and the logical model:

$$q^{t+1} = \arg\min_q \mathrm{KL}(q, P_{\mathcal{L}}^t \cdot P_{\mathcal{N}}^t) \tag{29}$$

The auxiliary distribution $q$ is then used to regularise each of the components by minimising the KL divergence between $q$ and the two individual distributions of the logical and neural model, respectively, i.e.

$$P_{\mathcal{L}}^{t+1} = \arg\min_{P_{\mathcal{L}}} \mathrm{KL}(q^t, P_{\mathcal{L}}) \quad \text{and} \quad P_{\mathcal{N}}^{t+1} = \arg\min_{P_{\mathcal{N}}} \mathrm{KL}(q^t, P_{\mathcal{N}}) .$$

The benefit is that the neural and logical components are trained jointly and learn from each other. However, note that the joint distribution in Equation (29) is constructed assuming independence between the logical and neural distributions, i.e. $P_{\mathcal{L}} \cdot P_{\mathcal{N}}$. This assumption is generally an oversimplification since both models receive the same input and, therefore, are not independent. Inference is performed using only the neural model, and the logical model is dropped after training. DPL has been used in unsupervised settings for entity-linking tasks and cross-sentence relation extraction.

**Concordia (Feldstein et al., 2023a).** Concordia wires a logical and neural component in three ways. Firstly, the neural outputs can be used as priors for the logical model. For example, in the recommendation task, the simplest option is to add the rule

$$w_i : \text{DNN}(\mathtt{U}, \mathtt{I}) \rightarrow \text{LIKES}(\mathtt{U}, \mathtt{I}) ,$$

where $\text{DNN}(\mathtt{U}, \mathtt{I})$ is the neural prediction for $\text{LIKES}(\mathtt{U}, \mathtt{I})$, i.e. the logical model is told that the neural prediction is true with some confidence $w_i$. Note that $w_i \in \boldsymbol{w}_{\mathcal{L}}$ can be retrained in each epoch (using Equation (15)) and the confidence in the neural predictions can increase with each training epoch. Secondly, the logical model is used to distil domain expertise into the neural network by minimising the KL divergence between the two models during the training of the neural model as an additional supervision signal, i.e.

$$\boldsymbol{w}_{\mathcal{N}}^{t+1} = \arg\min_{\boldsymbol{w}_{\mathcal{N}}} \left( \ell(\widehat{\boldsymbol{y}}, \boldsymbol{y}) + \mathrm{KL}(P_{\mathcal{N}}(Y|\boldsymbol{X}; \boldsymbol{w}_{\mathcal{N}}^t), P_{\mathcal{L}}(Y|\boldsymbol{X}; \boldsymbol{w}_{\mathcal{L}}^t)) \right) ,$$

where the first summand minimises the difference between the prediction and the label, and the second summand reduces the difference between the distributions of the models.

Thirdly, Concordia is the only model in this section using the logical model for inference via a gating network $\mathcal{G}$, which is used to combine the neural and logical model in a mixture-of-experts approach:

$$P_{\mathcal{C}}(Y|\boldsymbol{X}; \boldsymbol{w}_{\mathcal{N}}, \boldsymbol{w}_{\mathcal{L}}, \boldsymbol{w}_{\mathcal{G}}) = \mathcal{G}(\boldsymbol{X}; \boldsymbol{w}_{\mathcal{G}})P_{\mathcal{N}}(Y|\boldsymbol{X}; \boldsymbol{w}_{\mathcal{N}}) + (1 - \mathcal{G}(\boldsymbol{X}; \boldsymbol{w}_{\mathcal{G}}))P_{\mathcal{L}}(Y|\boldsymbol{X}; \boldsymbol{w}_{\mathcal{L}})$$

In contrast to $\pi$ in the teacher-student framework, the weighting by $\mathcal{G}(\boldsymbol{X}; \boldsymbol{w}_{\mathcal{G}})$ is not fixed or user-defined but rather depends on the input and is trained together with the other two components. The framework is flexible w.r.t. the logical model (in the current implementation an LGM is expected, e.g. MLNs or PSL). However, scalability has only been shown with PSL, and thus, scalability comes at the expense of restricting the expressiveness of $\mathcal{L}$. Concordia is also model-agnostic w.r.t. the neural model, which has allowed it to be applied to a variety of models (e.g. large language models, matrix factorisation, and convolutional neural networks) in unsupervised, semi-supervised and supervised settings both for classification and regression tasks in computer vision, NLP and recommendation systems.

### 4.1.3 Stratified Architectures

**TL;DR** (Stratified architectures). *While neural models achieve great overall accuracy, their predictions can violate safety constraints or background knowledge. Post-processing neural network predictions using, e.g. a propositional satisfiability (SAT) solver, can help identify and avoid such cases.*

Here, we present how neural networks can be regularised in a stratified fashion, and the implications of using a SAT solver rather than a KL divergence. Details of specific implementations of stratified architectures are presented in Section 4.2.1.

**Inference.** The logic is generally only used during training, and thus, inference is the same as for standard neural networks (Section 3.4).

**Training.** The neural network's loss function is extended with an additional regularisation term measuring how well the neural predictions satisfy a set of formulae, i.e.

$$\widehat{\boldsymbol{w}}_{\mathcal{N}}^{t+1} = \arg\min_{\boldsymbol{w}_{\mathcal{N}}} \left( \pi \ell(\widehat{\boldsymbol{y}}_{\mathcal{N}}, \boldsymbol{y}) + (1 - \pi)(1 - \text{SAT}(\boldsymbol{\varphi}(\widehat{\boldsymbol{y}}_{\mathcal{N}}))) \right) , \tag{30}$$

where $\text{SAT}(\boldsymbol{\varphi}(\hat{\boldsymbol{y}}_{\mathcal{N}})) \in [0, 1]$ is the satisfaction of the logical theory containing the constraints, given $\hat{\boldsymbol{y}}_{\mathcal{N}}$ the predictions of the neural model as assignments to the atoms in $\boldsymbol{\varphi}$, and $\pi$ is an optional parameter to weight the enforcement of constraints. The main difference between the frameworks in this section is the semantics used to compute the SAT.

**Remark 3** (Guarantees in direct supervision). *As $\pi \to 0$ in Equation (26) of parallel frameworks, the constraints are enforced more rigorously and if the logical model has hard rules, constraints might be satisfied. However, in this case, the neural network simply learns the logical model rather than predicting the labels, making the neural network obsolete. In contrast, in Equation (30) of stratified frameworks, as $\pi \to 0$ only predictions that violate the constraints are impacted. Figure 13 illustrates (in an oversimplified fashion) the impact of hard constraints and setting $\pi \to 0$.*

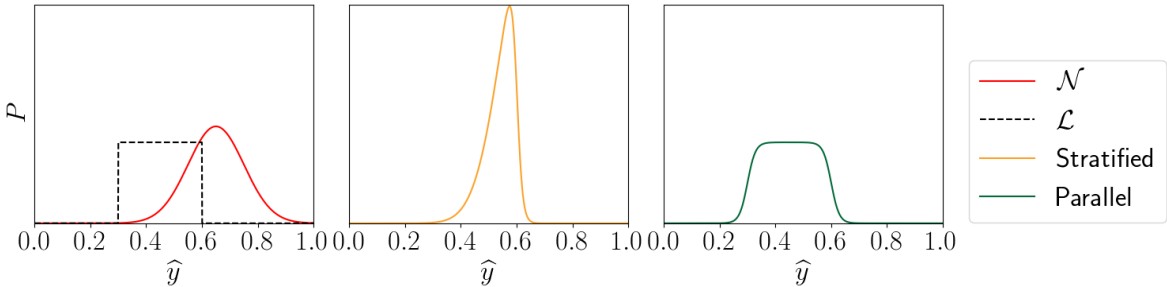

Figure 13: Impact of direct supervision on constraint satisfaction. The left plot displays probability distributions of a neural network $\mathcal{N}$ and a logical model $\mathcal{L}$. Here, $\mathcal{L}$ only constrains the output to be in $[0.3, 0.6]$ but does not discriminate between predictions within the interval. The middle plot illustrates a distribution of a framework using a SAT loss with $\pi \to 0$, and the right plot illustrates a framework with a KL loss with $\pi \to 0$.

### 4.1.4 Examples of Direct Stratified Supervision Frameworks

This section presents specific stratified stratified architectures, comparing the logical languages supported by the frameworks, the resulting loss functions, and their applications.

**DFL (Van Krieken et al., 2020).** Differentiable Fuzzy Logic (DFL) takes the outputs of the neural model as inputs to a logical model and computes the fuzzy maximum satisfiability of the theory and the derivatives of the satisfiability function w.r.t. the neural outputs. The derivatives are then used to update the parameters of the neural model to increase the satisfiability. The loss function is defined as

$$\ell_{DFL}(\boldsymbol{\varphi}; \widehat{\boldsymbol{y}}_{\mathcal{N}}) = - \sum_{\varphi_i \in \boldsymbol{\varphi}} w_i \cdot \text{SAT}(\varphi_i(\widehat{\boldsymbol{y}}_{\mathcal{N}})) ,$$

where $\widehat{\boldsymbol{y}}_{\mathcal{N}}$ are the neural predictions, $\varphi_i$ is a propositional formula, and SAT calculates the satisfaction of the formula given neural predictions. Van Krieken et al. (2020) compare a range of different implementations of fuzzy logic to compute $\text{SAT}(\varphi_i(\widehat{\boldsymbol{y}}_{\mathcal{N}}))$ and how the derivatives of the corresponding functions behave. They find that some common fuzzy logics do either not correct the premise or the conclusion and that they do not behave well in very imbalanced datasets. To counteract the identified edge cases, they introduce a new class of fuzzy logic called ***sigmoidal fuzzy logic***, which applies a sigmoid function to the satisfaction of a fuzzy

implication. The comparison of the different fuzzy logics illustrates the behaviour of modus ponens, modus tollens, distrust, and exception correction. However, the experiments are limited to the MNIST dataset (Deng, 2012) and simple rules.

**Semantic loss (Xu et al., 2018).** In order to regularise neural networks, Xu et al. (2018) proposed a semantic loss based on possible world semantics. The output of a neural network is mapped to a possible world of a propositional theory $\varphi$, which expresses the constraints on the neural network. The **_semantic loss_** is defined as

$$\ell_{\mathrm{SL}}(\varphi; P_{\mathcal{N}}) \propto - \log \sum_{\boldsymbol{x} \models \varphi} \prod_{i:\boldsymbol{x} \models X_i} P_{\mathcal{N}}(X_i = x_i) \prod_{i:\boldsymbol{x} \models \neg X_i} (1 - P_{\mathcal{N}}(X_i = x_i)),$$

i.e. the negative log probability of generating a state that satisfies the constraint when that state is sampled with the probabilities in $P_{\mathcal{N}}$. Interestingly this loss reduces to $\mathrm{WMC}(\varphi; P_{\mathcal{N}})$ (Section 3.3.3), where the weights are the probabilities for the different classes as predicted by the neural model. This loss is differentiable and syntax-independent (i.e. two formulae that are semantically equivalent have the same loss). Semantic loss achieves near state-of-the-art results on semi-supervised experiments on simple datasets (MNIST (Deng, 2012), FASHION (Xiao et al., 2017), CIFAR-10 (Krizhevsky, 2009)) while using $< 10\%$ of the training data. Further, Xu et al. (2018) show a significant improvement in constraint satisfaction when predicting structured objects in graphs, such as finding shortest paths. However, all comparisons are w.r.t. simple neural networks with generic formulae.

**Inconsistency loss (Minervini & Riedel, 2018).** The semantic loss proposed by Minervini & Riedel (2018) (called inconsistency loss), is computed as

$$\ell_{\mathrm{IL}}(\rho) = \max\{P_{\mathcal{N}}(\alpha_P) - P_{\mathcal{N}}(\alpha_C), 0\},$$

where $\alpha_P$ is the atom in the premise, and $\alpha_C$ is the atom in the conclusion of a logical rule $\rho$. Seeing that a logical rule is not satisfied if the premise is true but the conclusion is false, this loss penalises instances where the premise atom has a higher probability than the conclusion atom. Given this loss function and a propositional rule, Minervini & Riedel (2018) generate a set of adversarial examples where the neural model does not satisfy some of the constraints and train the neural model on these examples. While the overall performance only improves slightly, the presented experiments include comparisons of a neural network with and without the logical constraints on adversarial examples with positive results regarding robustness. This should be expected, as one model has specifically been trained on adversarial examples whereas the baseline has not. However, this framework illustrates, similarly to the other stratified frameworks, how neural models can be pushed to satisfy constraints, which in many applications could be more valuable than only improving the accuracy. A limitation of this framework is that it only supports a single rule with one premise and one conclusion atom. This framework was developed for NLP tasks.

**DL2 (Fischer et al., 2019).** Deep Learning with Differentiable Logic (DL2) proposes their own fuzzy logic similar to PSL in (17). Fischer et al. (2019) argue that, in some cases, PSL might stop optimising due to local optima, while DL2 would continue optimising due to different gradient computations. DL2 supports Boolean combinations of terms as constraints, where a term is either a constant, a variable, or a differentiable function over variables. In contrast to Xu et al. (2018), DL2 supports real variables (enabling constraints for regression tasks). In addition, Fischer et al. (2019) provide a language to query the neural network. This language allows, for example, to find all input-output pairs that satisfy certain requirements or to find neurons responsible for certain decisions, which could be of interest to explainability but has not been tested in that regard. Similarly to Xu et al. (2018), DL2 was tested on standard benchmarks (MNIST (Deng, 2012), FASHION (Xiao et al., 2017), CIFAR-10 (Krizhevsky, 2009)) with generic constraints, and has only been compared to purely neural but not neuro-symbolic baselines. While prediction accuracy slightly decreases in some experiments, the satisfaction of constraints increases significantly.

One limitation across all of the above frameworks is that they only support propositional logic (DL2 supports arithmetic expressions but no relations and quantifiers). In general, this is sufficient for the task at hand, as

the formulae are applied to the neural predictions and the number of possible outputs is generally limited. However, there are instances where a FOL constraint would help. For instance, when the task is to predict the actions of several people across a sequence of images (Example 14) numerous predictions are made. In this case, having logical constraints connecting the different predictions would be particularly helpful as the actions performed by the different people are likely to be linked and could be optimised jointly. One possible solution could be to lift the semantic loss using WFOMC (Van den Broeck et al., 2011) or WFOMI (Feldstein & Belle, 2021), neither of which has been tested so far. Rocktäschel et al. (2015) present a solution, which implements FOL, where the framework learns the entity and relation embeddings maximising the satisfaction of a fixed set of FOL formulae. Here, satisfaction is defined according to fuzzy semantics using the product t-norm (in contrast to the Łukasiewicz t-(co)norms used in PSL (17)). However, this framework is not model-agnostic and expects matrix factorisation neural networks.

## 4.2 Indirect Supervision

**TL;DR** (Indirect supervision)**.** *While neural models have their strength in pattern recognition, SRL frameworks lend themselves particularly well to complex reasoning. Thus, "divide and conquer" is the motto of the indirect supervision frameworks. The neural models are used for perception, and the identified patterns are passed to a (probabilistic) logic program as input for high-level reasoning. However, inconsistencies identified in the reasoning step can inform the training of the neural component. The frameworks in this section all rely on the same principle of training using a loss computed based on the abductive formula.*

Here, we present a unified framework for describing indirect supervision techniques and then highlight their main differences in Section 4.2.1. We abstract indirect supervision frameworks using the notation proposed by Tsamoura & Michael (2021).

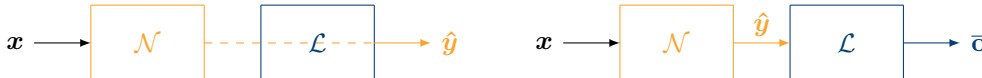

Figure 14: Abstract comparison of direct (left) and indirect (right) stratified architectures.

Similarly to stratified frameworks in Section 4.1.3, the logical component is stacked on top of the neural component. However, while in stratified direct supervision, the output of the framework is still the neural network's prediction (the logical component only "filters" the neural predictions), in indirect supervision the output of the neural component forms the input to the logical component and the output of the framework is the logical prediction. The difference is illustrated in Figure 14, where $\widehat{\boldsymbol{y}}$ are the neural predictions, and $\overline{\mathbf{o}}$ is the outcome predicted by the (probabilistic) logic program.

**Example 15** (Indirect Supervision (Adopted from Tsamoura & Michael (2021)))**.** Let our dataset consist of images of a chessboard and assume we have:

1. a DNN that recognises the pieces occurring in an image of a chessboard,

2. a logical theory that encodes the rules of chess, and

3. the status of the black king (i.e. the black king is in a draw, safe, or mate position).

The goal is to amend the parameters of the DNN so that the recognised pieces represent a chessboard in which the black king is in the given status. The main challenge here is that we are only given the status of the black king as training labels. In other words, the complete configuration of the board has to be deduced just from the image and the rules of the game.

Assuming that the training label for the input chessboard is "mate", **_abduction_** would return a logical formula encoding *all* possible chessboard configurations where the black king just got mated. If the DNN recognises any of the configurations, then reasoning using the rules of the logical theory would entail that the black king is in a mate state. Given a target, abduction returns *all* the possible inputs that should be provided to the logical theory in order to deduce the target when reasoning using the rules of the theory.

As for (probabilistic) logic programming (Section 3.2.3 and 3.3.4), let $\mathcal{A}$ be the set of **abducibles** and $\mathcal{O}$ the set of **outcomes**. For a given outcome $\overline{o} \subseteq \mathcal{O}$, let $\varphi_{\mathcal{A}}^{\overline{o}} := \mathtt{abduce}(\boldsymbol{\rho}; \mathcal{A}; \overline{o})$ denote the abductive formula – the disjunction of abductive proofs (Section 3.2.3). Observe that for any fixed outcome there might exist zero, one, or multiple abductive proofs. Let $\boldsymbol{X}$ be the space of possible inputs and $\boldsymbol{Y} = [0, 1]^k$ be the space of possible outputs of the neural model. We denote by $P_{\mathcal{N}}(\boldsymbol{y} \mid \boldsymbol{x}; \boldsymbol{w}_{\mathcal{N}})$ the probability distribution of the neural model. For notational simplicity, we assume that there is a function $\mu$ that maps each $y_i \in \boldsymbol{Y}$ to an abducible $\mu(y_i) = \overline{\alpha}_i \in \mathcal{A}$.

**Inference.** Given a logical model with a theory $\boldsymbol{\rho}$, the inference of the neuro-symbolic system is the process that maps an input in $\boldsymbol{X}$ to an outcome subset of $\mathcal{O}$ as follows: For a given input $\boldsymbol{x}$, the neural model computes the probabilities over $\boldsymbol{y}$, i.e. $P_{\mathcal{N}}(\boldsymbol{y}|\boldsymbol{x}; \boldsymbol{w}_{\mathcal{N}})$. The probability of an abducible $\mu(y_i) = \overline{\alpha}_i \in \mathcal{A}$ is $w_i = P_{\mathcal{N}}(y_i|\boldsymbol{x}; \boldsymbol{w}_{\mathcal{N}})$, and the logical model computes the outcome $\overline{o} = \mathtt{deduce}(\boldsymbol{\rho}; P_{\mathcal{N}}(\boldsymbol{y}|\boldsymbol{x}; \boldsymbol{w}_{\mathcal{N}})) \in \mathcal{O} \cup \{\bot\}$. Thus, inference proceeds by running the inference mechanism of the logical model over the inferences of the neural network. To simplify our notation, we use $h_{\boldsymbol{\rho}}(\boldsymbol{x})$ – the **hypothesis function** – to mean $\mathtt{deduce}(\boldsymbol{\rho}; P_{\mathcal{N}}(\boldsymbol{y}|\boldsymbol{x}; \boldsymbol{w}_{\mathcal{N}}))$. $\mathcal{L}$ can also implement deduction with a non-probabilistic logic program, where instead of using the probabilities, we use the predictions for the abducibles, e.g. if $P_{\mathcal{N}}(y_i|\boldsymbol{x}; \boldsymbol{w}_{\mathcal{N}}) > 0.5$ then $\overline{\alpha}_i \in \overline{\boldsymbol{\alpha}}$, and check which outcome is entailed by $\mathcal{P}(\boldsymbol{\rho}, \overline{\boldsymbol{\alpha}})$.

**Training.** As in standard supervised learning, consider a set of labelled samples of the form $\{\boldsymbol{x}_j, f(\boldsymbol{x}_j)\}_j$, with $f$ being the target function that we wish to learn, $\boldsymbol{x}_j$ corresponding to the features of the sample, and $f(\boldsymbol{x}_j)$ being the label of the sample. Training seeks to identify, after $t$ iterations over a training set of labelled samples, a hypothesis function $h_{\boldsymbol{\rho}}^t$ that sufficiently approximates the target function $f$ on a test set of labelled samples. Given a fixed theory $\boldsymbol{\rho}$ for the logical model, the only part of the hypothesis function $h_{\boldsymbol{\rho}}^t(\boldsymbol{x}) = \mathtt{deduce}(\boldsymbol{\rho}; P_{\mathcal{N}}(\boldsymbol{y}|\boldsymbol{x}; \boldsymbol{w}_{\mathcal{N}}))$ that remains to be learned is the function $P_{\mathcal{N}}(\boldsymbol{y}|\boldsymbol{x}; \boldsymbol{w}_{\mathcal{N}}))$ implemented by the neural component. $P_{\mathcal{N}}(\boldsymbol{y}|\boldsymbol{x}; \boldsymbol{w}_{\mathcal{N}})$ is learnt in two steps:

1. For the label $f(\boldsymbol{x}_j)$ of a given sample, viewed as a (typically singleton) subset of $\mathcal{O}$, find the **abductive feedback formula** $\mathtt{abduce}(\boldsymbol{\rho}; \mathcal{A}; f(\boldsymbol{x}))$, i.e.

$$\varphi_{\mathcal{A}}^{f(\boldsymbol{x})} := \bigvee_{\substack{\overline{\boldsymbol{\alpha}}_i \subseteq \mathcal{A} \\ \text{s.t. } \overline{\boldsymbol{\alpha}}_i \cup \boldsymbol{\rho} \models f(\boldsymbol{x})}} \left( \bigwedge_{\overline{\alpha}_j \in \overline{\boldsymbol{\alpha}}_i} \overline{\alpha}_j \right),$$

which consists of *all* abductive proofs $\overline{\boldsymbol{\alpha}}_i \subseteq \mathcal{A}$. The abductive feedback formula thereby captures all the acceptable outputs of the neural component that (together with the theory $\boldsymbol{\rho}$) lead the system to correctly infer $f(\boldsymbol{x})$.

2. The abductive feedback is used to supervise the neural component, by minimising

$$\ell_{\mathrm{IS}}(f(\boldsymbol{x}); P_{\mathcal{N}}(\boldsymbol{y}|\boldsymbol{x}; \boldsymbol{w}_{\mathcal{N}})) := -P_{\mathcal{L}}(f(\boldsymbol{x})|P_{\mathcal{N}}(\boldsymbol{y}|\boldsymbol{x}; \boldsymbol{w}_{\mathcal{N}})) , \tag{31}$$

which as remarked in Section 3.3.4 can be computed as $-\mathrm{WMC}(\varphi_{\mathcal{A}}^{f(\boldsymbol{x})}; P_{\mathcal{N}}(\boldsymbol{y}|\boldsymbol{x}; \boldsymbol{w}_{\mathcal{N}}))$. However, other fuzzy logic semantics to compute the satisfaction can be used as well (Tsamoura & Michael, 2021). Critically, the resulting loss function is differentiable, even if the theory $\boldsymbol{\rho}$ of the logical model is not. By differentiating the loss function, we can use backpropagation to update the neural model.

### 4.2.1 Examples of Indirect Supervision Frameworks

The techniques in this area differ w.r.t. the following aspects:

A1 The semantics of $\mathtt{deduce}$ performed by $\mathcal{L}$, (i.e. classical logic program or PLP).

A2 The subset of the derived abductive formula used for training the neural classifier $\mathcal{N}$ (e.g. the entire abductive formula or only the most promising proofs).

A3 The loss computation used for training $\mathcal{N}$ based on the abductive formula (i.e. minimising $-P_{\mathcal{L}}(f(\boldsymbol{x})|P_{\mathcal{N}}(\boldsymbol{y}|\boldsymbol{x}; \boldsymbol{w}_{\mathcal{N}}))$ in Equation (31) via WMC or fuzzy logic).

**DeepProbLog (Manhaeve et al., 2018).** DeepProbLog was the first framework proposed in this line of research. Regarding A1, DeepProbLog relies on the semantics of PLPs, while regarding A2 and A3, DeepProbLog uses the WMC of *all* the abductive proofs, i.e. the semantic loss introduced above (Xu et al., 2018), as the loss function for training the neural component. Computing the abductive formula is a computationally intensive task, severely affecting the scalability of the framework (Manhaeve et al., 2018). Recently, a few approximations to the original framework were proposed to tackle the problem of computing all the abductive proofs and then the WMC of the corresponding formula. For example, **_Scallop_** (Huang et al., 2021) trains the neural network considering only the top-$k$ abductive proofs and relies on the notion of provenance semirings (Green et al., 2007). Instead of using the top-$k$ proofs, Manhaeve et al. (2021) present an approach that relies on geometric mean approximations. Beyond the academic interest, Huang et al. (2021) have shown the merits of indirect supervision frameworks in training deep neural classifiers for visual question answering, i.e. answering natural language questions about an image.

**ABL (Dai et al., 2019).** Abductive Learning (ABL) is another framework in that line of research that employs an ad-hoc optimisation procedure. Regarding A1, ABL is indifferent to the semantics of $\mathcal{L}$ and could, for example, also use a classical (i.e. non-probabilistic) logic program. Regarding A2 and A3, ABL relies on a pseudo-labeling process to train the neural component. In each training iteration over a training dataset $\mathcal{D} = \{(\boldsymbol{x}_j, f(\boldsymbol{x}_j))\}_j$, ABL first considers different training data subsets $\mathcal{D}^t \subset \mathcal{D}$ and performs the following steps:

1. It gets the neural predictions for each element in $\mathcal{D}^t$.
2. It obscures a subset of the neural predictions, both within the same and across different training samples, i.e. it pretends to have no knowledge of the truth value of those facts.
3. It abduces the obscured predictions so that the resulting predictions are consistent with the background knowledge.

ABL performs these steps with different subsets of varying sizes. Let $\mathcal{D}^*$ be the largest $\mathcal{D}^t$ satisfying the theory after obscuring and abducing. For each $\{(\boldsymbol{x}_j, f(\boldsymbol{x}_j))\}_j \in \mathcal{D}^*$, ABL trains multiple times the neural component using obscured and abduced neural predictions. It was shown empirically that the optimisation procedure of obscuring and abducing neural predictions can be time-consuming and less accurate, compared to other techniques in this section, even when only a single abductive proof is computed (Tsamoura & Michael, 2021).

**NeuroLog (Tsamoura & Michael, 2021).** NeuroLog is an extension to DeepProbLog (Manhaeve et al., 2018), allowing for different logic semantics to be incorporated, e.g. the PLP semantics as proposed by Manhaeve et al. (2018), or the fuzzy logic semantics as proposed by Donadello et al. (2017). In addition, Tsamoura & Michael (2021) carried out an assessment of different state-of-the-art techniques for A2 and A3, which led to **_neural-guided projection_**– a proof-tree pruning technique that can prune proof tree branches that might be computationally intractable. Neural-guided projection is essentially a pseudo-labelling technique. There, the prediction of the neural component is used as a focus point, and only abductive proofs that are proximal perturbations of that point find their way into the abductive feedback. NeuroLog laid the foundations for abstracting indirect supervision frameworks, which later led to the theoretical analysis by Wang et al. (2024). Wang et al. (2024) present an analysis of different indirect supervision techniques and how changes to A2 and A3 affect the performance of such models. In addition, they derive Rademacher-style error bounds (Maurer, 2016) for WMC-based training of the neural classifiers, i.e. computing the WMC of the abductive proofs and then using its cross entropy as a loss to train the neural classifier. While NeuroLog improves in scalability compared to DeepProbLog and ABL (solving tasks prior art fails to run), the training input still had only $\mathcal{O}(10^3)$ samples, and thus, scalability remains the main limitation.

### 4.3 Discussion

All composite frameworks share the idea of keeping the logical and neural components separate. However, inherent to the differences in their architectures at a lower level (e.g. how the components are connected or the operations performed by the logical model), the benefits achieved are quite different.

**Structured reasoning and support of logic.**  Most frameworks presented in this section have been implemented and tested with only one type of SRL framework. Still, since the logical model is separate from the neural model, the logic those frameworks support is, generally speaking, not limited. This clean separation enables the usage of logical models that support complex (e.g. hierarchical and recursive) reasoning that is not supported by neural networks, which is especially taken advantage of in indirect supervision frameworks.

**Data need.**  Since logical models provide more knowledge, it is intuitive that less data is needed. By taking advantage of the entire distribution of the logical model, and thus, even more knowledge than other frameworks, parallel direct supervision frameworks seem particularly suited to reduce data need. However, while Concordia (Feldstein et al., 2023a) has shown empirically a reduction in data need by up to 20%, none of the frameworks provide theoretical results on the effect of background knowledge on sample complexity.

**Guarantees.**  Indirect supervision can provide guarantees for the overall framework, as the output of the framework is the prediction of the logic program, which can contain hard constraints. Wang et al. (2024) present theoretical results in that regard. For direct supervision frameworks, stratified architectures are particularly well suited to enforce constraints (Remark 3), and Xu et al. (2018) and Fischer et al. (2019) present very positive results in that aspect. However, no theoretical guarantees have been presented. In contrast, parallel direct supervision frameworks are not well suited to enforce guarantees.

**Scalability.**  Parallel architectures can scale well as they build on efficient LGMs and the resulting overhead remains small. Stratified direct supervision frameworks also seem to scale well, as the outputs are simply checked against a propositional formula. However, most of the frameworks have only been tested on small datasets and simple rules. In contrast, indirect supervision frameworks have limited scalability due to their reliance on PLPs.

**Knowledge integration.**  All frameworks presented in this section allow to integrate domain expertise. From an engineering perspective, the integration of domain expertise is simple as logical rules are very close to natural language and can be specified separately to the neural model. However, it is difficult to quantify how much of the integrated knowledge is actually captured by the neural network, compared to the models discussed in Section 5, where the knowledge is directly built into the neural architecture.

**Explainability.**  While there are some improvements in explainability over purely neural models, the impact on explainability for these frameworks remains limited. Indirect supervision offers explainability for the reasoning step which remains a white-box system. However, the explainability of the neural network, in all architectures discussed in this section, remains unaffected, as the logical models only guide the neural networks but no direct link between the background knowledge and the neural predictions has been established. DL2's query language (Fischer et al., 2019) could be of interest to the explainable AI community, as the language allows us to find neurons responsible for certain decisions but the framework has not been tested with explainability in mind.

## 5   Monolithic Frameworks

Up to this point, we surveyed logic-based regularisation approaches that have logical tools and solvers as components of the overall system. A natural question is how neural models could be constructed that inherently provide the capability of logical reasoning and instantiate expert knowledge. This leads us to the area of monolithic frameworks.

We identified two groups of monolithic frameworks. The first group, which we refer to as ***logically wired neural networks*** (Section 5.1), gives neurons a logical meaning and uses the edges of the neural network to implement logical rules. The second group, which we refer to as ***tensorised logic programs*** (Section 5.2), starts from a logic program and then maps logical symbols to differentiable objects. We refer the reader to d'Avila Garcez et al. (2019) for a detailed survey on frameworks that fit this section.

### 5.1 Logically Wired Neural Networks

**TL;DR** (Logically wired neural networks). *Frameworks, in this family, map logical atoms to neurons in neural networks, and the logical connectives (e.g. conjunctions and disjunctions) are emulated by the wiring between neurons. As a result, logical formulae (typically in the form of implication rules) can be replicated using neural computations.*

The earliest neuro-symbolic frameworks consist of simple neural networks whose neurons and connections directly represent logical rules in the knowledge base. Establishing a mapping between neurons and logical atoms enables us to interpret the activation of a neuron as either a positive or a negative literal. Such neural networks can then be used for logical inference, learning with background knowledge, and knowledge extraction. We distinguish between models using ***directed models*** (e.g. feed-forward neural networks (FNNs) or recurrent neural networks (RNNs)) and ***undirected models*** (e.g. RBMs).

**Inference.** The networks use standard inference techniques (Section 3.4). Because of the inherent interpretability of these approaches, knowledge extraction techniques are often proposed alongside inference.

**Training.** The frameworks based on directed models are trained using classic backpropagation with minor adjustments for recursive connections and various stopping conditions. RBM-based approaches can be trained with a variety of methods such as hybrid learning (Larochelle & Bengio, 2008) and contrastive divergence (Hinton, 2002) (Section 3.4.1).

#### 5.1.1 Frameworks Based on Directed Models

**KBANN (Towell & Shavlik, 1994).** One of the early frameworks that uses FNNs to implement propositional logic knowledge bases is knowledge-based artificial neural networks (KBANNs). KBANNs use a mapping from the rules, where for each logical variable in the knowledge base a neuron exists in the neural network, and a connection between neurons is created if and only if one of the neurons appears in the premise of a rule and the other neuron appears in the conclusion of the same rule. KBANN uses standard backpropagation to train the network and a logistic activation function (for all neurons), i.e given a vector of the incoming signals $\boldsymbol{x}$, the weights of the incoming edges $\boldsymbol{w}_{\mathcal{N}}$, and the bias of the neuron $\theta$, the output of the neuron is given by

$$\frac{1}{1 + \exp(-(\boldsymbol{w}_{\mathcal{N}}\boldsymbol{x} - \theta))}. \tag{32}$$

**C-I$^2$LP (d'Avila Garcez & Zaverucha, 1999).** The connectionist inductive learning and logic programming system (C-I$^2$LP) uses RNNs, where the atoms in the premises of the rules are mapped to input neurons in the neural network, a hidden layer implements the logical conjunctions of the rules, and the output layer consists of the atoms in the conclusions of the rules. C-I$^2$LP also uses standard backpropagation to train the network but, in contrast to KBANNs, uses a bipolar semi-linear activation function

$$\frac{2}{1 + \exp(-\beta \boldsymbol{w}_{\mathcal{N}}\boldsymbol{x})} - 1,$$

where $\beta$ is a hyperparameter to control the steepness of the activation, $\boldsymbol{x}$ are the inputs, and $\boldsymbol{w}_{\mathcal{N}}$ are the edge weights. C-I$^2$LP has been extended to CILP++ (França et al., 2014) to allow FOL via a propositionalisation technique inspired by ILP systems.

**Example 16** (KBANN and C-I$^2$LP). Consider a logic program consisting of a single rule

$$\mathrm{F_{AB}} \wedge \mathrm{L_{AT}} \rightarrow \mathrm{L_{BT}}, \tag{33}$$

where, as in Example 4, $\mathrm{F_{AB}}$ models whether Alice and Bob are friends, $\mathrm{L_{AT}}$ models whether Alice likes Star Trek, and $\mathrm{L_{BT}}$ models whether Bob likes Star Trek. Suppose $\mathrm{F_{AB}} = 1$ and $\mathrm{L_{AT}} = 1$ is fed to both networks. Figure 15 shows a KBANN and C-I$^2$LP for this program.

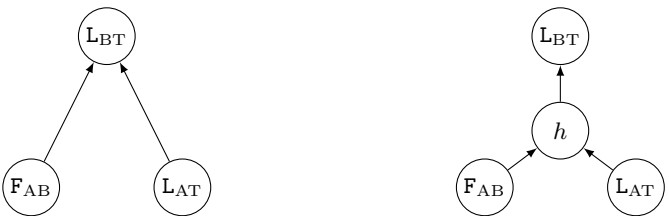

Figure 15: KBANN (left) and C-I²LP (right) implementation of the logic program (33). Both networks have a neuron for each atom, and the C-I²LP neural network has a hidden neuron $h$ for the rule itself.

In the case of **KBANN**, each edge is initialised with a weight $w$ (Towell & Shavlik (1994) suggest $w = 4$), which can later be refined using backpropagation. The bias of the $L_{BT}$ neuron, given $L$ the number of literals in the premise of the rule, is set as

$$\theta = \left(L - \frac{1}{2}\right)w = \frac{3}{2}w = 6.$$

The input to the $L_{BT}$ neuron is then $2w = 8$, and its activation value (Equation (32)) is

$$\frac{1}{1 + \exp(-(8 - \theta))} = \frac{1}{1 + \exp(-2)} \approx 0.88,$$

which is in $[0, 1]$, and so a value larger than 0.5 can be interpreted as $L_{BT}$ being set to True.

In the case of **C-I²LP**, we first set $M = \max(L_1, \ldots, L_n, R_1, \ldots R_n)$, where $L_i$ is the number of literals in the premise of rule $i$, and $R_i$ is the number of rules in the program with the same head atom as rule $i$. In our case, we only have one rule with two atoms in the premise, i.e. $M = 2$. Next, we need to choose the minimal activation $A$ to be the smallest value in $(0, 1)$ that counts as True. (Note that, unlike KBANN, C-I²LP has all values in the range $[-1, 1]$.) d'Avila Garcez & Zaverucha (1999) suggest to set $A > \frac{M-1}{M+1} = \frac{1}{3}$ (i.e. we set $A = 0.4$), and to set

$$w \geq \frac{2}{\beta} \times \frac{\ln(1 + A) - \ln(1 - A)}{M(A - 1) + A + 1} \approx 8.47,$$

where we set $\beta = 1$, and thus, we set $w = 8.5$. The activation of the $h$ neuron, given $F_{AB} = 1$ and $L_{AT} = 1$, becomes

$$\frac{2}{1 + \exp(-2\beta w)} - 1 \approx 1.$$

Finally, the activation of the $L_{BT}$ neuron becomes

$$\frac{2}{1 + \exp(-\beta w)} - 1 \approx 1.$$

Since $1 > A = 0.4$, we conclude that $L_{BT}$ is True.

Figure 16 shows a full implementation of our earlier Example 9 of both KBANN (left) and C-I²LP (right).

**Li & Srikumar (2019)**  Instead of creating a neural network from a logic program, one can alter a neural network (potentially introducing new neurons) in a way that biases the network towards satisfying some logical constraints. Li & Srikumar (2019) achieve this by recognising that the activation of some neurons (called **named neurons**) can be interpreted as the degree to which a proposition is satisfied. Suppose we want the neural network to satisfy rule $Z_1 \wedge \cdots \wedge Z_n \to Y$. For each atom, e.g. $Y$, we must first identify the corresponding neuron $y$. The goal is to increase the activation of $y$ if all $z_i$'s have high activation values but do so in a differentiable manner. Suppose originally we had that

$$y = g(\boldsymbol{w}_{\mathcal{N}} \boldsymbol{x}), \tag{34}$$

where $g$ is the activation function, $\boldsymbol{w}_{\mathcal{N}}$ are the network parameters, and $\boldsymbol{x}$ is the immediate input to $y$. Then we replace Equation (34) with

$$y = g\left(\boldsymbol{w}_{\mathcal{N}}\boldsymbol{x} + w_{\mathcal{L}} \max\left\{0, \sum_{i=1}^{n} z_i - n + 1\right\}\right), \tag{35}$$

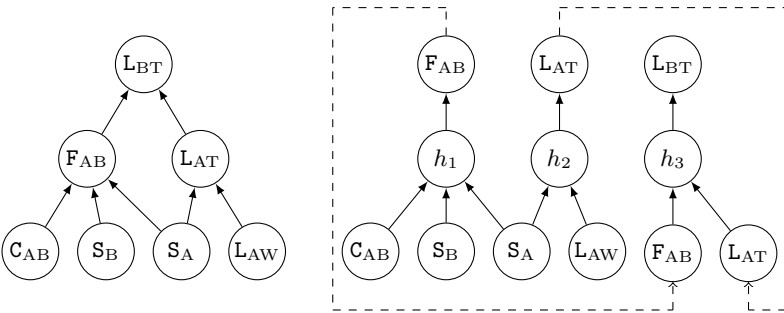

Figure 16: KBANN (left) and C-I$^2$LP (right) implementations of Example 9.

where $w_{\mathcal{L}} \geq 0$ is a hyperparameter. With Equation (35), the activation value of $y$ gets a boost if the activation values of $z_i$'s are sufficiently high. This equation is inspired by PSL (Bach et al., 2017). Li & Srikumar (2019) also discuss how Equation (35) changes depending on the structure of the premise, including the possibility of introducing new neurons to the neural network to accommodate, e.g. conjunctions of disjunctions. By considering arbitrarily large sets of atoms, one can similarly impose constraints in FOL.

The main limitation of the approaches discussed in this section is their inability to express arbitrary logic programs and perform inference on any logical variable within. Indeed, since KBANNs have no cycles, only the conclusions (and not the premises) of rules can be inferred. While C-I$^2$LP supports recurrent neural networks, there is still a restriction that the same variable can be either in the premise or the conclusion of a rule but never in both. To circumvent this issue, d'Avila Garcez et al. (2019) suggest using generative neural networks instead. The work by Li & Srikumar (2019) is similarly restrictive. First, the neural network must already possess specific neurons that are semantically equivalent to the propositions of interest. Second, the conclusion must be a conjunction of literals. Third, the rules must be ***acyclic*** w.r.t. the structure of the neural network, e.g. to enforce the rule $\texttt{X} \rightarrow \texttt{Y}$, the neural network is expected to have a directed path from $x$ to $y$.

### 5.1.2 Frameworks Based on Undirected Models

**Tran (2017)**   To handle the limited support for recursion by directed models, Tran (2017) suggests using RBMs instead. First, the knowledge base must be transformed into ***strict disjunctive normal form*** (SDNF), i.e. a disjunction of conjunctive clauses, at most one of which holds for any given truth assignment. An RBM is then constructed by creating a visible neuron for each literal and a hidden one for each clause. Then, minimising the energy of the RBM is equivalent to maximising the satisfiability of the input formula. Similarly to DNNs, one can construct ***deep belief networks*** (DBNs) by stacking RBMs, i.e. the hidden neurons of one RBM become the visible neurons of another (Smolensky, 1986). Similarly to KBANNs (Towell & Shavlik, 1994) and C-I$^2$LP (d'Avila Garcez & Zaverucha, 1999), RBM-based approaches are limited to propositional logic. Another limitation of Tran (2017) is that it requires the knowledge base to be in SDNF, and transforming a propositional formula to SDNF can be exponential-time. However, this transformation *can* be performed efficiently logical implications.

**DLNs  (Tran & d'Avila Garcez, 2016).**   Deep logic networks (DLNs) are DBNs that can be built from a (propositional) knowledge base. Alternatively, logical rules can be extracted from a trained DLN. The rules used by DLNs, called ***confidence rules***, declare an atom to be equivalent to a conjunction of literals, with a confidence value that works similarly to those in MLNs (Richardson & Domingos, 2006) and penalty logic (Pinkas, 1995). Knowledge extraction runs in $O(kn^2)$ time on a network with $k$ layers, each of which has at most $n$ neurons. However, confidence rules are extracted separately for each RBM in the stack, causing two issues for deep DLNs: they become less interpretable and either less computationally tractable or lossier. The authors also note that DLNs perform hardly better than DBNs on complex non-binary large-variance data such as images.

## 5.2 Tensorised Logic Programs

**TL;DR** (Tensorised logic programs). *Tensorised logic programs map logical symbols (e.g. predicates) to differentiable objects (vectors and tensors). This can be achieved whilst maintaining the map to the logical framework (e.g. by interpreting the neural components as logical formulae with real-number truth values) or by carrying out the logical reasoning entirely in the neural framework. Either way, end-to-end differentiability is obtained as a result.*

We review three frameworks that map (parts of) logic programs to differentiable objects: neural theorem provers (Rocktäschel & Riedel, 2017), logic tensor networks (Serafini & d'Avila Garcez, 2016; Serafini et al., 2017), and TensorLog (Cohen et al., 2020). These developments primarily arose from applications such as automated knowledge base construction. Here, natural language processing tools need to parse large texts to learn concepts and categories, e.g. human, mammal, parent, gene type, etc. However, natural language sources do not make every tuple explicit – often, they are common sense (e.g. humans are mammals, or parents of parents are grandparents). Owing to the scalability issue of dealing with thousands or even millions of tuples, there has been considerable effort and interest in enabling logical reasoning in neural networks to populate instances of logical rules. For example, embedding logical symbols, such as constants and predicates, in a vector space is a convenient way of discovering synonymous concepts.

### 5.2.1 Examples of Tensorised Logic Programs

**NTP (Rocktäschel & Riedel, 2017).** Neural theorem proving (NTP) implements an alternative DNN-based inference engine for logic programs. In a typical backward chaining inference algorithm for logic programs (Section 3.2.3; Example 7), both predicates and constants are assumed to be completely different unless they have the same name (Clocksin & Mellish, 2003; Hendricks et al., 2016). Instead, NTPs embed predicates and constants in a vector space and use vector distance measures to compare them. Neural networks are then recursively constructed in an attempt to find proofs for queries. Thus, NTPs jointly benefit from the use of logical rules as well as the similarity scores offered by neural models.

**Example 17** (NTP). Reconsider the recommendation setting and suppose we want to find all users in the knowledge base that like Star Trek, i.e. the query is LIKES(U, startrek). Moreover, suppose that the knowledge base already contains the fact LIKES(alice, starwars). The unification procedure between these two atoms would produce the substitution $\{U \mapsto \texttt{alice}\}$ along with a ***proof success score*** that depends on the distance between the embeddings of Star Trek and Star Wars. Let us assume that the distance is small, e.g. because people who like one of them tend to also like the other. Then, alice will be output as a probable candidate for liking Star Trek even though such a fact might not be deducible according to the standard logic programming semantics.

Training aims to learn the embeddings of all predicates and constants. ***Rule templates*** are used to guide the search for a logic program that best describes the data. A rule template is a rule with ***placeholder*** predicates that have to be replaced with predicates from the data. NTPs are trained using gradient descent by maximising the proof success scores of ground atoms in the input knowledge base and minimising this score for randomly sampled other ground atoms. The main advantage of NTPs is their robustness to inconsistent data, e.g. when two predicates or constants have different names but the same meaning. However, both training and inference are intractable for most real-world datasets (Rocktäschel & Riedel, 2017). Furthermore, the use of neural network-based similarity measures obfuscates the "reasoning" behind a decision, resulting in a less explainable system. Similarly, owing to the neural machinery, there are often no guarantees about logical consistency.

**LTNs (Serafini & d'Avila Garcez, 2016; Serafini et al., 2017).** Logic tensor networks (LTNs) are deep tensor networks that allow for learning in the presence of logical constraints by implementing ***real logic***. Real logic extends fuzzy logic with universal and existential quantification. Universal quantification is defined as some aggregation function (e.g. the mean) applied to the collection of all possible groundings. Existential quantification is instead Skolemized (Huth & Ryan, 2004) – enabling LTNs to support open-world semantics, i.e. the model does not assume that an existentially quantified constant comes from a finite *a priori*-defined list. Any FOL formula can then be grounded to a real number in $[0, 1]$ as follows:

- Each constant is mapped to a vector in $\mathbb{R}^n$ for some positive integer $n$.

- Each $m$-ary predicate is mapped to a $\mathbb{R}^{n \times m} \to [0,1]$ function.

- Each ground atom is then mapped to a real number in $[0,1]$, computed by applying the predicate function to the concatenation of constant vectors.

- The value of any FOL formula is computed using fuzzy logic semantics and the above-mentioned interpretations of quantification.

Such a grounding is defined by its embeddings of constants and predicates. Given a partial grounding and a set of formulae with confidence intervals, the learning task is to extend the partial grounding to a full function to maximise some notion of satisfiability. For example, maximising satisfiability can be achieved by minimising the Manhattan distance between groundings of formulae and their given confidence intervals, or by maximising the number of satisfied formulae when not provided with confidence intervals. Successful applications of LTNs include knowledge base completion (Serafini & d'Avila Garcez, 2016), assigning labels to objects in images and establishing semantic relations between those objects (Donadello et al., 2017; Serafini et al., 2017) as well as various examples using taxonomy and ancestral datasets (Bianchi & Hitzler, 2019). Bianchi & Hitzler (2019) found LTNs to excel when given data with little noise, i.e. where the input logical formulae are almost always satisfied, but, similarly to other work in this section, identified scalability issues.

**TensorLog (Cohen et al., 2020).** TensorLog is a probabilistic logic programming language that implements inference via matrix operations. It traces its lineage to probabilistic relational languages from SRL, except that it is integrated with neural networks. In TensorLog, queries are compiled into differentiable functions. This is achieved by interpreting the logical rules in fuzzy logic, i.e. conjunctions become a minimisation over real-valued atoms, and, as discussed in previous sections, fuzzy logic admits an easy integration with continuous optimisation and neural learning. In some restricted languages, TensorLog allows query computations without an explicit grounding, unlike, for example, PSL (Bach et al., 2017). In addition, a fragment of deductive databases, as admitted by "small" proof trees, is allowed, which is a deliberate attempt to limit reasoning whilst maintaining traceability.

To achieve tractability, TensorLog imposes some restrictions on the supported PLPs. TensorLog deals with non-recursive PLPs without function symbols and assumes up to two arguments per predicate. Let $\mathtt{D}$ denote the set of all constants in the database, TensorLog considers queries of the form $\mathrm{P}(\mathtt{a}, \mathtt{X})$, where $\mathrm{P}$ is a predicate, $\mathtt{a} \in \mathtt{D}$ is a logical constant, and $\mathtt{X}$ is a logical variable. Furthermore, TensorLog only considers chain-like rules of the form

$$\mathrm{P}_1(\mathtt{X}, \mathtt{Z}_1) \wedge \mathrm{P}_2(\mathtt{Z}_1, \mathtt{Z}_2) \wedge \cdots \wedge \mathrm{P}_n(\mathtt{Z}_{n-1}, \mathtt{Y}) \to \mathrm{P}(\mathtt{Y}, \mathtt{X}). \tag{36}$$

Given a query $\mathrm{P}(\mathtt{a}, \mathtt{X})$, TensorLog computes the probability of each answer to $\mathrm{P}(\mathtt{a}, \mathtt{X})$. We first describe how constants and probabilistic facts are represented in TensorLog. Let $(\overline{\boldsymbol{\alpha}}, p)$ be our database of probabilistic facts, where $\overline{\boldsymbol{\alpha}}$ is a set of facts (i.e. ground atoms), and $p \colon \overline{\boldsymbol{\alpha}} \to [0,1]$ is a mapping from facts to their probabilities. We fix an ordering of the constants in $\mathtt{D}$ and use a constant symbol $\mathtt{a}$ to denote the position of $\mathtt{a}$ in this ordering. TensorLog treats the facts in $\overline{\boldsymbol{\alpha}}$ as entries in $|\mathtt{D}| \times |\mathtt{D}|$ matrices. In particular, it associates each predicate $\mathrm{P}$ occurring in $\overline{\boldsymbol{\alpha}}$ with a $|\mathtt{D}| \times |\mathtt{D}|$ matrix $\boldsymbol{M}_{\mathrm{P}}$, where

$$\boldsymbol{M}_{\mathrm{P}}(\mathtt{a}, \mathtt{b}) = \begin{cases} p(\mathrm{P}(\mathtt{a}, \mathtt{b})) & \text{if } \mathrm{P}(\mathtt{a}, \mathtt{b}) \in \overline{\boldsymbol{\alpha}} \\ 0 & \text{otherwise} \end{cases}$$

for all constants $\mathtt{a}, \mathtt{b} \in \mathtt{D}$. Similarly, each constant $\mathtt{c} \in \mathtt{D}$ is encoded as a one-hot $|\mathtt{D}| \times 1$ vector $\boldsymbol{v}_{\mathtt{c}}$, where $\boldsymbol{v}_{\mathtt{c}}(\mathtt{c}) = 1$ and $\boldsymbol{v}_{\mathtt{c}}(\mathtt{c}') = 0$ for all constants $\mathtt{c}' \neq \mathtt{c}$. For a query of the form $\mathrm{P}(\mathtt{a}, \mathtt{X})$, TensorLog returns a $1 \times |\mathtt{D}|$ vector $\delta_{\mathtt{a}}$, where $\delta_{\mathtt{a}}(\mathtt{b})$ is the probability of $\mathrm{P}(\mathtt{a}, \mathtt{b})$ according to the given rules and facts (for any constant $\mathtt{b} \in \mathtt{D}$). Inference over a single rule can then be computed via a series of matrix multiplications. For example, if the head atom of Rule (36) is instantiated as $\mathrm{P}(\mathtt{a}, \mathtt{X})$, TensorLog would compute

$$\delta_{\mathtt{a}} = \boldsymbol{v}_{\mathtt{a}}^{\mathsf{T}} \prod_{i=1}^{n} \boldsymbol{M}_{\mathrm{P}_i}. \tag{37}$$

As all operations used by TensorLog are differentiable, one can easily learn the values of some of the matrices in equation (37) while keeping others fixed. Overall, besides the fuzzy interpretation of the logical rules in TensorLog, end-to-end differentiability offers a degree of transparency. Compared to NTPs, TensorLog is also reasonably efficient, although scalability becomes an issue with PLPs with a large number of constants. However, logical reasoning needs to be unrolled explicitly into neural computations, which needs *a priori*, often immutable, commitment to certain types of reasoning steps or proof depth. Moreover, TensorLog supports only chain-like rules, and there is no mechanism for extensibility, i.e. the ability to add new knowledge on a run-time basis.

### 5.3 Discussion

Monolithic frameworks implement logic through neural networks. Inherent to their architectures, such frameworks come with different strengths and weaknesses, compared to composite frameworks.

**Explainability.** Constructing neural models which emulate logic programs inherently leads to fully explainable models. Firstly, the neural models implement logic programs and thereby offer global explainability as the entire knowledge of the neural model can be expressed in logical rules, which are very close to natural language. Secondly, once a prediction has been made, one can back-trace through the neural network, identifying the rules that impacted the network's prediction.

**Knowledge integration.** Unlike composite frameworks, the knowledge is directly inserted into the neural model and thereby into every step of the architecture. This leads to a far tighter integration compared to composite frameworks.

**Scalability.** Scalability is a major issue for all of these systems (and limits their applicability in real-world settings) as proofs can be long, and recursively mimicking these might lead to exponentially many networks.

**Structured reasoning and support of logic.** Due to the way in which logical formulae have to be mapped to neural constructs, only limited fragments of logic are supported, e.g. Horn clauses and chain-like rules.

**Guarantees.** Most of the discussed approaches offer no guarantees of logical consistency or preserving background knowledge during training. Notable exceptions are the works of França et al. (2014) and Tran (2017) that ensure that the initial neural network is a faithful representation of the background knowledge.

**Data need.** The reduction in data need is difficult to quantify. Similarly to frameworks in the previous section, no theoretical guarantees regarding sample complexity are provided. Experimentally, a faithful comparison to purely neural networks is more difficult compared to composite frameworks, where $\mathcal{N}$ can be compared to $\mathcal{N} + \mathcal{L}$. Towell & Shavlik (1994) experimentally observe that KBANNs need less training data compared to pure FNNs because background knowledge biases the network in a simplifying manner. However, this comparison is w.r.t. an FNN with a single hidden layer. Tensorised logic programs are mainly compared to PLPs, and thus, a sample complexity improvement compared to neural networks cannot be quantified.

## 6 Related Work

The results obtained by empirical research suggest that neuro-symbolic AI has the potential to overcome part of the limitations of techniques relying only on neural networks. Unsurprisingly, neuro-symbolic AI has received increasing attention in recent years and there are already a couple of surveys on neuro-symbolic AI reporting on the achievements: d'Avila Garcez et al. (2019) focus on, what we call, monolithic neuro-symbolic frameworks. Marra et al. (2024) provide a perspective on neuro-symbolic AI through the lens of SRL, specifically PLPs, which is primarily influenced by the authors' work on DeepProbLog (Manhaeve et al., 2018). In contrast, Dash et al. (2022) present a more general overview of how domain expertise can be integrated into neural networks, including different ways the loss function of neural networks can

be augmented with symbolic constraints, e.g. the semantic-loss approach (Xu et al., 2018). d'Avila Garcez et al. (2022) offers comprehensive interpretations of neuro-symbolic AI, including the underlying motivations, challenges, and applications.

In this survey, we presented a map of the field through the lens of the architectures of the frameworks and identified meta-level properties that result from the architectural design choices. We primarily focused on regularisation approaches, as such models allow for a straightforward extension of existing neural models, and thus, are particularly easy for machine learning practitioners to adopt. We classified a large number of relevant work in the literature based on the supported logical languages (e.g. propositional or first-order logic) and model features (e.g. types of SRL frameworks and inference operations).

There are several benefits to having scoped our survey this way: Firstly, we provide a map that can be used to position future research w.r.t. other frameworks and identify closely related frameworks. Secondly, we are able to isolate and inspect the logical basis of regularisation approaches in a systematic and technically precise manner, linking the strengths and weaknesses of frameworks to their inherent composition. The underlying types of logical reasoning (e.g. MAP, SAT or abduction) are often glossed over and left implicit in the research literature despite the fact that they fundamentally affect the computational and correctness properties of the frameworks. Thirdly, this map provides researchers and engineers outside the area of neuro-symbolic AI with the necessary tools to navigate the neuro-symbolic landscape and find the architectures they need based on desired properties.

## 7 Conclusion

The expectations for neuro-symbolic AI, as outlined in the introduction, were manifold. While we discussed the different frameworks in terms of architecture, each architecture also addresses different expectations and concerns. Every framework tackles the limitations of neural networks to some extent. However, none of the architectures address all the limitations introduced at the outset of this survey but rather provide one particular benefit.

Broadly, when ***structured reasoning*** is required and the application allows for a clean separation of perception and reasoning, indirect supervision frameworks outperform other methods. Since perception and reasoning are split into two steps, each step is performed by the component best suited to the task. The neural model perceives patterns and then passes them on as inputs to the high-level reasoner. By separating the two tasks, one can use reasoning frameworks that support complex (e.g., hierarchical and recursive) logical formulae, including user-defined functions (e.g., arithmetic operations), such as ProbLog (De Raedt et al., 2007). However, scalability suffers as a consequence.

When presented with ***limited training data*** but domain expertise is provided, parallel direct supervision is likely to be the best option. Such circumstances are common in the industry as companies have vast amounts of expertise in their domain but typically low amounts of data as it is either expensive or simply not available due to privacy issues, such as in medical AI. These frameworks ensure ***scalability*** by keeping the neural network unchanged and using more scalable (lifted) SRL frameworks. However, such models only improve the accuracy and data needs of neural models but do not improve the explainability or satisfaction of constraints.

In safety-critical systems (e.g. autonomous driving), where ***guarantees*** are of concern, stratified direct supervision frameworks have the best track record, as such frameworks check every output of a neural model against a set of (potentially hard) constraints. Similar to parallel supervision, such frameworks can be scalable as the constraints are defined at the outset, and limited overhead is added compared to purely neural models. However, these frameworks come with limitations regarding complex reasoning and explainability.

If ***explainability*** is crucial to an application, the best bet is monolithic frameworks, as these models offer a high level of transparency, as neural networks can be mapped to interpretable logic programs. However, such frameworks come with limitations in scalability and the types of logical theories they support.

### 7.1 Future Avenues

**Scalability.**  Arguably, the most significant limitation of neuro-symbolic AI is the scalability as, in any case, they will have a non-zero overhead compared to purely neural models. Considerable effort has gone into scalable SRL frameworks. For example, as discussed in Remark 2, PSL is an efficient lifted graphical model. We believe that one option to improve scalability further would be to tensorise LGMs and take advantage of GPU computations. Regarding PLPs, one option to improve scalability is to develop algorithms that avoid redundant computations in the construction of proofs (Tsamoura et al., 2023) or finding approximations of the abductive formula.

**Learning formulae.**  Neural networks have the benefit that they do not need external inputs from domain experts to be developed. In contrast, symbolic methods generally rely on domain expertise to construct logical formulae. We briefly touched on the process of learning a logical model from data in Section 3.3.1 and Section 3.3.4. The main limitation is the scalability of the proposed frameworks. While there has been a push for scalable learners (Qu et al., 2021; Cheng et al., 2023; Feldstein et al., 2024), these frameworks only learn logical rules of specific shapes, and, thus, further research is needed.

**Mixing frameworks.**  We commented in this survey on how existing frameworks were suited for different properties but that none solved all the issues of neural networks mentioned in the introduction. A single neuro-symbolic framework that solves all of the limitations of neural networks seems far out of reach. The next logical step would be to take this integration further by combining different neuro-symbolic and SRL techniques into one system. One option could be to use a parallel architecture as the first stage in a stratified framework. Another option could be to combine a neural formula learner with a symbolic inductive learner and use the union as background knowledge in a graph neural network.

**Theoretical guarantees.**  It might seem intuitive that augmenting neural networks with domain expertise in the form of logical rules would enhance their performance by providing additional information. However, only a few theoretical results currently quantify the extent of this improvement, particularly in areas like improving data efficiency and establishing guarantees, theoretical results are missing. In terms of data efficiency, for instance, standard knowledge distillation offers theoretical guarantees regarding how much of the teacher model's prediction power is retained in the student model, but similar results are lacking for parallel direct supervision frameworks. Quantifying how domain knowledge reduces data requirements would enable more strategic data collection and usage, making training processes faster and less resource-intensive. Similarly, when it comes to guarantees, experimental evidence of stratified direct supervision shows that neural networks can be guided to follow constraints; however, theoretical guarantees are still missing on how closely these constraints are adhered to. Such guarantees on how logical rules enforce output constraints would make these systems more reliable and trustworthy, critical for applications in sensitive and high-stakes domains.

**Quantitative comparison.**  Finally, it is important to comment on the fact that while this is a technical survey, we have only explored the different architectures through a mathematical and computational lens. Like other surveys (d'Avila Garcez et al., 2019; d'Avila Garcez et al., 2022; Marra et al., 2024), we identified limitations and achievements in the current state but did not quantify them. A crucial element is missing to properly assess the current state of neuro-symbolic AI: a standardised benchmark for a quantitative analysis. This benchmark should include a variety of datasets, each equipped with domain knowledge. The datasets must be curated to allow evaluation of frameworks across various dimensions such as scalability, data need, and explainability rather than simply accuracy. Such a benchmark would streamline the evaluation of new frameworks, quantify their strengths and weaknesses, and highlight their merits. Additionally, it would help researchers identify gaps and guide future research. This survey could serve as a skeleton for a future survey following the same categorisation but evaluating metrics rather than theory.

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
