# OpenReview forum: "Mapping the Neuro-Symbolic AI Landscape by Architectures: A Handbook on Augmenting Deep Learning Through Symbolic Reasoning"
_TMLR — Rejected by TMLR_

### Review · Reviewer_mAvr · 2025-11-06

**Summary Of Contributions:**

This paper provides a handbook of different architectures used in the neurosymbolic literature. It provides a very good, structured high level overview of the existing approaches for Neurosymbolic learning and reasoning. There is no original, novel contribution.

The main content is in Sections 3,4,5. Section 3 introduces a long sequence of knowledge representation and reasoning frameworks, most of which are symbolic (logical and/or probabilistic), but also briefly introduces the basics of neural networks. The technical correctness of this section is very good and it is great that the authors use an unifying notation in the presentation. On the other hand, the descriptions are quite terse and some parts can be challenging for newcomers. Section 4 presents composite neuro-symbolic frameworks. The technical quality is again very good and I think that most of this section can be followed by non experts, who are looking for ways to enhance their purely neural systems. Section 5 briefly summarizes monolithic systems, that retain logical reasoning capabilities and explicit knowledge representation even after neural training. I found this section to be too dense and short, full of parts that are very hard to follow, especially for newcomers.

**Audience:**

Yes

**Audience Explanation:**

The authors had to do a hard balance between length, coverage and understandability and I am comfortable with the balance they choose, but I do find that general ML practitioners will not be able to follow large parts of the paper. So, even though the authors position the paper as an useful introduction for machine learning researchers that are not familiar with symbolic methods, I would not recommend this review for this group as I think many parts are not didactic enough for newcomers. On the other hand, I think it is a great handbook for those that are already somewhat familiar with neurosymbolic AI, but feel confused about the myriad different methods.

**Broader Impact Concerns:**

No concerns.

**Claims And Evidence:**

Yes

**Claims Explanation:**

The paper is technically very solid and provides a good coverage of th neurosymbolic research landscape. One very useful contribution is that they use a consistent terminology across different systems, making it easier to appreciate the differences and similarities. It is also very good at honestly highlighting the limitations of different approaches.

**Requested Changes:**

I have a long list of small comments, which are in general not critical to securing my recommendation and which are easy to fix. Where I think changes are critical, I will mark it with "CRITICAL".

page1:
"... do not lend themselves well to hierarchical or composite reasoning": I found this claim a bit vague and it would be nice to provide further intuition to the non expert readers. Maybe by providing an example.

page4:
"A comprehensive account of neurosymbolic concepts...." This paragraph would benefit from some rewriting. You bring in regularization techniques without explaining them and without connecting it to monolithic frameworks. I suggest you first talk about what you will say about monolithic frameworks (second half of the paragraph) and then explain how this relates to network regularization.

In Table 2, I found it confusing that you placed logical and neural models under the same entity (Models). The term "model" is a constant source of confusion in this interdisciplinary field and unless you are willing to go into some philosophical discussion about the underlying similarity, I think it is safer to emphasize that logical and neural models are completely different entities.

page5:
"We use I(X) to denote the set of all sets that can be obtained...": I think the second 'set' should be 'sequence', as the ordering is important: "... to denote the set of all sequences..."

page6:
"... denote the complement of X' in X..." replace 'in' with '\in'
In equation 4, you have 'x' on the right side, but it is unspecified. Maybe you could add that x=x' \cup x'^c

In the definition of conditional probabilities, maybe add that X_o \cup X_u = X

page8:
The description of parameterised factor graphs is hard for beginners. I think you should explain what parameterised RVs are and make the paragraph right after the TL;DR longer by a few sentences. Example 3 helps a lot, so maybe you could consider moving that example earlier.

page9:
CRITICAL
In the description of propositional logic, you have a single sentence that mentions quantification "...with a sentence being a formula in which each variable is quantified". This will not be understood for someone just learning about propositional logic. Do you ever need quantification later on? If so, introduce it properly. You later talk about theories in which you have formulas without quantifiers. I suggest you just remove this reference to quantifiers. Propositional formulas are usually without quantifiers, even though you can quantify over the boolean values.

page10:
CRITICAL
The description of First-Order Logic in 3.2.2. is too short and technical for any newcomer to understand. In it is current form it is only understandable for someone who already knows it, or at least knows various other logics. I understand that you don't want to spend pages on a longer introduction, but you should try to provide some more informal, intuitive descriptions. For example, the last sentence of 3.2.1. would be a good first sentence of this section. It is ok that the definitions are terse, but give something for those readers who will not process those parts.

page11:
CRITICAL
In the description of logic programming, you partition the Herbrand base into abducibles and outcomes. This separation is not inherent to logic programming, nothing prevents one to consider programs where, e.g., a fact is both provided and deducible via some rules. I suggest you provide 1-2 more sentences motivating the practical use cases where such sharp separation is justified. Also, it is not clear whether you want to do this separation on the level of predicates (you seem to assume that), or it is just some arbitrary partition of ground atoms.
In NeSy, we typically have a separation on the predicate level (neural predicates vs outcome predicates), but so far, you only talked about logic programming and the following perfectly ordinary logic program would not fit into your definition:

fact(1,0).
fact(N,F):-
  successor(N1,N),
  fact(N1,F1),
  mult(F1,N,F).


"A Herbrand interpretation I is a model M of P if all rules in \rho are satisfied" Besides the rules, the interpretation also has to satisfy the ground atoms \bar{\alpha}

In the paragraph labelled "Entailment", it is confusing when you write
\rho \cup \bar{\vec \alpha} \entails \vec \alpha
Here, \vec \alpha refers to completely different, unrelated facts in the left and right side of the entailment, even though they have the same "type". I suggest you use some index to distinguish them, e.g. the input atoms could be alpha_i and the entailed target atom alpha_t.

page 12:

"...predicates in the heads of the rules need to be distinct from the predicates in the input facts."
This is not true in general for logic programming, see the previous comment. Make it explicit that you do not consider recursive rules.

"since we known that" -> since we know that

At the definition of logical abduction, I think you need to make sure that only the "relevant" atoms are included. If the query is proven by atom_1, then we don't want a formula
(atom_1 \land atom_2) \lor (atom1 \land \lnot atom_2)

page 13:
In 3.3.1. you use the abbreviation "par-factor graph", but this was never introduced. Please introduce it either here or at 3.1.2.

page 17:
"PSL resorts to inference algorithms as per Remark 2": missing a point at the end of the sentence.

"Another notion that was proposed to unify logic with probability theory is weighted model counting..."
This sentence is confusing to me. As a unifier of these two frameworks, I would be looking for a knowledge representation framework that retains properties of both. But WMC is not a representation framework, rather a formal task such that several different frameworks can reduce inference to this task. Ontologically, WMC is more related to proof construction (on the logical side) and marginal computation (on the probabilistic side), each of which are crucial inference tasks for different knowledge representations. I suggest you rephrase this.

"by treating each Boolean variable X as a Bernoulli variable" -> "by treating each Boolean variable X as an independent Bernoulli variable"
In general, I suggest making the assumption of independence more explicit as it is a big/useful/questionable assumption in many applications of WMC.

page 21:
3.4.1 is too short to be understandable for newcomers.

page 21:

CRITICAL
Please revise the paragraph under Equation 26. I understand each sentence, but I do not see how they are connected and what this paragraph is trying to say.

page25:
4.1.3 Stratified Architectures: Can you explain the naming for these architectures? Why "stratified?"

page26:
"This section presents specific stratified stratified architectures" -> remove one 'stratified'

page27:
"...with positive results regarding robustness" -> this feels vague to me and I did not understand in what sense training with the logic constraint made the model better.

page28:
"...while in stratified direct supervision, the output..." remove the comma

page29:
when you talk about an outcome \bar{o}, it is not always clear to me whether you mean a single element or a subset of all outcomes. You seem to be treating it as a subset, but also assume that it is a singleton set. Please make this clearer.

page30:
"Let D^* be the largest D^t satisfying the theory after obscuring and abducing." I did not understand this sentence. Could you explain this a bit longer? Maybe even give an example?

page31:
Section 5, the description of Monolithic Frameworks is very dense. I do not think it is an useful introduction for those not familiar with these systems. I appreciate that the authors are trying to be complete by including this section, but they also admit that this section is not their focus and I wonder if it were better to just leave it out. The main merit of the section is that it collects the related works and helps interested readers to find what systems to read about.  I had several paragraph marked as hard to understand, which I will not list here, I just want to indicate that this section is not useful to teach newcomers.

page34:
"However, this transformation can be performed efficiently logical implications." You are missing a preposition in this sentence.

page38:
"this map provides researchers and engineers outside the area of neurosymbolic AI with the necessary tools to navigate the neuro-symbolic landscape"
Unfortunately, I disagree with this claim. The paper would be hard for newcomers, while it would be really useful for people already familiar with some neurosybolic systems.

---

> ### Author Response · Authors · 2025-12-12
> **Response [1/4]**
>
> Thank you for the very detailed review and suggestions. We have already managed to integrate the suggestions in the new revision.
>
> > page1: "... do not lend themselves well to hierarchical or composite reasoning": I found this claim a bit vague and it would be nice to provide further intuition to the non expert readers. Maybe by providing an example.
>
> Done. Updated text: *“Neural networks are particularly suited for pattern recognition but do not lend themselves well to hierarchical or composite reasoning, e.g. reasoning that requires multiple steps that depend on each other, and do not differentiate between causality and correlation [citation].”*
>
> > page4: "A comprehensive account of neurosymbolic concepts...." This paragraph would benefit from some rewriting. You bring in regularization techniques without explaining them and without connecting it to monolithic frameworks. I suggest you first talk about what you will say about monolithic frameworks (second half of the paragraph) and then explain how this relates to network regularization.
>
> Done - it now reads:  *“In order to position composite frameworks in relation to other approaches in the field, Section _ discusses their complement – monolithic frameworks. We keep the discussion of such frameworks brief and refer the reader to [citation] for a detailed analysis of such systems. While it is not in the scope of this survey to discuss every subcategory of this rapidly evolving field and Figure _ will likely need additional branches in the coming years, we aim with this section to cover the area of neuro-symbolic AI more broadly.”*
>
> > In Table 2, I found it confusing that you placed logical and neural models under the same entity (Models). The term "model" is a constant source of confusion in this interdisciplinary field and unless you are willing to go into some philosophical discussion about the underlying similarity, I think it is safer to emphasize that logical and neural models are completely different entities.
>
> We understand the concern but believe that where Table 2 sits, that this is not the right place for this discussion as the reader would first need to have an understanding of the different entities for that discussion. Conversely, we hope that by the end of the section, that the reader will understand the obvious difference across the two entities. Here (Table 2) we simply want to group the main sub-systems in a notational manner (calligraphic letters).
>
> > page5: "We use I(X) to denote the set of all sets that can be obtained...": I think the second 'set' should be 'sequence', as the ordering is important: "... to denote the set of all sequences..."
>
> Since I(X) is defined as a mapping, and then \bm{I)(X) is the set of all such sets, we believe the current formulation to be correct, e.g. {{x1 ->0, x2 -> 0}, {x1 ->1, x2 ->0}, …}. defining it as such allows us later to perform set operations.
>
> > page6: "... denote the complement of X' in X..." replace 'in' with '\in' In equation 4, you have 'x' on the right side, but it is unspecified. Maybe you could add that x=x' \cup x'^c
>
> We have updated the marginals equation as requested. However, regarding ‘\in’ vs ‘in’: Here ‘\in’ would be incorrect, since the ‘in’ here constrains the domain of the ‘complement’, as in “(X’c is the complement of X’) in X.” We understand that this might be confusing and added a mathematical definition of the complement.
>
> > In the definition of conditional probabilities, maybe add that X_o \cup X_u = X
> Done: Now reads *“such that $X_u = X \setminus X_o$”.*
>
> > page8: The description of parameterised factor graphs is hard for beginners. I think you should explain what parameterised RVs are and make the paragraph right after the TL;DR longer by a few sentences. Example 3 helps a lot, so maybe you could consider moving that example earlier.
>
> Done: We added a sentence to explain par-RVs and moved the example earlier.
>
> > page9: CRITICAL In the description of propositional logic, you have a single sentence that mentions quantification "...with a sentence being a formula in which each variable is quantified". This will not be understood for someone just learning about propositional logic. Do you ever need quantification later on? If so, introduce it properly. You later talk about theories in which you have formulas without quantifiers. I suggest you just remove this reference to quantifiers. Propositional formulas are usually without quantifiers, even though you can quantify over the boolean values.
>
> Done: We removed quantification from propositional logic, and defined sentences in FOL instead.

---

> > ### Author Response · Authors · 2025-12-12
> > **Response [2/4]**
> >
> > > page10: CRITICAL The description of First-Order Logic in 3.2.2. is too short and technical for any newcomer to understand. In its current form it is only understandable for someone who already knows it, or at least knows various other logics. I understand that you don't want to spend pages on a longer introduction, but you should try to provide some more informal, intuitive descriptions. For example, the last sentence of 3.2.1. would be a good first sentence of this section. It is ok that the definitions are terse, but give something for those readers who will not process those parts. <
> >
> > Done: We have moved the last paragraph from 3.2.1 to the beginning of the FOL section as requested. However, we have kept the length of the section otherwise as it was. We understand that some definitions might be difficult to follow but we believe that Example 5 should be enough to provide intuition of how FOL relates to propostional logic, and the individual definitions across the section have all examples to illustrate them as well, e.g. *“For example, $atom := friends(U_1,U_2)$ is an atom consisting of the predicate $friends$ and variables $U_1$ and $U_2$, $groundatom := friends(alice,bob)$ is a ground atom, and, for a substitution $\sigma := {U_1 \mapsto alice, U_2 \mapsto bob}$,  $atom \sigma \equiv groundatom$”.*
> >
> > > page11: CRITICAL In the description of logic programming, you partition the Herbrand base into abducibles and outcomes. This separation is not inherent to logic programming, nothing prevents one to consider programs where, e.g., a fact is both provided and deducible via some rules. I suggest you provide 1-2 more sentences motivating the practical use cases where such sharp separation is justified. Also, it is not clear whether you want to do this separation on the level of predicates (you seem to assume that), or it is just some arbitrary partition of ground atoms. In NeSy, we typically have a separation on the predicate level (neural predicates vs outcome predicates), but so far, you only talked about logic programming and the following perfectly ordinary logic program would not fit into your definition:
> > fact(1,0). fact(N,F):- successor(N1,N), fact(N1,F1), mult(F1,N,F). <
> >
> > This is quite often the case in specific logic programming frameworks like Datalog (Materializing Knowledge Bases via Trigger Graphs, VLDB 2021) and ProbLog (see, “Inference and learning in probabilistic logic programs using weighted Boolean formulas”, TPLP 2015).
> > This separation does not compromise expressivity as we can have axioms of the form R(X1,...,XN) :- R_input(X1,...,XN). In Example 7, we have “likes” both in the abducibles and outcomes, with the difference that as an abducible it is denoted as “knownlikes” and as an outcome by “likes”.
> >
> > > "A Herbrand interpretation I is a model M of P if all rules in \rho are satisfied" Besides the rules, the interpretation also has to satisfy the ground atoms \bar{\alpha} <
> >
> > Done.
> >
> > > In the paragraph labelled "Entailment", it is confusing when you write \rho \cup \bar{\vec \alpha} \entails \vec \alpha Here, \vec \alpha refers to completely different, unrelated facts in the left and right side of the entailment, even though they have the same "type". I suggest you use some index to distinguish them, e.g. the input atoms could be alpha_i and the entailed target atom alpha_t. <
> >
> > Done.
> >
> > > page 12: "...predicates in the heads of the rules need to be distinct from the predicates in the input facts." This is not true in general for logic programming, see the previous comment. Make it explicit that you do not consider recursive rules. <
> >
> > Please see our response above.
> >
> > > "since we known that" -> since we know that <
> >
> > Done
> >
> > > At the definition of logical abduction, I think you need to make sure that only the "relevant" atoms are included. If the query is proven by atom_1, then we don't want a formula (atom_1 \land atom_2) \lor (atom1 \land \lnot atom_2) <
> >
> > This is already the case, since Equation (11) is defined as a disjunction over conjunctions of ground atoms and not literals.
> >
> > > page 13: In 3.3.1. you use the abbreviation "par-factor graph", but this was never introduced. Please introduce it either here or at 3.1.2 <
> >
> > Done
> >
> > > page 17: "PSL resorts to inference algorithms as per Remark 2": missing a point at the end of the sentence. <
> >
> > Done

---

> > > ### Author Response · Authors · 2025-12-12
> > > **Response [3/4]**
> > >
> > > > "Another notion that was proposed to unify logic with probability theory is weighted model counting..." This sentence is confusing to me. As a unifier of these two frameworks, I would be looking for a knowledge representation framework that retains properties of both. But WMC is not a representation framework, rather a formal task such that several different frameworks can reduce inference to this task. Ontologically, WMC is more related to proof construction (on the logical side) and marginal computation (on the probabilistic side), each of which are crucial inference tasks for different knowledge representations. I suggest you rephrase this. <
> > >
> > > Done: We fully agree with the reviewer that WMC is a formal task and should not be seen as a framework. We somewhat struggled with the reviewer’s remark though as we are not certain how our phrasing gave the impression that WMC is “a unifier of these two frameworks” when stating “notion [...] unifying logic with probability theory”, neither of which are frameworks. We have rephrased it now, as follows, and kindly ask the reviewer to let us know whether the new phrasing removes any confusion: *“​​A formal task unifying logic and probability theory that is solved as a component of many neuro-symbolic frameworks is weighted model counting (WMC) and its extensions.”*
> > >
> > > > "by treating each Boolean variable X as a Bernoulli variable" -> "by treating each Boolean variable X as an independent Bernoulli variable" In general, I suggest making the assumption of independence more explicit as it is a big/useful/questionable assumption in many applications of WMC. <
> > >
> > > Done.
> > >
> > > > page 21: 3.4.1 is too short to be understandable for newcomers. <
> > >
> > > We understand that this section would be too short for newcomers to implement RBMs. However, RBMs are minimally used only in Section 5. We believe that this gives enough background to follow that section.
> > >
> > > > page 21: CRITICAL Please revise the paragraph under Equation 26. I understand each sentence, but I do not see how they are connected and what this paragraph is trying to say. <
> > >
> > > Done – it now reads: *“where $\pi$ is an optional parameter to control the strength of the logical supervision. Notice that Equation (26) only modifies the training of the neural network by adding supervision from the logical model – it does not modify the training of the logical model. Indeed, one approach to training the logical model in parallel architectures is to keep the training of the logical model unmodified, i.e. using Equation (14), and thus, independent from the neural network. However, as illustrated in detail in the next section, more recent frameworks present methods to also use the neural network to train the logical model for mutual improvements. DPL compute a joint distribution from the neural and logical probability distributions. This joint distribution is then used to supervise the training of both the neural and logical component analogously to Equation (26). In contrast, Concordia use the neural prediction as an additional predicate in the knowledge base of the logical model. Example 14  motivates why comparing entire probability distributions provides more information than simply comparing the predictions.”*
> > >
> > > > page25: 4.1.3 Stratified Architectures: Can you explain the naming for these architectures? Why "stratified?" <
> > >
> > > “Stratified” means that the frameworks are layered or stacked. We use it to distinguish parallel architectures, where the neural and logical model perform tasks at the same time vs sequentially. Please refer to Figure 11.
> > >
> > > > page26: "This section presents specific stratified stratified architectures" -> remove one 'stratified' <
> > >
> > > Done

---

> > > > ### Author Response · Authors · 2025-12-12
> > > > **Response [4/4]**
> > > >
> > > > > page27: "...with positive results regarding robustness" -> this feels vague to me and I did not understand in what sense training with the logic constraint made the model better.
> > > >
> > > > The model performed better on adversarial examples compared to the pure neural network method, quote from the paper: “Furthermore, we can see that regularised models are generally more robust to adversarial examples.”
> > > > However, we will replace this paper with more recent publications as requested by other reviewers, since we believe that this paper does not add much additional content compared to the other papers and we want to keep it at 3-4 example frameworks per section.
> > > >
> > > > > page28: "...while in stratified direct supervision, the output..." remove the comma
> > > >
> > > > Done
> > > >
> > > > > page29: when you talk about an outcome \bar{o}, it is not always clear to me whether you mean a single element or a subset of all outcomes. You seem to be treating it as a subset, but also assume that it is a singleton set. Please make this clearer.
> > > >
> > > > In this context, outcome \bar{o} is a subset of all outcomes. We did not see where we referred to it as a singleton.
> > > >
> > > > > page30: "Let D^* be the largest D^t satisfying the theory after obscuring and abducing." I did not understand this sentence. Could you explain this a bit longer? Maybe even give an example?
> > > >
> > > > Assume the goal is to predict a sum of two digits, and the neural predictions for two numbers is (1,4) with a target sum of 2. The frameworks randomly masks predictions e.g. (1, _) (“obscuring”) and chooses abductive proofs to fill in the missing values e.g. (1,1) (“abducing”). It does so for all training instances, e.g. given predictions {(1,4); (9,2); (3,2)} it obscures randomly, e.g. {(1,_); (_,2); (_,2)} then abduces the obscured entries {(1,1);(0,2);(0,2)} and then uses the abduced values to train the neural model, i.e. change the weights so that the predictions of the neural network match the values of the abduced values.
> > > > Please let us know whether this example is clear and we will add it to the final version.
> > > >
> > > > > page31: Section 5, the description of Monolithic Frameworks is very dense. I do not think it is a useful introduction for those not familiar with these systems. I appreciate that the authors are trying to be complete by including this section, but they also admit that this section is not their focus and I wonder if it were better to just leave it out. The main merit of the section is that it collects the related works and helps interested readers to find what systems to read about. I had several paragraph marked as hard to understand, which I will not list here, I just want to indicate that this section is not useful to teach newcomers.
> > > >
> > > > Please see our general comment about content addition/subtraction.
> > > >
> > > > > page34: "However, this transformation can be performed efficiently logical implications." You are missing a preposition in this sentence.
> > > >
> > > > Done
> > > >
> > > > > page38: "this map provides researchers and engineers outside the area of neurosymbolic AI with the necessary tools to navigate the neuro-symbolic landscape" Unfortunately, I disagree with this claim. The paper would be hard for newcomers, while it would be really useful for people already familiar with some neurosybolic systems.
> > > >
> > > > Please see our general comment regarding audiences and the perspective of the other reviewers.

---

> > > > > ### Comment · Reviewer_mAvr · 2025-12-15
> > > > > **Response to the responses**
> > > > >
> > > > > Dear Authors,
> > > > >
> > > > > Thanks for your response and for incorporating most of my suggestions. I am overall comfortable with your response and I accept those few items that you did not want to change.
> > > > >
> > > > > Two quick answers:
> > > > >
> > > > > page29: when you talk about an outcome \bar{o}, it is not always clear to me whether you mean a single element or a subset of all outcomes. You seem to be treating it as a subset, but also assume that it is a singleton set. Please make this clearer.
> > > > > In this context, outcome \bar{o} is a subset of all outcomes.
> > > > > We did not see where we referred to it as a singleton.
> > > > >
> > > > > In the inference section of page 29, you seem to write that it is a single element, but maybe it is just a typo:
> > > > > \bar{o} = decuce(...) \in \mathcal{O} \cup {\bottom}
> > > > >
> > > > > ----
> > > > > page30: "Let D^* be the largest D^t satisfying the theory after obscuring and abducing." I did not understand this sentence. Could you explain this a bit longer? Maybe even give an example?
> > > > > Assume the goal is to predict a sum of two digits, and the neural predictions for two numbers is (1,4) with a target sum of 2. The frameworks randomly masks predictions e.g. (1, ) (“obscuring”) and chooses abductive proofs to fill in the missing values e.g. (1,1) (“abducing”). It does so for all training instances, e.g. given predictions {(1,4); (9,2); (3,2)} it obscures randomly, e.g. {(1,); (,2); (,2)} then abduces the obscured entries {(1,1);(0,2);(0,2)} and then uses the abduced values to train the neural model, i.e. change the weights so that the predictions of the neural network match the values of the abduced values. Please let us know whether this example is clear and we will add it to the final version.
> > > > >
> > > > > Yes, this is clear and very helpful, thanks. I believe adding these kind of examples make a huge difference with respect to what kind of newcomer can understand the text.

---

### Review · Reviewer_NME3 · 2025-11-16

**Summary Of Contributions:**

The paper presents a survey of neuro-symbolic AI (NeSy) from an architecture-based perspective: the core contribution of the paper is a taxonomy of NeSy approaches based on the architectural design decisions behind their realization. The NeSy landscape is divided into "Composite Frameworks", where neural and logical components remain separate and communicate via interfaces; and "Monolithic Frameworks", where logic/reasoning is embedded  into the neural architecture. Composite frameworks are further divided into "Direct Supervision" ones, where the symbolic model guides the training process, or filters the NN predictions to ensure compliance with requirements; and "Indirect Supervision" frameworks, where the symbolic model is used to abductively generate missing labels and induce a differentiable loss that allows to train the system with the indirect signal. In turn, monolithic frameworks are divided into "Logically Wired NNs", where the logic is translated into the neural architecture and "Tensorised Logic Programs", where the logic is translated into tensors.

For each of the families above, the paper goes into an analysis illustrating some representative approaches per family, using high-level running examples, but also, in-depth, worked-out examples of a framework's functionality on occasion (e.g. BANN, C-I2LP).

Moreover, at the end of each section on a particular family, a discussion follows that positions the family in the NeSy landscape from the perspective of the six dimensions/aspects - outlined in the Introduction - that motivate the development of NeSy techniques, i.e. (i) Structured reasoning and support of logic; (ii) Data need; (iii) Guarantees; (iv) Scalability; (v) Knowledge integration and (vi) Explainability.

This taxonomy and its presentation is useful and constitute a strong contribution that can help researchers and practitioners navigate the NeSy literature, which is arguably quite fragmented. Although several aspects of the taxonomy have already been addressed in other surveys, i.e. Garcez et al. (2019) - monolithic approaches, Marra et al. (2024) - composite frameworks, Dash et al. (2022) - knowledge integration in NNs, the paper clearly positions itself against existing work in Section 6.

However, a "summary" section, perhaps with a concise and informative table, outlining the main findings and the tradeoffs of the various approaches w.r.t. the six NeSy dimensions would be valuable. It would also strengthen the second claimed contribution of the paper, i.e. a "handbook flavor" that the authors have tried to give to the paper. Although this would be very useful, this handbook perspective is currently not clear in the paper. The paper does provide guidance on the different NeSy families, but it often remains at a conceptual level.

A third claimed contribution is an introduction to Statistical Relational Learning, as the main means to integrate structured reasoning into NeSy AI. The purpose and length of this part is a bit questionable, see below.

Strengths:
- The paper is mostly clear and well-written.
- The NeSy AI desiderata, codified in the six NeSy dimensions/aspects in the introduction is a useful guidance on how to think about NeSy AI and what to look for when utilizing NeSy AI techniques.
- The core of the paper, i.e. the presented NeSy AI taxonomy and its exposition (Sections 4 and 5) is useful and comprehensive, often with a tutorial flavor. Researchers and practitioners can clearly benefit from it.
- Both a content strength and a presentation weakness: valuable information is contained in the Conclusions section (Section 7), especially regarding open problems/future research. I am saying that this is a weakness because this section could be extended (see below for concrete suggestions) and be a core section in the paper. In any case, this is not conclusions section material.

Weaknesses:
- The presentation could benefit from a tighter narrative: the authors state in Section 2 that the will mostly focus on composite frameworks, since these are more straightforward to work with, especially for newcomers to NeSy AI who wish to combine NNs with knowledge as quickly and easily as possible. As per this view by the authors, monolithic frameworks are only presented briefly, so as to have a "comprehensive account of neuro-symbolic concepts". However, this is not how things work in the paper. The transition from Section 4 (composite) to Section 5 (monolithic) is a bit abrupt and the monolithic part does not read as a "complement" to the composite part for conceptual completeness. It is not that brief either, 6 pages vs 9 pages for composite. In fact, I do not see a reason for a focus on composite approaches. Both have their strengths and weaknesses (e.g. ease of implementation + formal probabilistic semantics + interpetability vs scalability) and the paper already does a good job outlining the main aspects of both kinds of approaches. What is missing a more clear and "codified" account of such strengths and weaknesses, something like a summary section mentioned above, that would guide researchers and practitioners to select the proper framework given different reasoning, data, safety/correctness, interpretability and scalability requirements/constraints.
- The handbook perspective is unclear. The point above about codifying tradeoffs between frameworks could strengthen that. In addition, practical examples and "recipes" from the perspective of a practitioner could be included to support the handbook flavor of the paper. For instance, a concrete application on a reasoning-augmented neural task and a skeleton, perhaps with pseudocode, or actual code snippets, on how to implement it as a composite NeSy system. Mentioning specific tools and libraries especially on the symbolic side of things would also be very helpful for composite frameworks (solvers, reasoners, compilers e.g. circuit compilers, SAT/SMT solvers, ASP/ProbLog reasoners).
- Section 3 on SRL is too lengthy (18 pages out of the 40 in total). It reads as a tutorial on SRL and although some parts are useful for the rest of the paper (MAP/MPE, abduction, WMC) most of the material in this section is never used again. There are several resources for SRL tutorials and having such a lengthy introduction to SRL in a NeSy survey/handbook does not seem very useful. I would gladly read a good 40-pages survey/tutorial/handbook on NeSy AI, but such a length warrants additional content, that would replace a large part of the ~ 20 pages on SRL and logic - see specific suggestions below, in addition to the points above for strengthening the presentation and the handbook perspective.
- Conclusions (Section 7): The first part (up to Section 7.1) could be the basis of the more structured comparison between the different families of approaches and a guideline of when to use which. In my view, this, along with Figure 1 in the paper would be the main take-away of the paper. Section 7.1: There is valuable information here, which by no means should be presented in the conclusions section. This should be a separate section on open problems/research directions, where each of the topics currently in the list should be discussed in more details and with additional references. I would also add at least two more directions: (a) reasoning shortcuts and mitigation strategies; (b) NeSy in the era of LLMs/foundation models/Generative AI.

**Additional Comments:**

No additional comments

**Audience:**

Yes

**Audience Explanation:**

The survey is very useful for researchers and practitioners in the broader AI/ML community, as explained in the summary of contributions.

**Broader Impact Concerns:**

No concerns

**Claims And Evidence:**

Yes

**Claims Explanation:**

Although the core contribution of this survey (the NeSy AI systems taxonomy) can be strengthened in terms of presentation and narrative, it is useful and missing from the current literature.

The handbook perspective is less clear, but I think that it can be highlighted in a revised version of the paper, as outlined in the summary of contributions above.

**Requested Changes:**

- A more cohesive and structured comparison of the different families of NeSy approaches and some codified guidelines of when to use which. That could replace the current presentation of monolithic approaches being a complement to the main focus of compositional ones for conceptual completeness, which currently does not work in the paper (it makes it a bit fragmented and inconclusive). In my view, such a comparison, guidelines and practical considerations for using each framework would be the main take-away message of the paper.
- Strengthen the handbook perspective of the paper as outlined in the summary of contributions above.
- Strengthen the open problems/future research section and present it as a separate section (not in the conclusions).
- Reduce the length of Section 3 to what is necessary to follow the material in the rest of the paper. Use the extra space to strengthen more important aspects in the paper, as outlined above.

---

### Review · Reviewer_oUL7 · 2025-11-28

**Summary Of Contributions:**

This survey aims to map the neuro-symbolic AI (NeSy) landscape through an architectural taxonomy that distinguishes composite and monolithic approaches, further subdivided by supervision type and logic–network integration strategies. The manuscript includes a substantial amount of background material on probabilistic graphical models, SRL, and logic-based learning, and provides a structured overview of many classical NeSy frameworks.

**Audience:**

Yes

**Audience Explanation:**

The initial sections that introduce key notions from SRL, probabilistic logic programming, and graphical models are very well written.

The notation is consistent, the running examples are helpful, and the TL;DR-style summaries make the content accessible to newcomers.

As an introductory primer for students entering NeSy from a deep-learning background, these parts of the survey are really useful.

**Broader Impact Concerns:**

No concerns

**Claims And Evidence:**

No

**Claims Explanation:**

The largest gap in the survey is in its coverage of post-2022 developments.

A careful count of the reference list shows:

- 2022 references: 4

- 2023 references: 5

- 2024 references: 3

- 2025 references: 0

In addition, most of the 2023–2024 citations originate from the same research group, which gives the impression that the recent literature is filtered through a narrow subset of the community. Given how fast the field is evolving, this is a significant limitation for a survey whose goal is to map the current landscape.

A broader and more balanced inclusion of work from the last three years would considerably strengthen the paper.

**Requested Changes:**

The current taxonomy is heavily SRL-centric. It does not naturally accommodate more recent NeSy paradigms such as diffusion-based generative NeSy, prototype-augmented NeSy, or concept-bottleneck-driven architectures.

Cross-cutting dimensions are implicit rather than explicit.

For example:

– generative vs discriminative models,

– hard vs soft constraints,

– symbolic supervision vs induced concepts,

– shallow vs multi-step reasoning.

Making these axes explicit could help situate many of the newer methods. Several emerging categories have no natural home in the current map.

A small extension of the taxonomy—e.g., adding branches for generative symbolic latents or concept-bottleneck NeSy—would greatly increase its accuracy for 2023–2025 work.

Additionally, I believe a brief discussion on the problems connected to the reasoning shortcuts could really help in pointing the reader towards possible future research directions.

Below I include some references which might be useful to guide the authors. Notice that this is a small subsection - and for a thorough mapping additional research is needed.

**Reasoning shortcuts:**

Emanuele Marconato, Stefano Teso, Antonio Vergari, and Andrea Passerini. Not all neuro-symbolic concepts are created equal: Analysis and mitigation of reasoning shortcuts. In NeurIPS, 2023b.

Emanuele Marconato, Samuele Bortolotti, Emile van Krieken, Antonio Vergari, Andrea Passerini, and Stefano Teso. BEARS Make Neuro-Symbolic Models Aware of their Reasoning Shortcuts. Uncertainty in AI, 2024.

Samuele Bortolotti, Emanuele Marconato, Tommaso Carraro, Paolo Morettin, Emile van Krieken, Antonio Vergari, Stefano Teso, and Andrea Passerini. A neuro-symbolic benchmark suite for concept quality and reasoning shortcuts. In A. Globerson, L. Mackey, D. Belgrave, A. Fan, U. Paquet, J. Tomczak, and C. Zhang, editors, Advances in Neural Information Processing Systems, volume 37, pages 115861–115905. Curran Associates, Inc., 2024.

Emile van Krieken, Pasquale Minervini, Edoardo Ponti, and Antonio Vergari. Neurosymbolic reasoning shortcuts under the independence assumption. In Proceedings of the 19th International Conference on Neurosymbolic Learning and Reasoning, volume 284 of Proceedings of Machine Learning Research. PMLR, 2025a.

**Concept Bottleneck models**

Pietro Barbiero, Gabriele Ciravegna, Francesco Giannini, Mateo Espinosa Zarlenga, Lucie Charlotte Magister, Alberto Tonda, Pietro Lio, Frederic Precioso, Mateja Jamnik, Giuseppe Marra Interpretable Neural-Symbolic Concept Reasoning Proceedings of the 40th International Conference on Machine Learning, PMLR 202:1801-1825, 2023.

**Hard Constraints:**

Nick Hoernle, Rafael Michael Karampatsis, Vaishak Belle, and Kobi Gal. Multiplexnet: Towards fully satisfied logical constraints in neural networks. In AAAI, 2022.

Lennert De Smet, Pedro Zuidberg Dos Martires, Robin Manhaeve, Giuseppe Marra, Angelika Kimmig, and Luc De Raedt. Neural probabilistic logic programming in discrete-continuous domains. In Robin J. Evans and Ilya Shpitser, editors, Proceedings of the Thirty-Ninth Conference on Uncertainty in Artificial Intelligence,

Eleonora Giunchiglia, Alex Tatomir, Mihaela Catalina Stoian, and Thomas Lukasiewicz. Ccn+: A neurosymbolic framework for deep learning with requirements. International Journal of Approximate Reasoning, page 109124, 2024.

Connor Pryor, Charles Dickens, Eriq Augustine, Alon Albalak, William Wang, and Lise Getoor. Neupsl: Neural probabilistic soft logic. arXiv preprint arXiv:2205.14268, 2022.

Kareem Ahmed, Stefano Teso, Kai-Wei Chang, Guy Van den Broeck, and Antonio Vergari. Semantic probabilistic layers for neuro-symbolic learning. In S. Koyejo, S. Mohamed, A. Agarwal, D. Belgrave, K. Cho, and A. Oh, editors, Advances in Neural Information Processing Systems, volume 35, pages 29944–29959. Curran Associates, Inc., 2022.

**Generative models**

Emile van Krieken, Pasquale Minervini, Edoardo Ponti, and Antonio Vergari. Neurosymbolic diffusion models. arXiv preprint arXiv:2505.13138, 2025.

Ruoyan Li, Dipti Ranjan Sahu, Guy Van den Broeck, Zhe Zeng Deep Generative Models with Hard Linear Equality Constraints arXiv preprint
arrive: 2502.05416 2025

Mihaela Cătălina Stoian, Salijona Dyrmishi, Maxime Cordy, Thomas Lukasiewicz, Eleonora Giunchiglia How Realistic Is Your Synthetic Data? Constraining Deep Generative Models for Tabular Data ICLR 2024

**Induced concepts**

Alessandro Daniele, Tommaso Campari, Sagar Malhotra, and Luciano Serafini. Deep symbolic learning: discovering symbols and rules from perceptions. In IJCAI, 2023

---

### Decision · Action_Editor_2hbv · 2026-05-08

**Recommendation:** Reject

**Additional Comments:**

The required revisions are structural rather than minor, which is why a major revision rather than a minor revision is appropriate. Specifically, a future submission should:

1. Rebalance Section 3 against the rest of the paper. Reviewer NME3's suggestion is reasonable: explicit signposting of which subsections are prerequisites for the technical core, with material not used downstream either trimmed or clearly marked as optional. The current ~18:40 ratio is not defensible for a survey whose stated focus is the architectural mapping.
2. Extend the taxonomy itself — not just the bibliography — to accommodate post-2022 paradigms within the main map. Generative neuro-symbolic models, concept-bottleneck architectures, reasoning shortcuts, and hard-constraint enforcement are not "future avenues"; they are present-tense branches of the field with peer-reviewed publications at top venues. Reviewer oUL7 supplied concrete references for each. Deferring these to a "Future Avenues" paragraph does not address the structural concern.
3. Strengthen the handbook flavor with the concrete pointers Reviewer NME3 requested — specific solvers, reasoners, compilers (circuit compilers, SAT/SMT solvers, ASP/ProbLog reasoners) — and pseudocode where feasible. Without these, the handbook claim is not substantiated.
4. Improve pedagogical accessibility in the technical core, particularly Section 5, per Reviewer mAvr. If the stated audience is ML practitioners new to symbolic methods, then sections that even an expert reviewer finds too dense to recommend to that audience need rewriting.

A revision addressing these four points would, in my judgment, plausibly meet the bar for TMLR.

**Audience:**

Yes

**Audience Explanation:**

All three reviewers agreed on this point, and so do I. Reviewer oUL7 wrote that the introductory sections are "very well written" and "really useful" as an introductory primer. Reviewer NME3 described the core taxonomy as "useful and comprehensive, often with a tutorial flavor" and stated that "researchers and practitioners can clearly benefit from it." Reviewer mAvr called the work "a serious and useful piece of work that would benefit many readers." The taxonomy in Figure 1, the unifying notation across frameworks, and the SRL primer have clear value for the TMLR audience. The contribution is real; the question is whether the current form delivers on it.

**Claims And Evidence:**

No

**Claims Explanation:**

The submission's central claims are that it provides (i) a comprehensive map of the neuro-symbolic AI landscape, (ii) a handbook for ML practitioners new to symbolic methods, and (iii) a gentle introduction to SRL. The reviewers' assessments converge on the conclusion that these claims are only partially supported.

On comprehensiveness, Reviewer oUL7 documented a structural coverage gap: "2025 references: 0" with "most of the 2023–2024 citations originat[ing] from the same research group, which gives the impression that the recent literature is filtered through a narrow subset of the community." Entire branches of recent neuro-symbolic AI — generative neuro-symbolic models, concept-bottleneck architectures, reasoning shortcuts, hard-constraint enforcement — are not naturally accommodated in the proposed taxonomy, and concrete references for each were supplied. The authors' rebuttal — that the survey "has been in the works for a couple of years and was first made available in 2023 on arXiv" — is, if anything, an argument against acceptance rather than for it. A handbook submitted in 2025 is evaluated against the field as it stands in 2025, not as it stood when drafting began. The maturity argument the authors apply to generative neuro-symbolic AI ("the topic is still too immature to justify a full subsection") is also weakened by the list of peer-reviewed venues (NeurIPS 2022, ICML 2023, ICLR 2025, NeurIPS 2024, NeurIPS 2025) that Reviewer oUL7 supplied in the follow-up.

On the handbook claim, Reviewer mAvr — despite recommending Accept — was explicit: "even though the authors position the paper as an useful introduction for machine learning researchers that are not familiar with symbolic methods, I would not recommend this review for this group as I think many parts are not didactic enough for newcomers." On Section 5: "the description of Monolithic Frameworks is very dense. I do not think it is an useful introduction [...] I wonder if it were better to just leave it out." On the authors' positioning claim: "Unfortunately, I disagree with this claim. The paper would be hard for newcomers, while it would be really useful for people already familiar with some neurosybolic systems."

On the SRL primer, two reviewers independently flagged the disproportionate length and weight of Section 3. Reviewer NME3 was explicit: "Section 3 on SRL is too lengthy (18 pages out of the 40 in total) [...] having such a lengthy introduction to SRL in a NeSy survey/handbook does not seem very useful."

The rebuttal repeatedly invokes the differences across the three reviews as grounds for making no structural change — the claim being, in essence, that no single revision could satisfy all viewpoints simultaneously. I do not accept this framing. It is the responsibility of the authors to deliver a manuscript that meets the bar; it is not the responsibility of the reviewers, or of the journal, to harmonize their views into a single instruction the authors can then execute. Small disagreements are normal and can reasonably be navigated. The disagreements here are neither small nor peripheral: they concern the proportion of the paper devoted to background, the scope and recency of the taxonomy, and whether the stated audience can actually follow the technical core. These are structural to what a survey is and to what it claims to deliver. A rebuttal that treats such concerns as cancelling each other out, rather than engaging with each on its merits, does not move the manuscript closer to the bar.

**Resubmission Of Major Revision:**

The authors may consider submitting a major revision at a later time.